# A Synoptic Review of the Cartilaginous Fishes (Chondrichthyes: Holocephali, Elasmobranchii) from the Upper Jurassic Konservat-Lagerstätten of Southern Germany: Taxonomy, Diversity, and Faunal Relationships

Eduardo Villalobos-Segura [1,*], Sebastian Stumpf [1], Julia Türtscher [1,2], Patrick L. Jambura [1,2], Arnaud Begat [1,2], Faviel A. López-Romero [1], Jan Fischer [3] and Jürgen Kriwet [1,2]

1   Evolutionary Morphology Research Group, Department of Palaeontology, Faculty of Earth Sciences, Geography and Astronomy, University of Vienna, Josef-Holaubek-Platz 2, 1090 Vienna, Austria
2   Vienna Doctoral School of Ecology and Evolution (VDSEE), University of Vienna, Djerassiplatz 1, 1030 Vienna, Austria
3   Urweltmuseum GEOSKOP/Burg Lichtenberg (Pfalz), Burgstraße 19, 66871 Thallichtenberg, Germany
*   Correspondence: elasmo177@gmail.com or eduardo.villalobos.segura@univie.ac.at

**Abstract:** The Late Jurassic-Early Cretaceous (164–100 Ma) represents one of the main transitional periods in life history. Recent studies unveiled a complex scenario in which abiotic and biotic factors and drivers on regional and global scales due to the fragmentation of Pangaea resulted in dramatic faunal and ecological turnovers in terrestrial and marine environments. However, chondrichthyan faunas from this interval have received surprisingly little recognition. The presence of numerous entire skeletons of chondrichthyans preserved in several localities in southern Germany, often referred to as Konservat-Lagerstätten (e.g., Nusplingen and the Solnhofen Archipelago), provides a unique opportunity of to study the taxonomic composition of these assemblages, their ecological distributions and adaptations, and evolutionary histories in detail. However, even after 160 years of study, the current knowledge of southern Germany's Late Jurassic chondrichthyan diversity remains incomplete. Over the last 20 years, the systematic study and bulk sampling of southern Germany's Late Jurassic deposits significantly increased the number of known fossil chondrichthyan genera from the region (32 in the present study). In the present work, the fossil record, and the taxonomic composition of Late Jurassic chondrichthyans from southern Germany are reviewed and compared with several contemporaneous assemblages from other sites in Europe. Our results suggest, *inter alia*, that the Late Jurassic chondrichthyans displayed extended distributions within Europe. However, it nevertheless also is evident that the taxonomy of Late Jurassic chondrichthyans is in urgent need of revision.

**Keywords:** Chondrichthyes; diversity; biogeography; Kimmeridgian; Tithonian; Late Jurassic; Solnhofen Archipelago; Wattendorf; Nusplingen; Germany

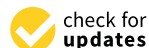



## 1. Introduction

During the Late Jurassic, extensive tropical to subtropical carbonate platforms extended from Southern France via Switzerland to Southern Germany. These platforms corresponded to the northern shore of the Tethyan Ocean [1–3] and include the limestones outcrops in Southern France (Canjuers and Cerin: [4,5]), Switzerland (Southern Jura Mountains: [6], Baden-Württemberg (Nusplingen: [7]), and the Southern Franconian Alb in Bavaria, often collectively referred to as 'Solnhofen Plattenkalk' or Solnhofen Archipelago (see [8,9]). Long known for their exceptionally preserved fossils (e.g., [10,11]), these localities yielded specimens that allow studying the anatomy and morphology of fossil organisms in detail that generally is only rarely possible (e.g., [12–14]). The nearly 400 years of constant collection and study makes these Plattenkalks one of the better-documented

fossil deposits in the world, placing them among the most prominent, diverse, and remarkable fossil Lagerstätten for the Late Jurassic globally. Furthermore, the presence of several contemporaneous sites with similar carbonate platform settings such as Cerin and Canjuers in France (see [4,5]), La Casita Formation in Mexico [15], and Vaca Muerta Formation in Argentina [16] allow the unique opportunity to investigate a potential biodiversity hotspot in deep time and place the Solnhofen Archipelago biota in a global context.

The presence of exceptional and mostly holomorphic cartilaginous fishes in these Plattenkalks, of which the fossil record normally consists of non-cartilaginous remains, e.g., teeth, denticles, and fin spines only allows to establish their cranial and postcranial morphology in detail and contributes so to our understanding of their early evolution. The Kimmeridgian-Tithonian chondrichthyan faunas (chimaeras, sharks, rays) of Southern Germany and France are among Europe's most diversified Late Jurassic assemblages, comprising about 30 species in 20 genera including representatives of all major living elasmobranch orders (e.g., Heterodontiformes, Orectolobiformes, Lamniformes, Carcharhiniformes, Hexanchiformes, and Squatiniformes), but also of extinct clades (†Hybodontoidea, †Synechodontiformes, and †Protospinacidae), together with the presence of representatives of the holocephalian order Chimaeriformes. Consequently, studying these Plattenkalk associations provides valuable insight into their community structures, faunistic dynamics, and early evolution of elasmobranch traits. However, it also provides a solid basis for understanding the impact of the J/K boundary, which was a critical evolutionary time interval [17] on elasmobranch associations and allows analysing the biological signal and importance of Konservat-Lagerstätten for diversity analyses in deep time.

The present study represents a synoptic review of the chondrichthyan faunas of the Upper Jurassic Konservat-Lagerstätten of Europe, with an update of previous studies on Nusplingen and Solnhofen Archipelago elasmobranchs presented by Kriwet and Klug [18,19]. The specific goals thus are to (1) summarise our current knowledge but also to provide novel information about the taxonomy and systematic position of chondrichthyan fishes (chimaeroids, sharks, rays) from the Solnhofen Archipelago, (2) highlight uncertainties regarding various specimens and taxa that still persist, and (3) identify future research directions.

## 2. Materials and Methods

The fossil chondrichthyan material described in this synoptic review comes from the Kimmeridgian localities of Nusplingen and Wattendorf and the Kimmeridgian–Tithonian Solnhofen Archipelago (see below). It comprises predominantly holomorphic as well as partially disarticulated specimens, and to a lesser degree isolated dental remains. All material is housed in institutional collections (for abbreviations of the corresponding collections see below) and was collected in the course of the last 160 years. Unfortunately, precise provenance information is not always available, especially for specimens in historic collections from the Solnhofen Archipelago. Therefore, all specimens from the Solnhofen Archipelago are considered as coming from a single site in the faunal relationship analysis.

### 2.1. Systematic and Taxonomic Considerations

We follow here the more traditional view of Compagno [20,21] that considers all extant sharks and rays as well as their extinct relatives that fall phylogenetically within this group (crown-group concept) to be members of the Neoselachii, conversely to Maisey [22], who considers Neoselachii to be equivalent with Elasmobranchii leaving the phylogenetic position of †Hybodontiformes ambiguous. The systematic arrangement of Holocephali follows Stahl [23] and that of Neoselachii follows Cappetta [24], Thies and Leidner [25], and Kriwet and Klug [19].

### 2.2. Methodological Approaches

For analysing the faunal relationships of the Solnhofen Archipelago chondrichthyans, we combined the results from this review with the information gathered from the literature regarding the presence of genera throughout the Late Jurassic occurring in 10 localities

(Supplementary Table S1). The faunal composition was compared across the localities, employing the Sorensen index of dissimilarity [26]. The analysis was carried out with the R package 'eco.dist' [27] using the Unweighted Pair Group Method with Arithmetic Means (UPGMA) as the clustering algorithm. The number of clusters from the resulting dendrogram was estimated using the average silhouette width [28], with the function 'find_k' in the R package 'dendextend' [29]. The dissimilarity among localities was further analysed using their corresponding time bins through a non-metric multidimensional scaling (MDS) [30] to observe possible dispersion between the localities through time and within the same stages. The MDS analysis was performed with the function 'metaMDS' of the R package 'vegan' [31]. Finally, we generated a heatmap of the localities with the distance matrix for a pairwise comparison. Presence of shared taxa (here genera) between localities was determined by using the function 'heatmap.2' of the R package 'gplots' [32]. For the present analysis, the genus was used as the study unit after considering the heterogenic nature of the fossil remains and records used in the similarity/dissimilarity analysis. As such, indexes that consider abundances were also not used, considering the sampling differences across the localities included. While a rarefaction approach could have been used, we still would have to work around the heterogeneity of the fossil record across the different localities, composed of teeth, fin spines, fragmentary skeletal remains, and holomorphic specimens and its effects on the estimation of frequencies. Because of this, only the absence/presence of genera was used.

### 3. Geographical and Geological Setting

#### 3.1. Wattendorf

The Wattendorf locality is in the northern part of the Franconian Alb, about 25 km northeast of the city of Bamberg (Figure 1A) and was first mentioned in the late 19th century by von Gümbel [33], but it was not until the beginning of the 21st century that systematic excavations started at Wattendorf [34]. The locality, covering an outcrop area of about 0.3 km$^2$ [35,36], exposes a sequence of finely laminated limestones referred to the late Kimmeridgian Torleite Formation [34,37]. The limestones are interpreted as having been deposited in a shallow, locally restricted cove, which opened towards a deeper lagoonal depocenter with anoxic or dysoxic bottom-water conditions [34].

The Wattendorf locality has revealed abundant plant, invertebrate, and vertebrate fossils of exceptional preservation (e.g., [35,36,38,39]). The chondrichthyan fauna from Wattendorf, which is limited to a small number of species, has yet not been studied in any detail and has in fact been barely noticed so far [35,36].

#### 3.2. Solnhofen Archipelago

In the southern part of the Franconian Alb, Late Jurassic deposits crop out at several localities (Figure 1A), providing access to thick packages of finely laminated limestones that are famous for having produced an extraordinary diverse and exquisitely preserved plethora of marine and terrestrial plants and animals, among them the iconic early bird †*Archaeopteryx* (e.g., [40–42]). Spanning for about 3.5 million years, from the late Kimmeridgian to the early Tithonian [43,44], these limestones have been deposited in closely associated depocenters that were delimited by sponge-microbial mounds and associated coral bioherms [9,45]. This complex lagoonal environment, collectively referred to as the Solnhofen Archipelago, was situated at about 40° N at the north-western edge of the Tethys Ocean (Figure 1B). Stagnant, hypersaline bottom-water conditions coupled with rapid burial during periodic storm events probably led to the pristine fossil preservation by inhibiting decomposition and scavenging activities [3,45].

Comprising different depositional and stratigraphic settings, the limestones of the Late Jurassic Solnhofen Archipelago have been classified into different formal geological formations [37]. The vast majority of historically collected fossils comes from deposits referred to the early Tithonian Altmühltal Formation (previously known as the 'Solnhofener Plattenkalke'), which have been exploited for a variety of practical uses since Roman

times [46]. A similarly diverse array of fossils has been revealed to occur not only in limestones referred to the laterally equivalent Painten Formation [45,47], but also in those assigned to the overlying Mörnsheim and the underlying Torleite formations [48,49].

The chondrichthyan fauna from the Solnhofen Archipelago has attracted much research attention since the first half of the 19th century, resulting in an ever-growing body of literature (e.g., [18,19,50–60]). However, as most chondrichthyan material from the Solnhofen Archipelago consists of historical collections, specific stratigraphic information is lacking for most of the specimens.

### 3.3. Nusplingen

Located in the southwestern part of the Swabian Alb, the Nusplingen locality covers an outcrop area of about 2.5 km$^2$, exposes a 10 to 15-metre-thick section of finely laminated limestones [61,62]. This sedimentary succession, which is dated to the late Kimmeridgian [43,44], was deposited in a locally restricted, less than 100-metre-deep lagoonal depocenter with stagnant water conditions at the seafloor [63].

Systematic excavations at the Nusplingen locality, conducted by the Staatliches Museum für Naturkunde Stuttgart since 1993, have uncovered a highly diverse assemblage of more than 400 fossil plant, invertebrate and vertebrate species, making it one the most productive Jurassic fossil sites in the world [62]. The chondrichthyans from the late Kimmeridgian Nusplingen locality have been subject to a number of studies since the end of the 19th century (e.g., [64–71]) and to date, two holocephalians and seven elasmobranchs have been identified.

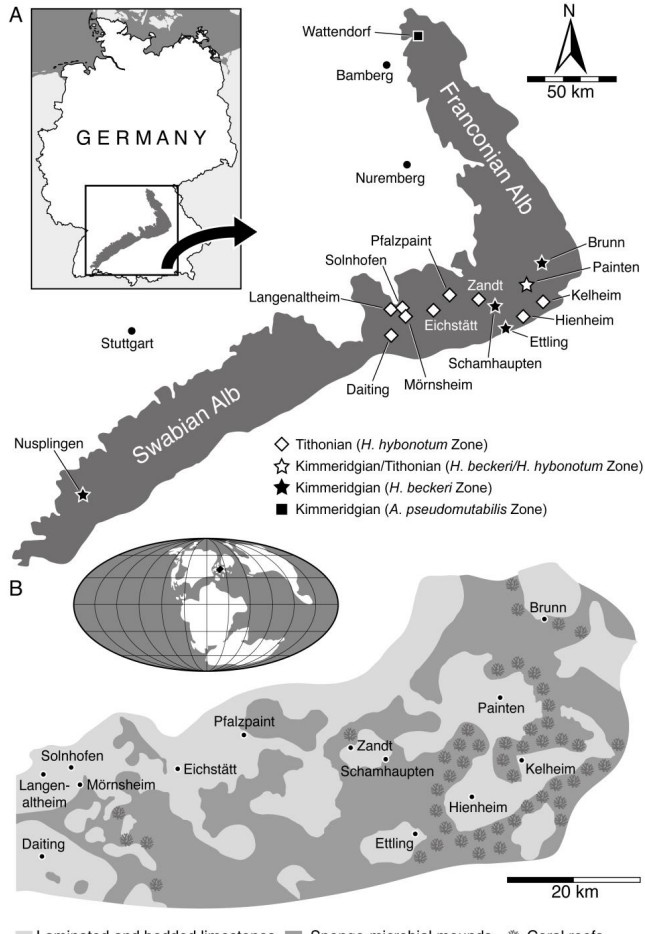

**Figure 1.** (**A**) Geographic and (**B**) palaeogeographic location maps (modified from [34,72]; biostratigraphic information based on [44,73]).

## 4. Taxonomic Review

### 4.1. Holocephali

Holocephalians are an ancient and peculiar group of cartilaginous fishes with a fossil record extending almost 360 million years back into the Middle Devonian, ca. 385 Ma [74]. Like most chondrichthyans, holocephalians present prismatic cartilaginous skeletons, but they differ from the latter in various anatomical features such as presence of a holostylic jaw suspension (upper jaw firmly fused to the neurocranium), a flap of connective tissue covering their gill openings, and an un-constricted notochord throughout life (also present in extant hexanchiforms). Additionally, their calcified axial skeleton presents characteristic annular and acellular structures, which do not represent proper vertebral centra or neural and haemal arches. A massive single element of the axial skeleton, a so-called synarcual, is developed posterior to their neurocranium, which presumably serves to dissipate forces for the sometimes-enormous pressures acting on the anterior axial skeleton during prey capture and manipulation [19,75,76]. Like in elasmobranchs, the pelvic girdle in male holocephalians presents modified structures called claspers for internal fertilization and also displays paired prepelvic tenacula. Males additionally sport a frontal tenaculum that is an elongated, club-like structure with densely arranged denticles at the anterior end that presumably holds tight to the female's pectoral fin during copulation [75].

Extant holocephalians only represent a minor portion of the total holocephalian diversity through time. Their fossil record suggests the establishment of all major holocephalian's morphological traits at least 300 Ma [77] and that the peak of their diversity occurred in the Carboniferous (approx. 359–299 Ma), followed by a significant decline with the final extinction of most Palaeozoic forms during the Permian/Triassic extinction event [78]. The survival of holocephalians across the Permian/Triassic extinction event probably was facilitated by holocephalians taking refuge or completely adapting to deep-sea conditions [78], which also supports the scarce fossil record of holocephalian remains in shallow water deposits during the early Mesozoic.

It was not until the Jurassic that holocephalians underwent a major evolutionary transition, with the successive replacement of plesiomorphic representatives of the squalorajoids and myriacanthoids by more advanced chimaeroids (to which all living holocephalians belong) [79].

Chimaeroid tooth plates, fin spines, rare egg cases, partially preserved skeletons as well as many holomorphic specimens, are commonly found in the Jurassic Plattenkalk deposits of the Southern Franconian Alb (Solnhofen Archipelago) and Southwestern Swabian Alb (Nusplingen) in Southern Germany but have not been reported from other contemporaneous Plattenkalk deposits such as Cerin [80,81] or Canjuers [4] in France up to now. Notably, dentitions of these 'modern' chimaeroids are reduced, consisting of two pairs of hypermineralized tooth plates in the upper jaw and one in the lower jaw, which is not permanently replaced as in elasmobranchs but grows throughout their life. Such a grinding dentition indicates a durophagous feeding adaptation targeting predominantly (but probably not exclusively) hard-shelled prey as in their modern counterparts [82].

The holomorphic specimens found in the Plattenkalk deposits of Southern Germany resemble living taxa presenting elongated and laterally flattened bodies, with a slender vertebral spine that supports the anterior margin of the first dorsal fin and articulates with the synarcual. Moreover, these specimens present elongated, whip-like caudal fins, large pectoral fins, comparatively small pelvic fins, two dorsal fins with the first one being high and triangular, while the second one being low and elongated, and a dorsal fin spine which most likely was connected basally to a venom gland like in all extant chimaeroids [75,83].

### 4.1.1. Chimaeriformes

Class Chondrichthyes Huxley, 1880 [84]
Subclass Subterbranchialia Zangerl, 1979 [85]
Superorder Holocephali Bonaparte, 1832 [86]
Order Chimaeriformes Obruchev, 1953 [87]

Suborder Myriacanthoidei Patterson, 1965 [88]
Family Chimaeropsidae Patterson, 1965 [88]
Genus *Chimaeropsis* Zittel, 1887 [89]
†*Chimaeropsis paradoxa* Zittel, 1887 [89] and †*C. franconicus* Münster, 1840 [90]

In the 19th century, many chimaeroid species were founded based on isolated remains or partially to completely preserved specimens from the famous Plattenkalk deposits of the Solnhofen Archipelago and Nusplingen. Perhaps the very first remains of a holocephalian from the Plattenkalks of the Franconian Alb, an isolated fin spine measuring ca. 13 cm in total length, was described by Münster [90] (pl. 3, pl. 4, figure 1), who interpreted it as a new species of extinct myriacanthoid holocephalian, †*Myriacanthus franconicus* (Figure 2). The specimen was recovered from Late Jurassic sediments at Streitberg, close to the castle Rabenstein in Franconia and donated in 1839 or 1840 to the 'Kreis-Naturalien-Cabinet' in 1840 [91], which was founded in 1832 by Graf Georg zu Münster. The specimen was mentioned one last time by Giebel [92] and subsequently considered to have been lost [93]. The 'Kreis-Naturalien-Cabinet' was re-named into 'Urwelt-Museum—Oberfränkisches Erdgeschichtliches Museum Bayreuth' in 1964 and it was possible to re-locate the specimen in the collection of this museum, where it is deposited under collection number BT 5160.00.

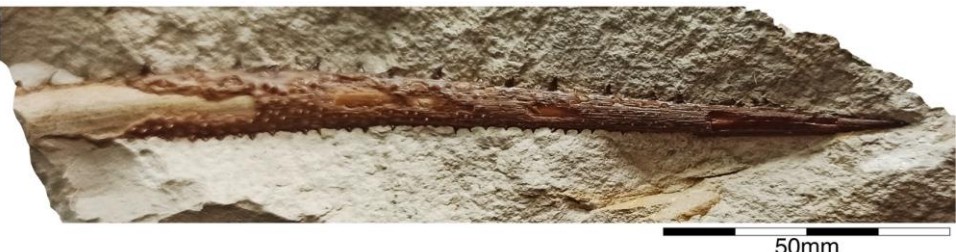

**Figure 2.** Dorsal fin spine of the myriacanthoid, †*Chimaeropsis? franconicus* (BT 5160.00) from the early Tithonian? (Upper Jurassic) of Streitberg near the castle ruin of Rabenstein.

A few years later, von Meyer [83,94] (pl. 8, figure 1) (see also Riess [95] (pl. 4, figure 10)) presented the first disarticulated dentition associated with some very poorly preserved other skeletal elements of a holocephalian from the Franconian Alb, which is stored in the Bayerische Staatssammlung für Paläontologie und Geologie in Munich (Bavaria) under number SNSB-BSPG AS I 1330 (Figure 3). Von Meyer was uncertain about the kind of animal to which these remains might have belonged. Finally, he concluded that these remains belonged to a turtle, never considering these elements to represent dental remains. The true nature of these remains as dental parts of a holocephalian was revealed when the German palaeontologist Karl Alfred von Zittel described an almost complete chimaeroid fossil measuring ca. 1 m in length from the Eichstätt area in the Solnhofen Archipelago [89]. Unfortunately, he did not provide detailed descriptions and only figured the dentition [89] (figure 126). He, nevertheless, recognized characteristic morphological features allowing him to identify it as a new taxon, which he named †*Chimaeropsis paradoxa*. By comparing its dentition with that of von Meyer [94], he concluded that the latter represents the jaw elements of a smaller specimen of †*Chimaeropsis paradoxa*. Riess [95] (pl. 2, figures 91-3; pl. 3, figures 1–9) subsequently presented additional drawings of this specimen. Based on his observations, he concluded that the isolated fin spine of †*Myriacanthus franconicus* actually has to be assigned to †*Chimaeropsis paradoxa*. Woodward [96] also supported its inclusion in †*Chimaeropsis*. However, the latter author considered the fin spine as a remain of a different species without providing any explanation.

†*Chimaeropsis paradoxa* is the stratigraphically youngest known member of myriacanthoid holocephalians and is very rare in the Late Jurassic Plattenkalks. So far, it has only been reported in the deposits of the Southern Franconian Alb. Unfortunately, the single holomorphic specimen and the isolated fin spine were lost during World War II, so our knowledge of this species is limited to the studies mentioned above and a few preserved

tooth plates. However, Lauer et al. [97] presented a new holomorphic specimen (measuring 730 mm in length) in part and counterpart of this very rare and last myriacanthoid. The remains are currently housed in the Lauer Foundation for Paleontology, Science and Education collection under number LF 2317 (currently under study by C. Duffin, Surrey, UK). Additional, well-preserved remains are housed in several private collections (see e.g., [98]).

†*Chimaeropsis paradoxa* is easily differentiated from all other Late Jurassic Plattenkalk holocephalians by the presence of a tuberculated fin spine with anterior rows of well-developed but irregular denticles [90], and by the presence of four pairs of tuberculated dermal plates at the posterior margin of the skull roof. It also presents the typical myriacanthoid dentition (paired anterior and posterior upper tooth plates, paired lower mandibular tooth plates, and unpaired lower symphyseal tooth plate) and a reduced squamation consisting of small, conical to star-like scales. A series of enlarged scales are present along the frontal midline of the skull [50,95]. The nature of the caudal fin remains dubious to some extent, but it seemingly is protocercal (diphycercal), with at least a small hypocaudal lobe preserved in specimen LF 2317. Lauer et al. [97] also mentioned the lack of a posterior row of denticles on the dorsal fin spine, which corresponds to the descriptions of Zittel [89] and Riess [95] of the lost holotype. Conversely, the isolated fin spine described by Münster [89] displays well-developed and irregular denticles along its posterior margin, which would support Woodward's [96] assumption that the isolated fin spine described by Münster [90] represents a different species, †*Chimaeropsis franconicus*, as already hypothesized by Patterson [88]. This, however, requires detailed studies of the fin spines of both the articulated specimen in the Lauer Foundation collection and the isolated fin spine in the Urwelt-Museum.

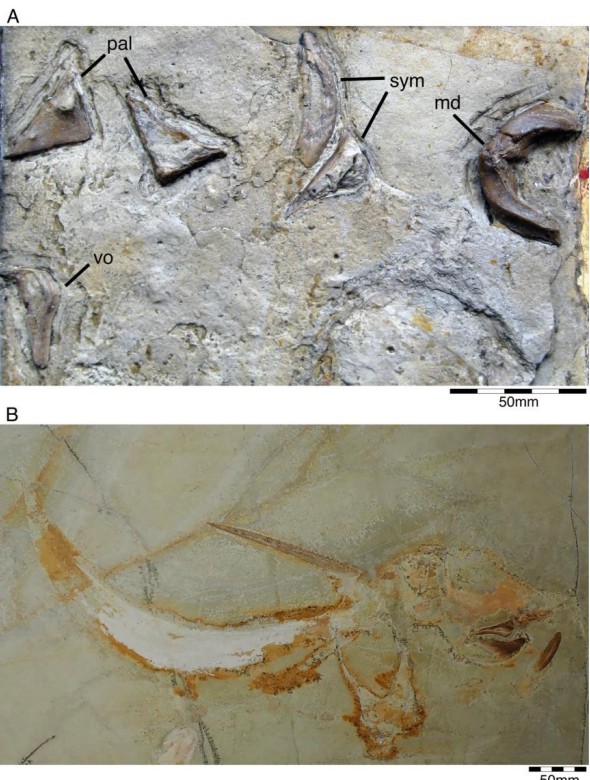

**Figure 3.** †*Chimaeropsis paradoxa* von Zittel, 1887 [89]. (**A**) Fragmentary cranial remains (SNSB-BSPG AS I 1330) consisting mainly of dental parts from the early Tithonian of Solnhofen. This specimen was figured by von Meyer in Münster [50] (pl. 8, figure 1) and Riess [93] (pl. 3, figure 10). Abbreviations: md, mandibular tooth plates; sym, symphyseal tooth plates; pal, palatinal tooth plates; vo, vomerine tooth plate. (**B**) Holomorphic specimen (LF 2317) from the early Tithonian (Upper Jurassic) of Eichstätt. This specimen also was figured by Duffin et al. 98 (figure 9C). Photo provided and reproduced with permission by B. Lauer (Lauer Foundation for Paleontology).

Suborder Chimaeroidei Patterson, 1965 [88]
Family Callorhynchidae Garman, 1901 [99]
Genus †*Ischyodus* Egerton, 1843 [100]
†*Ischyodus egertoni* (Buckland, 1936 [101])

Most holocephalians found in the Plattenkalks of the Southern Franconian Alb are all members of advanced chimaeroids and most specimens can be assigned to the extinct genus †*Ischyodus*. Currently, 15 species assigned to †*Ischyodus* have been identified from the Upper Jurassic of Europe [23]. Four of these species, †*Chimaera (Ischyodus) quenstedti* Wagner, 1857 [102], †*I. schuebleri* Quenstedt, 1858 [103], †*I. suevicus* Philippi, 1897 [104] and †*Chimaera (Ganodus) avitus* Wagner, 1862 [105] were reported from the Kimmeridgian Plattenkalks of Nusplingen, while †*I. schuebleri*, †*C. (I.) quenstedti* and †*C. (G.) avitus* also occur in the Tithonian of the Franconian Alb.

Quenstedt [106] noted a complete dentition, fin spine, and the outline of the dorsal fin and the lateral line system on a relatively incompletely preserved skeletal remain of a holocephalian from the Franconian Plattenkalks in the private collection of the country doctor, Carl Friedrich Häberlein (1828–1871) of Pappenheim. However, he did not provide a detailed morphological description, any figure or taxonomic interpretation. Friedrich Häberlein sold most of his fossils, including the famous first skeleton of †*Archaeopteryx*, to the Natural History Museum in London (UK). However, he sold the holocephalian specimen, which was mentioned by the naturalist Quenstedt [106] to the Bayerische Staatssammlung für Paläontologie und Geologie. Wagner [102] finally presented a short account without figure of this first skeleton of a holocephalian from the Upper Jurassic Plattenkalks of Southern Germany, which he named *Chimaera* (†*Ischyodus*) *quenstedti* in honour of Friedrich August Quenstedt (1809–1889), who first made the specimen public. Wagner [107] subsequently presented a more detailed description of this unique specimen, including a figure of the fin spine. Based on this, Riess [95] (pl. 1, figures 1–5; pl. 2, figures 1–7) considered †*Ischyodus* to be the correct genus name (for a genus diagnosis of †*Ischyodus* (see [108]) and accordingly transferred the species to it, providing additional descriptions, especially of the dentition, accompanied by corresponding figures including also one of the holomorphic specimens. However, this figure is very blurred, so not much information could be retrieved from it.

Sadly, the holotype of this species was destroyed during World War II, when the Bayerische Staatssammlung für Paläontologie und Geologie in Munich, where the specimen was housed, was bombed in the last weeks of the war. A cast of the dorsal fin spine, which is housed in the Natural History Museum London (NHMUK P 38005) is the only part of the holotype that endured [79]. Unfortunately, we were not able to locate this cast under the mentioned collection number. Nevertheless, additional specimens, including isolated tooth plates and fin spines, were recovered from the Plattenkalks of the Solnhofen Archipelago and Nusplingen (Figure 4).

According to Wagner's [102,107] and Riess's [95] descriptions and figures, the holotype specimen was approx. 1.5 m long, displayed a well-preserved dentition and denticles, while the neurocranium, dorsal fin spine and appendicular skeleton were incomplete. Wagner [102] also indicated the presence of C-shaped circumchordal rings in the trunk region, which Riess [95] interpreted as scales that partly enclosed the lateral line canal. These scales (see [88,109]) surrounding the open lateral sensory line are a characteristic feature in all extant chimaeroids [110].

An additional specimen housed in the Lauer Foundation (LF 1369) of exquisite preservation displaying a fleshy and long, tapering snout protruding well anterior to the head on which the sensory lines extend. The dorsal fin spine is very long and unornamented, and the caudal fin is heterocercal conversely to the protocercal caudal fin in †*Chimaeropsis paradoxa*.

Consequently, †*Ischyodus* represents an extinct genus of plough-nosed chimaeras (Callorhinchidae) based on all available morphological information. The Callorhinchidae includes the extinct genera †*Brachymylus*, *Edaphodon*, *Pochymylus*, and the extant *Callorhinchus* (the elephant-nosed chimaeras; also presented in the fossil record). Presumably, †*Ischyodus*

thrived mainly in deeper waters of the oceans and only migrated into shallower, near-shore waters for reproduction, similar to some extent callorhinchids and the extant *Chimaera monstrosa*. Sporadic records of †*Ischyodus* (only isolated remains) also were reported from shallow marine deposits of England, Germany, France, and Spain (e.g., [108]).

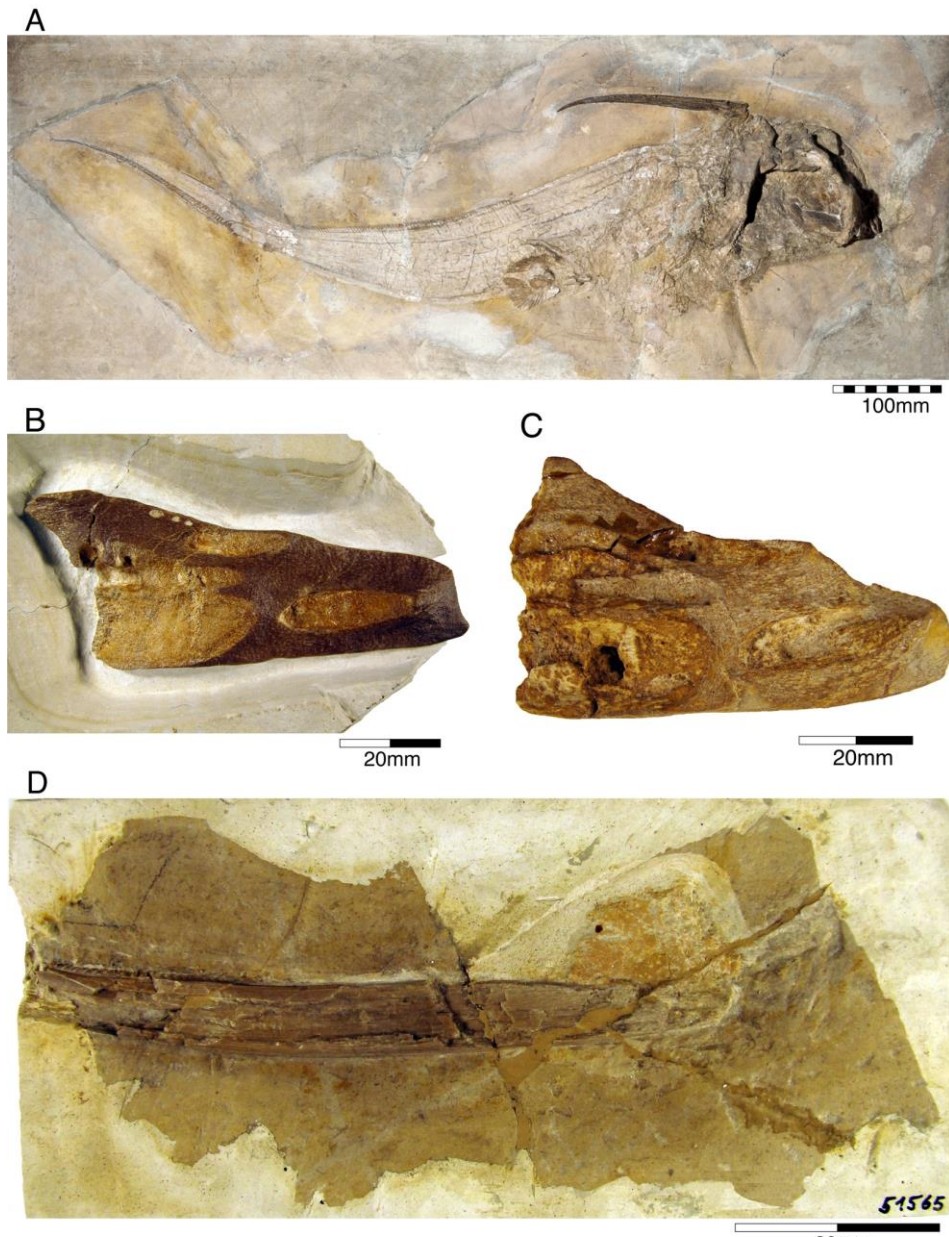

**Figure 4.** †*Ischyodus egertoni* (Buckland, 1936 [101]). (**A**) Holomorphic specimen (SNSB-BSPG 1954 I 366) from the early Tithonian Altmühltal Formation of Eichstätt, labelled as †*I. quenstedti*. This specimen was figured by Kriwet and Klug [19] (figure 675). (**B**) Right palatinal tooth plate (SMNS 96078/28) from the late Kimmeridgian Nusplingen Plattenkalk of the quarry of the Gesellschaft für Naturkunde Württemberg), labelled as †*I. quenstedti*. (**C**) Right palatinal tooth plate (SMNS 51566) from the upper Kimmeridgian of Nusplingen, labelled as †*I. schuebleri*. This specimen was collected by Fraas in 1855 and figured by Schweizer [66] (pl. 8, figures 5 and 6). (**D**) Isolated fin spine (SMNS 51565) from the upper Kimmeridgian of Nusplingen, labelled as †*I. schuebleri*.

Among callorhynchids, †*Ischyodus* has the longest fossil record ranging from the Middle Jurassic (Bajocian) to the Neogene (Miocene-Pliocene). However, most species

are known only by dentitions. Consequently, our knowledge of their anatomy is based almost exclusively on †*I. quendstedti* (the only species with skeletal remains). In the Upper Jurassic Plattenkalks of Nusplingen, fossil remains of this callorhynchid are more common than in the Plattenkalks of the Solnhofen Archipelago. Philippi [104] examined five slabs of fragmentary skeletal remains and additional isolated teeth, which are housed in the Paläontologische Sammlung of the Universität Tübingen (four slabs) and the Staatliches Museum für Naturkunde Stuttgart, Germany (one small slab). He concluded that the dental remains of these specimens are intermediate between †*Ischyodus quenstedti* and †*Ischyodus schuebleri* in morphology and size and assigned them to a new species, †*Ischyodus suevicus*.

Quenstedt [103] (pl. 96, figure 39) also described another chimaeroid (†*Ischyodus schuebleri*), based on a fragmentary mandibular tooth plate found on the Swabian Alb. Riess [95] described two more mandibular tooth plates of this species form Upper Jurassic Plattenkalks of Kelheim. Von Ammon [111] described associated dental plates of this species, providing additional information on the palatine tooth plates. However, vomers seemingly were not preserved.

Riess [95] also presented a detailed review of Late Jurassic chimaeroids, synonymizing the species †*I. suevicus* and †*I. rostratus* from the Tithonian of Northern Germany with †*I. schuebleri*. Heimberg [112] also supported this interpretation, who provided a detailed description of a slightly disarticulated specimen (GPIT 19192) that was recovered from the Nusplingen Plattenkalks in the early 20th century. Schweizer [66] accepted Heimberg's interpretation and also assigned an isolated fin spine from Nusplingen presented without a figure by Fraas [65] as '†*Ichtyodorulith*' to this species. According to Schweizer [66], three different chimaeroids occurred in the Plattenkalks of Nusplingen: †*I. avitus* (now considered not to be a member of †*Ischyodus*; see also below), †*I. schuebleri*, and †*I. quenstedti*.

Duffin (in Stahl [23]), conversely, considered †*I. schuebleri* a junior synonym of †*I. quendstedti*, leaving only a single species of †*Ischyodus* in the Upper Jurassic Plattenkalks of the Solnhofen Archipelago and the Swabian Alb. Nevertheless, this species seems to be very common in the Southern German Plattenkalks and a large number of specimens have been recovered up to now, which are housed in institutional (e.g., BMMS 45456, SNSB-BSPG 1954 I 366, GPIT 19192, LF 139, NHMUK 37021, SMNS 51566), but also in private collections (e.g., private collection of U. Resch: https://www.steinkern.de/praeparation-und-bergung/solnhofener-plattenkalke/1298-ischyodus.html, accessed on 15 March 2022), making †*I. quendstedti* the best-known †*Ischyodus* species from Germany, despite the loss of its holotype. Popov et al. [113], when reviewing the Kimmeridgian holocephalians from western Europe placed, †*I. schuebleri*, †*I. suevicus*, and †*I quenstedti* tentatively (with question mark) into synonymy with †*I. egertoni* from the Middle and Upper Jurassic of Southern England. A detailed comparison of the dentition of †*I. quenstedti* with that of †*I. egertoni* reveals many significant similarities between both species and we also consider both to be synonymous.

Popov and Shapovalov [114] also mention the presence of the extant long-nosed chimaeroid, *Harriotta*, in the Solnhofen Archipelago, based on a small specimen from the upper Kimmeridgian of Wattendorf in the Northern Franconian Alb, which was described and figured by Mäuser ([36,115] (figure 38); [36] (525, text-figure 1013)) as a juvenile specimen of †*Ischyodus* sp. (Figure 5). So far, a single extinct species of *Harriotta*, †*H. lehmann* Werdelin, 1986 [116] has been reported from the Upper Cretaceous limestones of Lebanon. The main identification feature of *Harriotta*'s dental plates is the presence of hypermineralized ovoids arranged in regular series, rods and blocks of hypermineralized dentine in tritors on the occlusal surface [117–119]. The dental apparatus of the Wattendorf specimen exposes both parts of ventral mandibular dentition, the posterior portions of the palates, and the anterior parts of the vomers in occlusal view. While the mandibular and palatine tooth plates do not allow any unambiguous taxonomic assignment. The two slightly anteriorly displaced vomers with their seemingly sub-rectangular outline with vertical series of small territorial pads along the labial margin and the mandibular outline suggest that this is a juvenile specimen of †*Ischyodus egertoni* (†*I. quenstedti*), instead of a member of *Harriotta*.

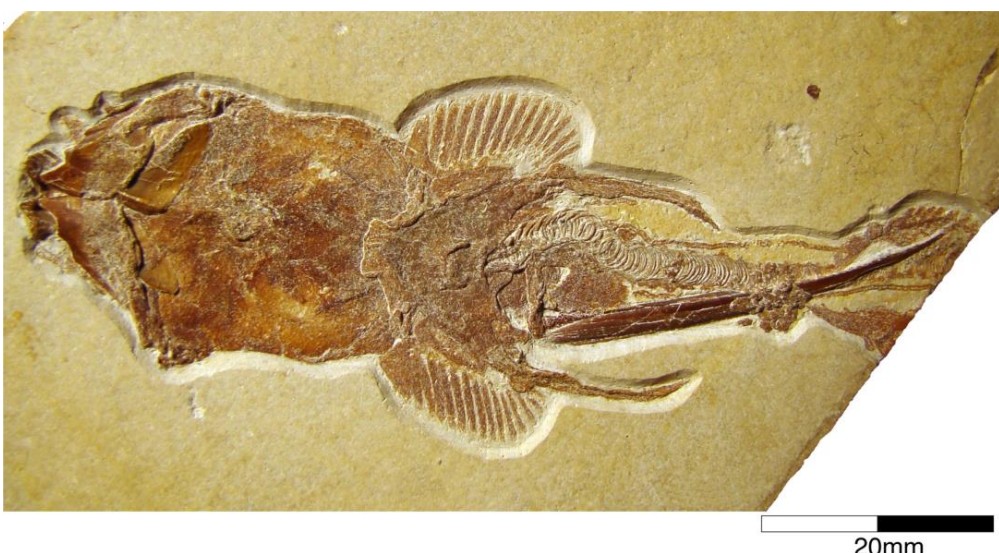

**Figure 5.** Still undescribed, juvenile specimen of †*Ischyodus egertoni* (Buckland, 1936 [101]) (NMB without number) from the late Kimmeridgian Torleite Formation of Wattendorf in the northern part of the Franconian Alb.

**Callorhynchidae indet:** Remains of other chimaeroid (SMNS 95823/4) collected in 2011 from the Kimmeridgian beds of Nusplingen were assigned to †*Elasmodectes avitus* (von Meyer, 1862 [120]) by Schweigert et al. [121]. Preserved in ventral view, the dentition of this specimen displays a distinct and strong descending lamina on the mandibular plates (Figure 6), which resembles that of †*Ischyodus* (Callorhynchidae) rather than †*Elasmodectes avitus* (Rhinochimeridae), which lacks mandibular descending laminae (see below). Duffin [122] presented detailed descriptions and discussions of this specimen and concluded that it also could not be assigned to †*Ischyodus*. A second, incomplete specimen with its dentition accessible in ventral view (SMNS 80144/22) was recovered from the Nusplingen Plattenkalks in 1994 and initially assigned to †*Ischyodus schuebleri*. However, this specimen resembles SMNS 95823/4 closely and Duffin [79,122] subsequently identified both as *Callorhynchidae* indet. A more specific identification, unfortunately, is not possible because of its incomplete nature and its tooth plates being preserved in basal view [123] (J.K. pers. obser.). Nevertheless, these two remains indicate that the diversity of chimaeroids in the Upper Jurassic of Southern Germany was greater than previously assumed.

Family Rhinochimaeridae Garman, 1901 [99]
Genus †*Elasmodectes* Newton, 1878 [124]
†*Elasmodectes avitus* (von Meyer, 1862 [120])

The third known holocephalian from the Late Jurassic Plattenkalks is †*Elasmodectes avitus*. This species occurs in the Kimmeridgian of Nusplingen and the Tithonian of the Solnhofen Archipelago. Initially, von Meyer [125] named this species †*Chimaera* (*Ganodus*) *prisca* and subsequently renamed it †*C.* (*Ganodus*) *avitus* (von Meyer, 1860 [125]). Riess [95] later transferred it erroneously to †*Ischyodus*. Recently, Duffin [79] and Lauer et al. [126] relocated this species to the genus †*Elasmodectes*, based on rigorous comparisons with Jurassic and Cretaceous chimaeroids, e.g., †*Elasmodectes willetti* Newton, 1878 [124] from the Upper Cretaceous English Chalk. The holotype (TM 6599 (part) and TM6600 (counterpart)) is a holomorphic specimen, which was long assumed to have been lost [79].

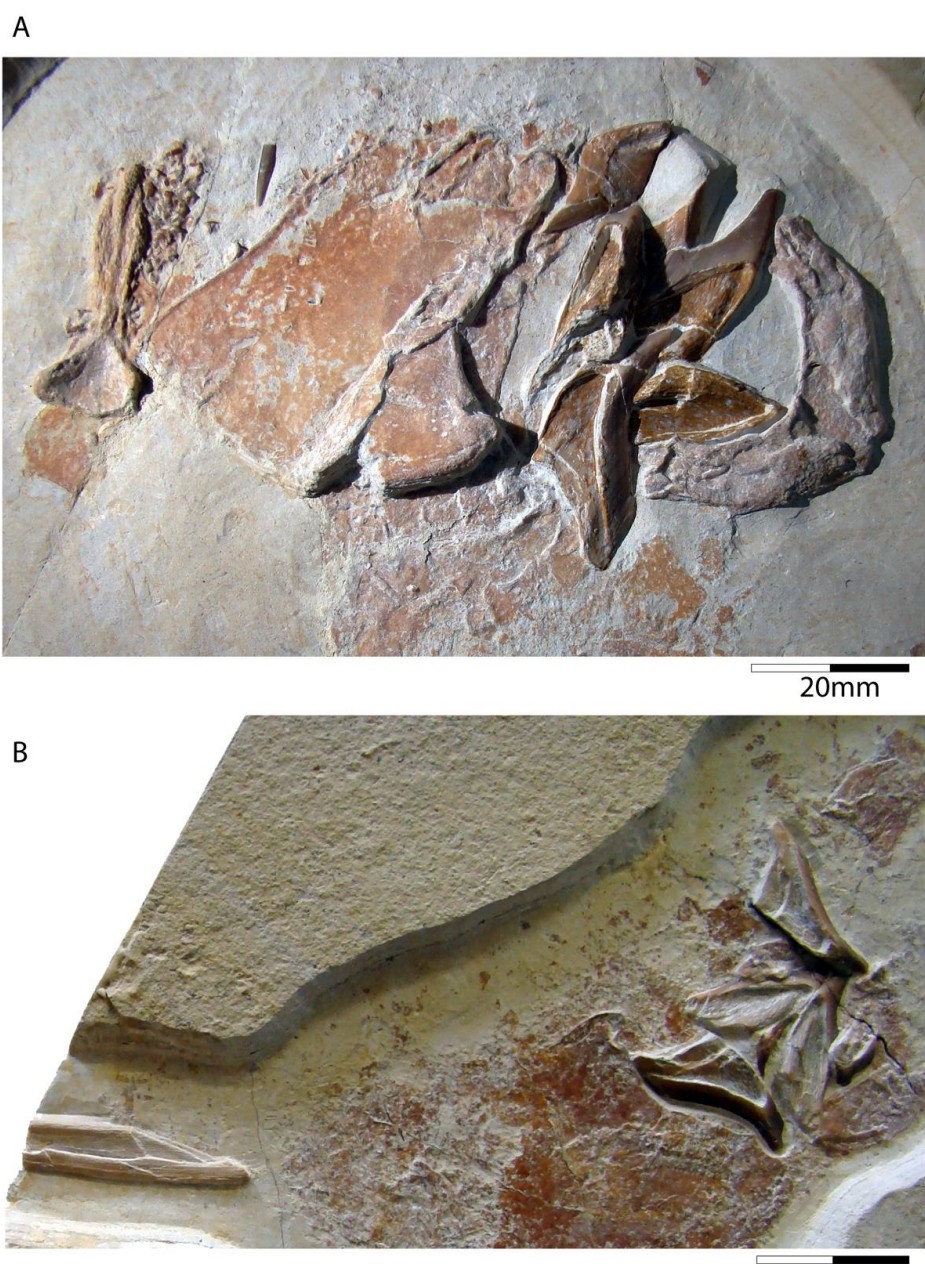

**Figure 6.** Callorhynchidae indet. (**A**) Cranial remains and basis of fin spine (SMNS 80144/22) from the late Kimmeridgian Nusplingen Plattenkalk of the quarry of the Gesellschaft für Naturkunde Württemberg, labelled as †*I. schuebleri*. (**B**) Cranial remains including the frontal clasper and complete dentition (SMNS 95823/4) from the late Kimmeridgian Nusplingen Plattenkalk of the quarry of the Gesellschaft für Naturkunde Württemberg, labelled as †*Elasmodectes avitus*. It was collected in 2011 and figured by Schweigert et al. [118] (pl. 4).

Descriptions and figures of †*E. avitus*, including the reconstruction of its dentition, are provided by Riess [95], von Ammon [111,127], Schweizer [66], and Duffin [79,122]. The numerous specimens of this chimaeroid from Nusplingen and the Solnhofen Archipelago provide abundant morphological traits for detailed analyses (e.g., JME SOS 3149a, JME SOS 4003, NMB (without number), SMNK (without number), SMNS 8387/1, SMNS 11049, SMNS 51427, SMNS 51564, SMNS 80142/16, SMNS 86901/35, SMNS 95400, SMNS 95823/4, SNSB-BSPG 1885 IX 7, SNSB-BSPG 1885 IX 8, SNSB-BSPG 1885 XI 507, SNSB-BSPG 1908 I 139a, b, SNSB-BSPG AS I 863, SNSB-BSPG AS I 864, and SNSB-BSPG AS I 865) (Figure 7).

†*Elasmodectes avitus* was a rather small chimaeroid reaching up to 50 cm in total body length. It is characterized by a large and bulky head with a fleshy and elongated snout, which, nevertheless, is shorter than in extant rhinochimaeroids; a relatively long frontal tenaculum, which is anteriorly expanded and spatulate-like, with a dense cover of posteriorly directed denticles (Figure 8), facing a small patch on the skull that also bears posteriorly directed denticles; two pairs of specialized denticles between first and second dorsal fins and three specialized denticles between second dorsal and caudal fins; second dorsal fin low and elongated; a dorsal fin spine that is unornamented (e.g., LF 2322); and a protocercal caudal fin with small epicaudal und hypocaudal lobes. The dentition is characterized by hypermineralized tritorial rods forming a sectorial (cutting) dentition consisting of paired vomerine and palate tooth plates in the upper jaw, the lack of a descending mandibular lamina and labio-lingually compressed mandibular tooth plates with beaded hypermineralized tritors, which are arranged along the occlusal crest.

†*lschyodus quenstedti* differentiates from †*E. avitus* easily by being significantly larger (max. length of 1500 mm), possessing well-developed, hypermineralized tritors on the dental plates forming a crushing dentition, and by the presence of a lingual descending lamina and a heterocercal caudal fin.

†*Elasmodectes* was previously grouped with †*lschyodus* and some other Cretaceous chimaeroids such as †*Edaphodon* into the family Edaphodontidae to distinguish extinct chimaeroids from extant forms. However, such a grouping does not correspond to a natural division based on phylogenetic principles. Duffin [79] suspects very close relationships between †*Elasmodectes* and the extant long-nosed chimaerids (Rhinochimaeridae) based on dental morphological features (lack of descending lamina and sectorial dentition). Thus, †*Elasmodectes avitus* and †*Elasmodectes secans* from the English Kimmeridgian currently represent the oldest known long-nosed chimaeras (Rhinochimaeridae). The characteristic, elongated snout of recent long-nosed chimaeras, equipped with numerous nerves serving to find prey, consists mainly of soft tissue, making it very difficult to be preserved in the fossil record. However, some specimens from the Plattenkalks of the Southern Franconian Alb present evidence of an elongated snout, in the form of structures extending far forward in front of the jaws, consisting of densely arranged calcified rings surrounding the mucous channels in the snout.

Egg Capsules of Holocephalians

Egg capsules of modern holocephalians are characterised by their spindle-shaped form with a three-fold division consisting of a bulbous central fusiform body that tapers gradually towards the end into a truncate rostral section and at the other one progressing into a long and slender caudal section. The lateral edges of the capsule are accompanied by a striking wing-like and ribbed, lateral web [128,129]. Differences in the capsule shape, width of the web as well as rib characteristics are diagnostic for the family level [23]. Up to now, fossil holocephalian egg capsules have been described from shallow marine strata of Eurasia, North America, and New Zealand [128]. So far, eleven distinguishable species [129] are summarised under the parataxonomic ichnogenus †*Chimaerotheca* Brown, 1946 [130]. Egg capsules of holocephalians are scarce in the fossil record from the Upper Jurassic Plattenkalks of the Solnhofen Archipelago. Altogether, half a dozen specimens can be found in several public and private collections (J.F. and J.K., pers. observations). Recently, they were described as †*Chimaerotheca schernfeldensis* by Duffin et al. [98], based on a double capsule in the collection of the Lauer Foundation, Illinois, and two other specimens. One specimen (NHMUK PX Z.183) is housed in the Natural History Museum London (Figure 9), where it is part of the Egerton Collection that was compiled in 1830. The other capsule was briefly described and illustrated by Reichenbach-Klinke and Frickhinger [131] and subsequently mentioned and depicted repeatedly [132–135] but is currently considered to be lost. †*Chimaerotheca schernfeldensis* has an overall length of up to 370 mm and a width of about 70 mm [98], which makes it larger than all known capsules of modern day holocephalians as well as all other †*Chimaerotheca* specimens [128]. According to the shape

of the central capsule, the membrane outline, and its ribbing characteristics, the Plattenkalk species resembles present-day rhinochimaerid egg capsules. Based on the large size and size ratios of present-day holocephalians and their egg capsules, Reichenbach-Klinke and Frickhinger [131] and Duffin et al. [98,132] already assumed that †*Ischyodus egertoni* was the most likely producer. In this respect, it should be noted, however, that †*Ischyodus* is a callorhynchid holocephalian, while the capsules show rhinochimaerid traits, at least in general comparison with present-day specimens.

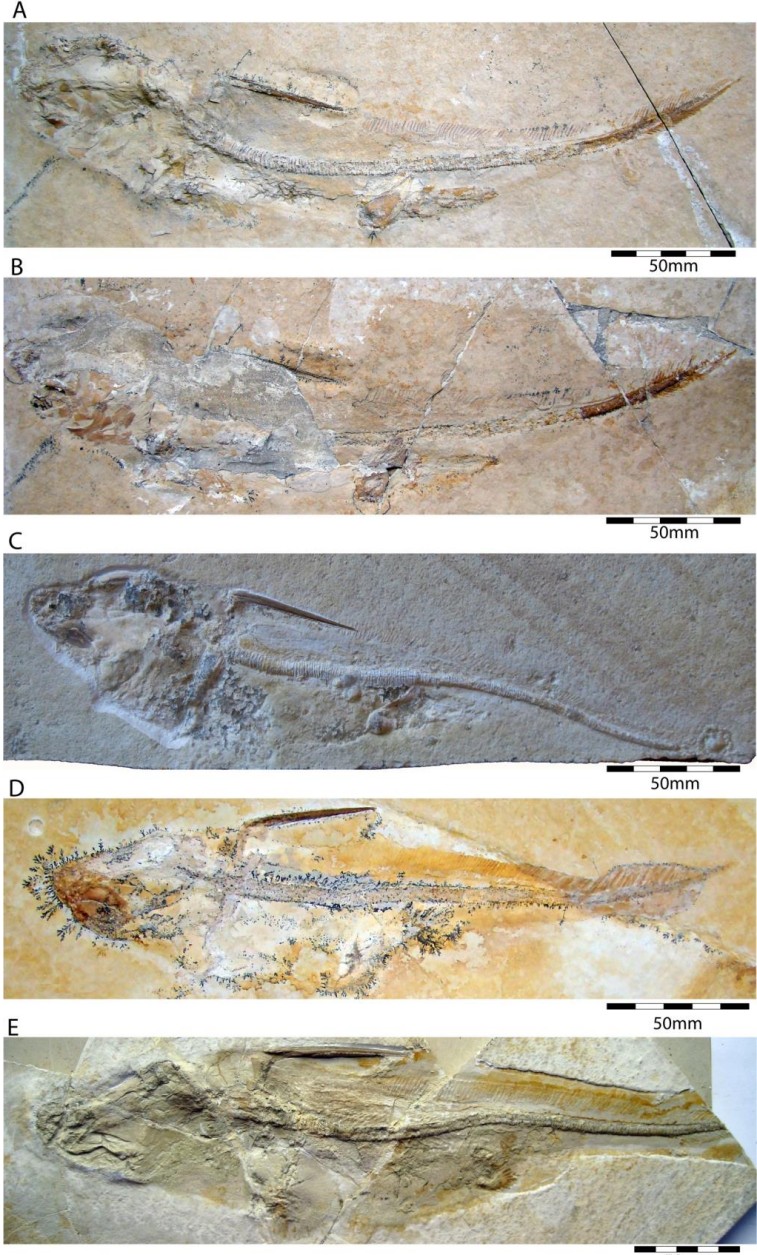

**Figure 7.** Specimens of †*Elasmodectes avitus* (von Meyer, 1862 [120]) from the Solnhofen Archipelago. (**A**) Holotype (SNSB-BSPG 1908 I 39a) from the early Tithonian Altmühltal Formation of Blumenberg near Eichstätt. (**B**) Counterpart of the holotype (SNSB-BSPG 1908 I 39b). (**C**) Male specimen (JME SOS 3149a) from the early Tithonian Altmühltal Formation of Blumenberg near Eichstätt displaying the pelvic claspers. (**D**) Holomorphic specimen (SMNK without number) displaying the well-preserved caudal fin from the early Tithonian Altmühltal Formation of Solnhofen. (**E**) Holomorphic specimen (JME SOS 4003) from the early Tithonian Altmühltal Formation of Wintershof.

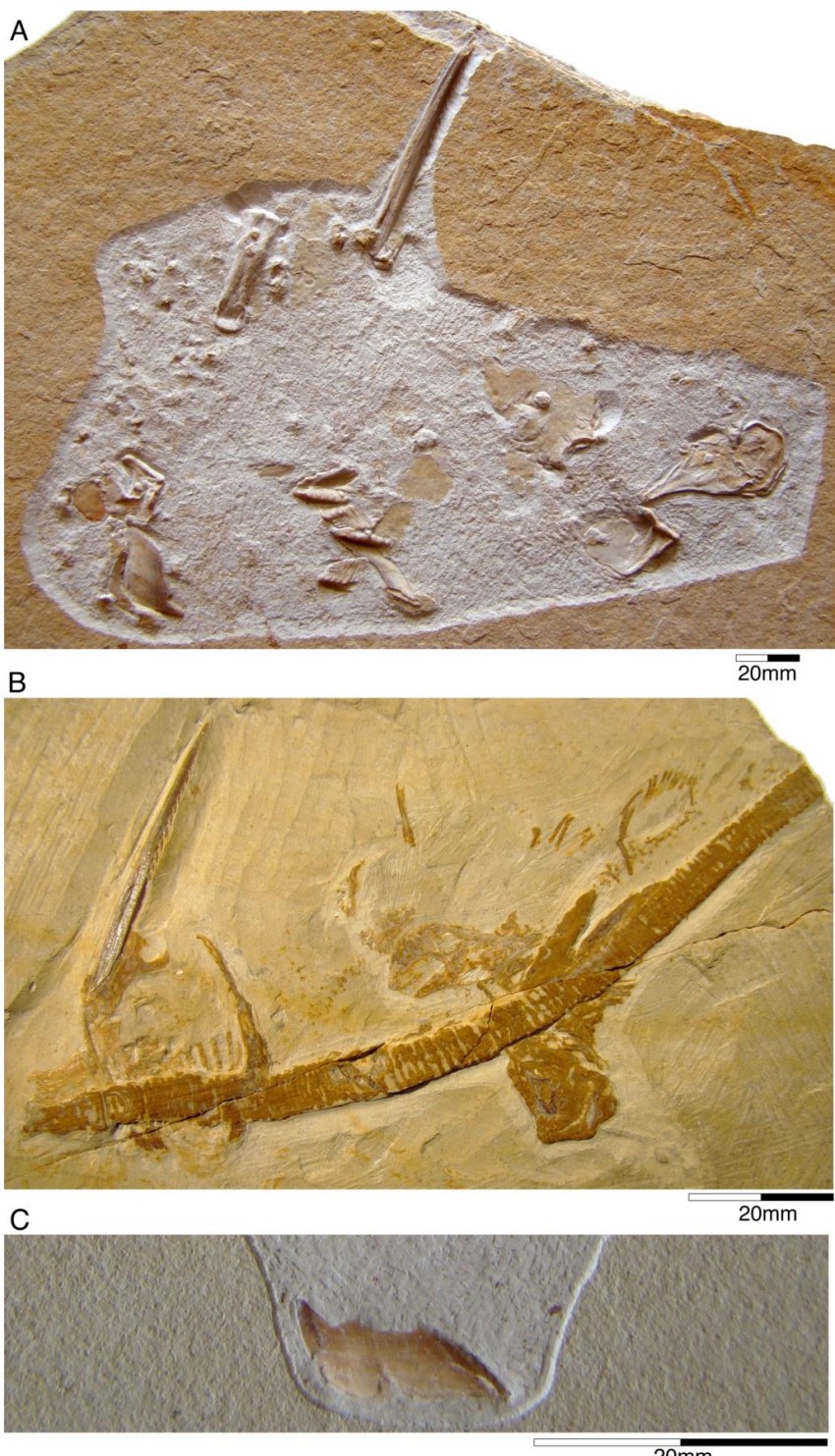

**Figure 8.** Specimens of †*Elasmodectes avitus* (von Meyer, 1862 [120]) from the upper Kimmeridgian (Upper Jurassic) of Nusplingen. (**A**) Disarticulated male specimen (SMNS 80142/16) of the quarry of Egesheim auf dem Westerberg displaying cranial remains including the frontal clasper and dorsal fin spine. This specimen was recovered in 1993 and figured by Duffin [78] (figure 4). (**B**) Partially preserved skeleton (SMNS 51564) displaying part of the vertebral column, the dorsal fins including the fin spine and parts of the pelvic girdle. The specimen was figured by Schweizer [66] (figure 14, pl. 12, figures 1–4). (**C**) Strongly disarticulated specimen (SMNS 86901/35) of the quarry of the Gesellschaft für Naturkunde Württemberg) displaying the left mandibular tooth plate.

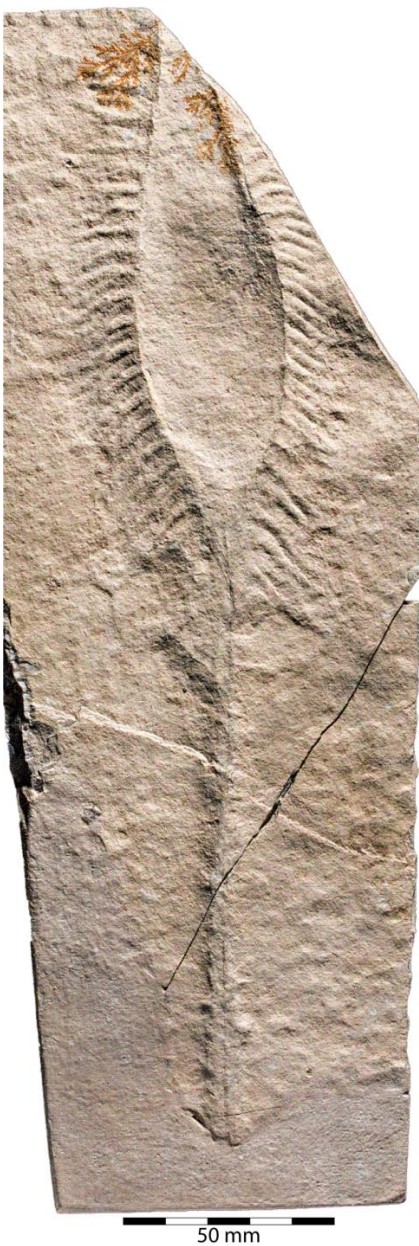

**Figure 9.** †*Chimaerotheca schernfeldensis* Duffin et al., 2021 [98], also see [132] (NHMUK PV Z.183) egg capsule morphotypes from the Solnhofen Archipelago.

*4.2. Euselachians*

Euselachians include crown-group elasmobranchs, i.e., chondrichthyans with calcified vertebral centra (sharks and rays) and their sister group, the extinct hybodontiforms as suggested by Maisey et al. [136,137] and Villalobos-Segura et al. [138], contrary to Hay [139] and Regan [140], who considered hybodonts to be a member of '*Cestracion*'-like sharks (which originally included *Heterodontus*). Consequently, they represent a relatively ancient group, with a fossil record extending back into the Early-Middle Carboniferous (approx. 350 Ma).

Euselachians are not necessarily the equivalent of the total elasmobranch group (sensu Bonaparte [141]), as currently, there are at least two contrasting views about the systematic content and characterization of the total elasmobranch group. One hypothesis excludes many chondrichthyan lineages, such as chimaeroids, symmoriiforms, ctenacanths, and xenacanths [142], and the other hypothesis excludes chimaeroids and symmoriiforms but includes ctenacanths and xenacanths [143–146].

According to Maisey [137,147] and Maisey and Lane [148], euselachians share the presence of a chondrified medial wall in the otic capsules; an endolymphatic fossa with chondrified floor; a perilymphatic fenestra; a glossopharyngeal canal, formed by closure of the embryonic metotic fissure (this fissure also is closed in modern chimaeroids); separation of the posterior semicircular canal from the labyrinth; posterior semicircular canal forms an almost complete circle; a ventral gap between the hyoid and first branchial arch [149]; all but the first hypobranchial directed posteriorly [149,150]. The presence of teeth with a complex enameloid ultrastructure with more than a single layer (a multi-layered arrange is also present in some ctenacanthiforms; see [151]); and the presence of monodontode ('non-growing') denticles, a term that is inaccurately used for isolated 'growing' denticles from Palaeozoic strata, which are very generally referred to as 'hybodontiform', but are probably chondrichthyan oropharyngeal scales rather than skin denticles [137,152]. Moreover, modern chimaeroids and several extinct holocephalians display monodontod denticles. Polyodontode denticles occur in †*Deltoptychius* [88], and 'growing' denticles (of 'protacrodontid' type) occur in some Palaeozoic stem holocephalians (e.g., [78]). This seemingly disjunct distribution suggests that monodontode denticles possibly arose independently in euselachians and holocephalians.

Maisey [153] and Maisey et al. [137] proposed another character that unites this group, which is the fusion between paired halves of the pelvic girdle to form a puboischiadic bar. However, in recent years, increasing evidence suggests that this is a misinterpretation in fossil remains, and both males and females of several hybodontiforms, present and unfused pelvic girdle (see [154]. A preliminary re-examination of the late Pennsylvanian hybodontiform, †*Hamiltonichthys*, using UV light seems to support this, but a more detailed study is needed (E.V.S. and S.S., pers. observations).

### 4.2.1. †Hybodontiformes

Subclass Elasmobranchii Bonaparte, 1838 [141]
Cohort Euselachii Hay, 1902 [139]
Subcohort indet.
Order †Hybodontiformes Maisey, 1975 [153]

Hybodontiforms are the extinct sister group of neoselachians (modern sharks and rays) and form a speciose clade of Palaeozoic to Mesozoic shark-like chondrichthyans characterized by distinct cranial and postcranial traits [136,155]. However, even after almost two centuries of extensive research, there still is no reliable phylogenetic framework for hybodontiforms yet available, thus leaving their interrelationships dubious and unresolved (e.g., [60,154,156]).

All hybodontiforms have a rather robust body with two dorsal fins supported by heavily ornamented spines, displaying numerous retrorse denticles arranged along the posterior midline [157]. One of the most conspicuous features of hybodontiforms is the presence of a single or double pair of cephalic spines on the skull behind the orbit [158]. These peculiar structures, each with a root-like base carrying a prominent hook-shaped denticle, are restricted to males, but their function remains ambiguous.

Extending for more than 290 Ma, from the Late Devonian to the Late Cretaceous, hybodontiforms boast an extensive fossil record that mainly consists of isolated teeth, which document various adaptive traits in relation to prey and feeding modes (see [24] and references therein). Hybodontiforms flourished during the Triassic and Jurassic, when they expanded into various environments, ranging from fully marine to continental settings (e.g., [159–163]). From the Early Cretaceous onwards, the diversity of hybodontiforms saw a decline, particularly in marine ecosystems, before they predominantly occurred in continental environments where they thrived until they finally vanished close to the end of the Cretaceous (e.g., [164–167]).

The European Late Jurassic hybodontiform fossil record displays a homogeneous distribution pattern dominated by large-bodied (exceeding two meters in maximum length) epipelagic forms of intermediate trophic position [168]. Conversely, small-bodied hybodon-

tiforms appear to have been rare and rather limited in their facies distribution, inhabiting predominantly marginal marine depositional environments with reduced or fluctuating salinities (e.g., [168,169]). In the Solnhofen Archipelago, hybodontiforms form a rare faunal component, whereas they are completely absent from the lithographic limestones of Nusplingen, France and Poland. So far, three hybodontiform taxa have been identified from the Solnhofen Archipelago, two of which have been named to date.

Superfamily †Hybodontoidea Owen, 1846 [170]
Family †Hybodontidae Owen, 1846 [170]
Subfamily †Hybodontinae Owen, 1846 [170]
Genus †*Hybodus* Agassi, 1837 [171]
†*Hybodus fraasi* Brown, 1900 [172]

†*Hybodus fraasi* is a poorly known, medium-sized species that has originally been described and named by Brown [172] based on an almost complete but rather poorly preserved female skeleton from the lower Tithonian of Solnhofen (Figure 10). Maisey [173] tentatively transferred †*H. fraasi* to the genus †*Egertonodus*, originally introduced for †*Hybodus basanus* Egerton, 1844 [174], from the Lower Cretaceous of England. However, this taxonomic scheme was rejected by other authors [19,175], particularly due to the poor preservation of the holotype. The sediments of the Solnhofen Archipelago yielded two additional skeletons that have been referred to as †*H. fraasi* [19,176] (Figure 10), but these identifications need further scrutiny pending their detailed morphological analysis.

Subfamily †Acrodontinae Casier, 1959 [177]
Genus †*Asteracanthus* Agassiz, 1837 [171]
†*Asteracanthus ornatissimus* (Agassiz, 1837 [171])

With a body length exceeding two meters, †*Asteracanthus* Agassiz [171] certainly is one of, if not the largest chondrichthyan known to have roamed the Solnhofen Archipelago. The genus is represented by a single, almost complete skeleton from the lower Tithonian of Solnhofen representing a female individual (Figure 11). The specimen was first figured by [19], who assigned it to †*Hybodus obtusus* Agassiz, 1839 [171], a species now considered a junior synonym of †*Asteracanthus ornatissimus* [154,171].

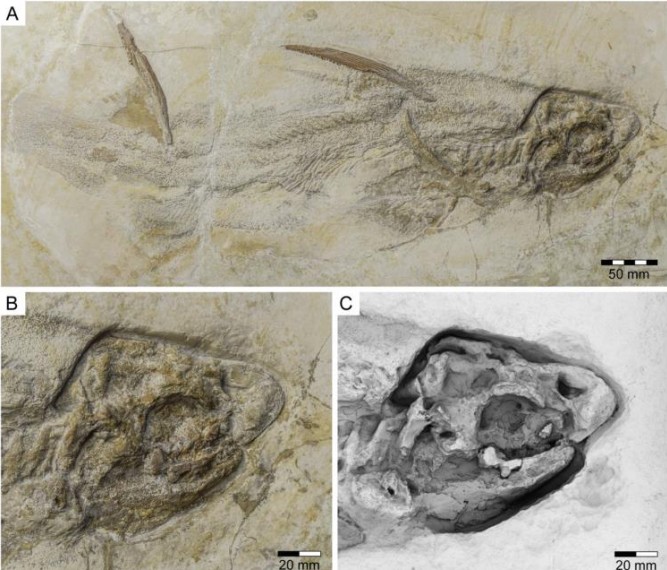

**Figure 10.** Articulated skeleton of †*Hybodus fraasi* Brown, 1900 [172] from the early Tithonian of Solnhofen. (**A**) Complete specimen (SNSB-BSPG 1899 I 2, holotype). (**B**) Close-up view of head. (**C**) Ambient occlusion photogrammetric model of head (courtesy of Christoph Kettler).

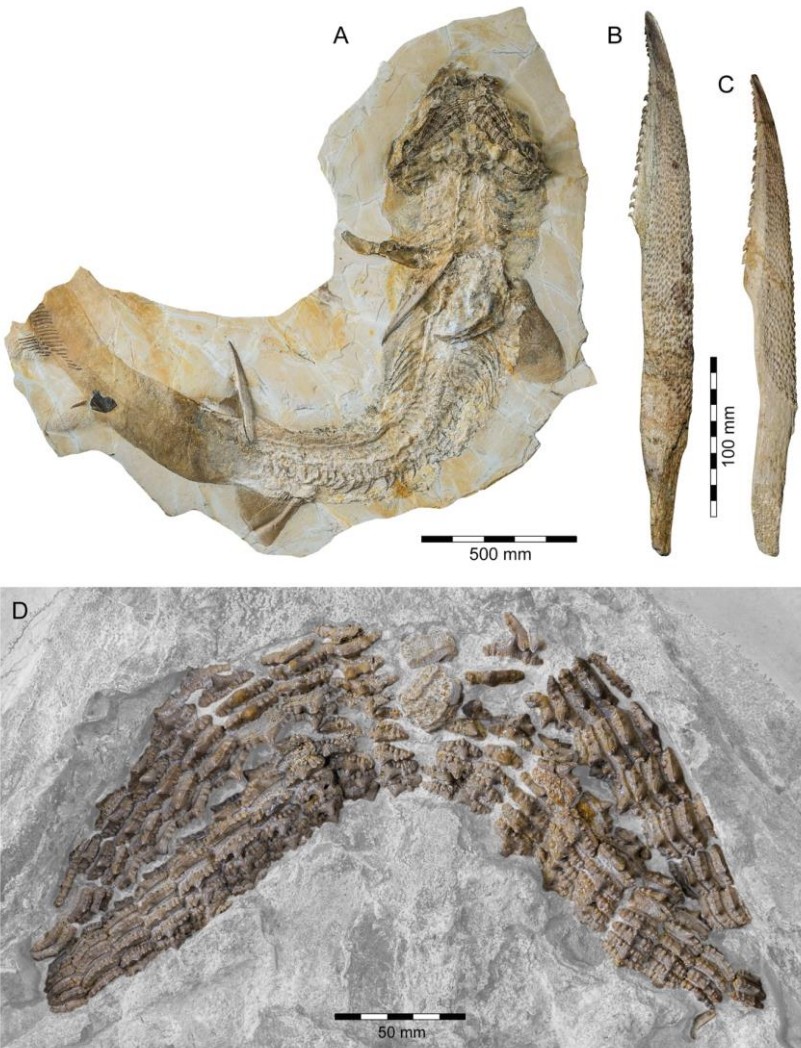

**Figure 11.** Specimen of †*Asteracanthus ornatissimus* Agassiz, 1837 [171] from the lower Tithonian of Solnhofen. (**A**) Complete specimen (PBP-SOL-8003). Close-up view of (**B**) anterior and (**C**) posterior dorsal fin spine. (**D**) Dentition.

†*Asteracanthus* was originally introduced based upon isolated tuberculate dorsal fin spines from the Upper Jurassic of Europe. Later, and the genus †Strophodus Agassiz, 1838 [171], was established on the basis of low-crowned durophagous crushing teeth, and synonymized with †*Asteracanthus* following the discovery of associated dental and skeletal material from the Middle Jurassic of England [178–180]. This long accepted taxonomic scheme has recently been challenged by Stumpf et al. [154], who proposed an amended diagnosis for †*Asteracanthus*, particularly based on the female skeleton from Solnhofen. Key morphological features displayed by this specimen include the presence of tuberculate dorsal fin spines and high-crowned multicuspid teeth that markedly differ from the prominent durophagous crushing teeth that have traditionally been assigned to †*Asteracanthus*. This consequently led Stumpf et al. [60] to resurrect the genus †*Strophodus*, which in fact can be readily distinguished from any other hybodontiform by its characteristic crushing teeth. †*Asteracanthus* is currently considered monotypic including only the type species, †*A. ornatissimus*, whose stratigraphic range, as now understood, is Bathonian to Valanginian (see also [167]).

†*Asteracanthus* has a robust and bulky body with large, well-rounded pectoral fins, suggesting rather sluggish swimming capabilities. Its mouth is subterminal and equipped with an effective grasping-type dentition suitable for processing a wide dietary spectrum

including scavenging behaviours, which also were suggested based on †*Asteracanthus* teeth that were found associated with marine reptile remains [179].

Genus †*Strophodus* Agassiz, 1838 [171]
†*Strophodus* sp.

†*Strophodus* is a speciose genus that thrived the Mesozoic seas for more than 130 million years, from the Middle Triassic to the Early Cretaceous [181–183]. The Solnhofen Archipelago has yielded rare dental and skeletal material attributable to †*Strophodus*, including an articulated but crushed set of jaws with teeth from the lower Tithonian of Mühlheim (Figure 12). According to Pfeil [184], this specimen may be conspecific with †*Strophodus smithwoodwardi* Peyer, 1946 [185] from the Toarcian of Switzerland, pending a more comprehensive comparative study. An additional but yet un-described set of jaws is held in a private collection, suggesting that †*Strophodus* might have been more common than previously thought. In addition, a fragmentary dorsal fin spine previously assigned to †*Asteracanthus ornatissimus* [18,19] might belong to either †*Strophodus* or †*Asteracanthus*.

Unlike in most other hybodontiforms, the palatoquadrates of †*Strophodus* meet at a well-developed median symphysis that extends far posteriorly for about three quarters the maximum length of the jaw elements. This condition, together with the presence of a highly specialized heterodont crushing dentition, indicates some resemblance to modern heterodontiform sharks, which are well-known for exploiting various types of hard-shelled benthic and epibenthic prey [186]. †*Strophodus* has been interpreted to have been a bottom-dwelling taxon that mainly fed on epifaunal hard-shelled invertebrates [24,176], although isotopic data suggest an epipelagic rather than nektobenthic mode of life [187–189].

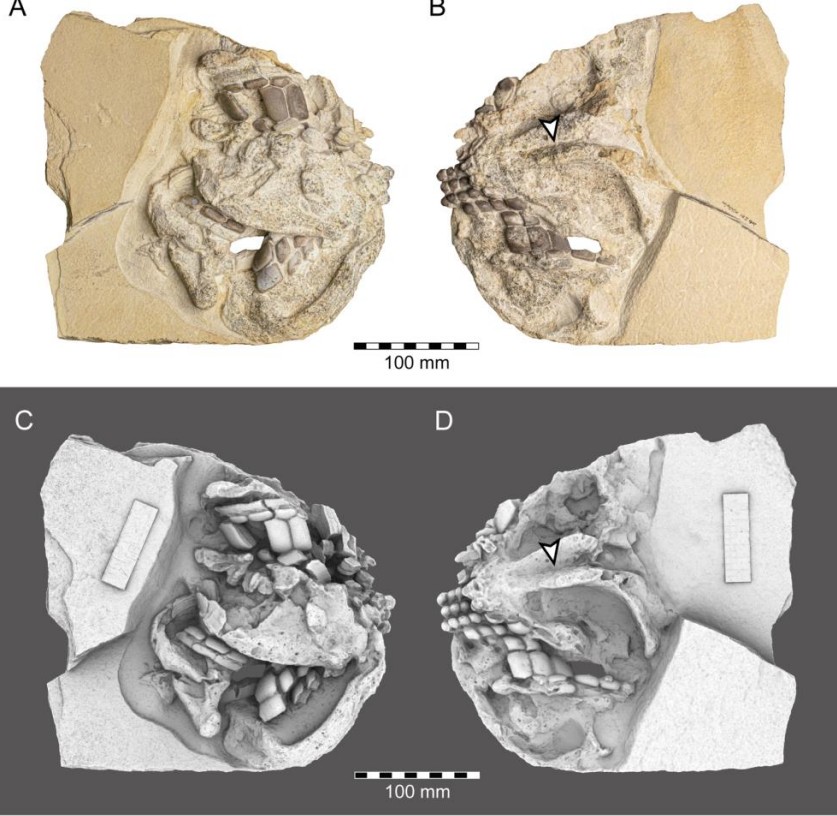

**Figure 12.** Jaws and associated teeth of †*Strophodus* sp. from the early Tithonian of Mörnsheim. (**A**,**B**) Complete specimen (SNSB-BSPG 2010 I 91). (**C**,**D**) ambient occlusion photogrammetric model (arrows point to symphysis connecting palatoquadrates; photogrammetric model courtesy of Christoph Kettler).

4.2.2. Subcohort Neoselachii Sensu Compagno, 1977 [21]

Neoselachians form a highly diversified clade of chondrichthyans encompassing all extant selachimorphs (sharks), batomorphs (rays), and their immediate fossil relatives, e.g., †Synechodontiformes, a member of the stem-group lineage. Considering the sister group relationship with hybodontiforms followed in the present study (see [136–138]), neoselachians would have a maximum age no older than that of hybodontiforms considering both to be sister groups resulting in a fossil record tracing the evolutionary history of neoselachians back to at least 250 Ma [24,136].

Neoselachians are distinguished from other chondrichthyan groups by the segmentation of their notochord by calcified vertebral centra, a fused puboischiadic bar (pelvic girdle), and reduction in the segments between the pelvic metapterygium and the mixopterygial cartilage. The fin spines, when present, have a smooth surface with an enameloid-covered crown and without posterior/posterolateral denticle rows [136,149,155,190]. Fusion or articulation of the right and left halves of the pectoral girdle has been proposed as another synapomorphy of this group recently (e.g., [21,137]). However, all hexanchiforms present a separation of both halves [138,191]). The bell-shaped basicranial carotid-aortic circuit [192] was also proposed as another possible uniting character, but remains untested within a phylogenetic framework.

Currently, two major groups of elasmobranchs are recognized: selachomophs (sharks) and batomorphs (rays; often also called 'Batoidea'). A consensus exists among palaeobiologists and ichthyologists on the placement of batomorphs as the sister group of selachimorphs, which was recovered initially by molecular analysis [133,193]. Using morphological traits within a strict cladistic framework support this relationship (e.g., [138]). However, this hypothesis needs further testing by additional studies as uncertainties regarding the relationships between these groups, but also within these groups, remain, even when molecular data are used (e.g., [194]).

The latest unequivocal neoselachian fossil record comes from the Early Jurassic, followed by rapid diversification episodes during the Late Cretaceous. Several dental remains of possible neoselachian were recovered from older Middle Triassic, (e.g., [195]) or even Permian deposits [196]. However, currently, neoselachian teeth include no unambiguous synapomorphy to differentiate them from other groups, rendering these remains systematically uncertain (see [197]).

4.2.3. †Synechodontiformes

Order †Synechodontiformes Duffin and Ward, 1993 [198]

Synechodontiform sharks were common and widely distributed during the Mesozoic era (e.g., [199–203]). They are among the most diverse chondrichthyan groups recovered from deposits of Nusplingen and in the Solnhofen Archipelago and are represented by three families, the †Palaeospinacidae, †Paraorthacodontidae, and †Orthocodontidae [204]. The phylogenetic intra- and interrelationships of these groups have been discussed for years (e.g., [136,198,204–206]), and in the last decade, they have been recognized as stem-group members of neoselachians [204]. The monophyly of †Synechodontiformes is supported by two synapomorphies: tooth roots displaying a conspicuous pseudopolyaulacorhize vascularization pattern and roots with a labial depression below the crown. Here, we follow the phylogenetic hypothesis of Klug [204], although we acknowledge that it remains to be verified by future research. That is because both characters are assumed to also be present in various euselachians, e.g., hexanchiforms and some hybodontiforms [24,195,207], and thus might have evolved convergently.

Family †Orthacodontidae de Beaumont, 1960 [51]
Genus †*Sphenodus* Agassiz, 1843 [171]

This genus is known from the Solnhofen Archipelago on the basis of incomplete skeletal remains and some isolated teeth. Initially referred to as *Oxyrhina* (Fraas, 1855 [65]) by Quenstedt [106], these remains suggest that †*Sphenodus* was a large shark (total body

length ca. 2–3 m), with a fusiform body indicating an active swimmer and predator. The dentition of †*Sphenodus* is similar to that of today's sand tiger shark with closely spaced, large, and accentuated teeth, which protrude labially from the jaws. The morphology and arrangement of these teeth are characteristic for grasping soft-bodied prey, e.g., squids. †*Sphenodus* seemingly was very abundant and widespread during the Jurassic, with many nominal species that have been described from Europe up to now (e.g., [51,198]). Most of these species, however, are founded on single teeth, some of them even being based upon cusps with the root missing. Two species of †*Sphenodus* have been described from the Upper Jurassic of Southern Germany so far: †*S. macer* (Quenstedt, 1851 [106]), and †*S. nitidus* Wagner, 1861 [107]. De Beaumont [51] and Schweizer [66] considered †*S. nitidus* and †*S. macer* to be synonymous, whereas Woodward [179] placed †*S. macer* into synonymy with †*S. longidens* Agassiz, 1843 [171]. Currently, following Musper [208], both species are considered †*Sphenodus nitidus* and †*Sphenodus macer* valid and are the only two species known from skeletal remains from the Solnhofen Plattenkalks.

†*Sphenodus macer* (Quenstedt, 1851 [106])

The holotype of this species was collected from upper Kimmeridgian deposits of Egesheim on the Heuberg Plateau in the SW Swabian Jura (Baden-Württemberg, SW Germany, which is located close to Nusplingen. This specimen is currently housed in the Staatliches Museum für Naturkunde Stuttgart (SMNS 80142-44). It is preserved in ventral view and displays the characteristic fusiform body with parts of its jaws, branchial skeleton, pectoral and pelvic girdles, paired fins, and caudal fin being preserved (Figure 13A). Kriwet and Klug [18] and Klug [204] assumed that †*Sphenodus* has two subtriangular and well-separated dorsal fins based on the reconstruction provided by Böttcher and Duffin [68]. This seems rather hypothetical as these authors do not mention any specific number of dorsal fins even in their description (see also [19]). Better-preserved specimens from the genus confirm that this shark only had a single, posteriorly placed dorsal fin, similar to the condition seen in †*Paraorthacodus* (see below).

†*Sphenodus macer* is differentiated from other species within the genus on the basis of their tooth morphology. Its teeth are small, with the cusp reaching a height of up to 18 mm. Their cusps are only slightly inclined distally and very faintly distorted. The cutting edges of the cusp are symmetrical (located on both sides), and the root is narrow. This morphology suggests a clutching-type dentition, which contrasts with the tearing-type dentition displayed by †*S. nitidus* (see [68] for an extended description).

†*Sphenodus nitidus* Wagner, 1862 [105]

The holotype of this species was collected from early Tithonian deposits of the Solnhofen Archipelago in Southern Germany. The highly disarticulated and incomplete specimen does not allow reconstructing unambiguously its body shape (Figure 13C). However, additional specimens of this species recently have been collected, one of which is housed in the Staatliches Museum für Naturkunde Stuttgart (SMNS) and which currently is under study. This specimen is very well preserved and has revealed several key morphological features of this species clarifying some issues related to the taxonomic and phylogenetic relations of this genus (Figure 13D). Accordingly, this species is, inter alia, differentiated from †*S. macer* by its tooth morphology with teeth measuring up to 22.4 mm in height with a high central cusp, which is upright in labial view and has two well-developed mesial and distal cutting edges, a tooth height to length ratio of ca. 2.4, a narrow root (see [68]). The dentition displays a weak dignathic heterodonty and is of the tearing-type according to Cappetta [24].

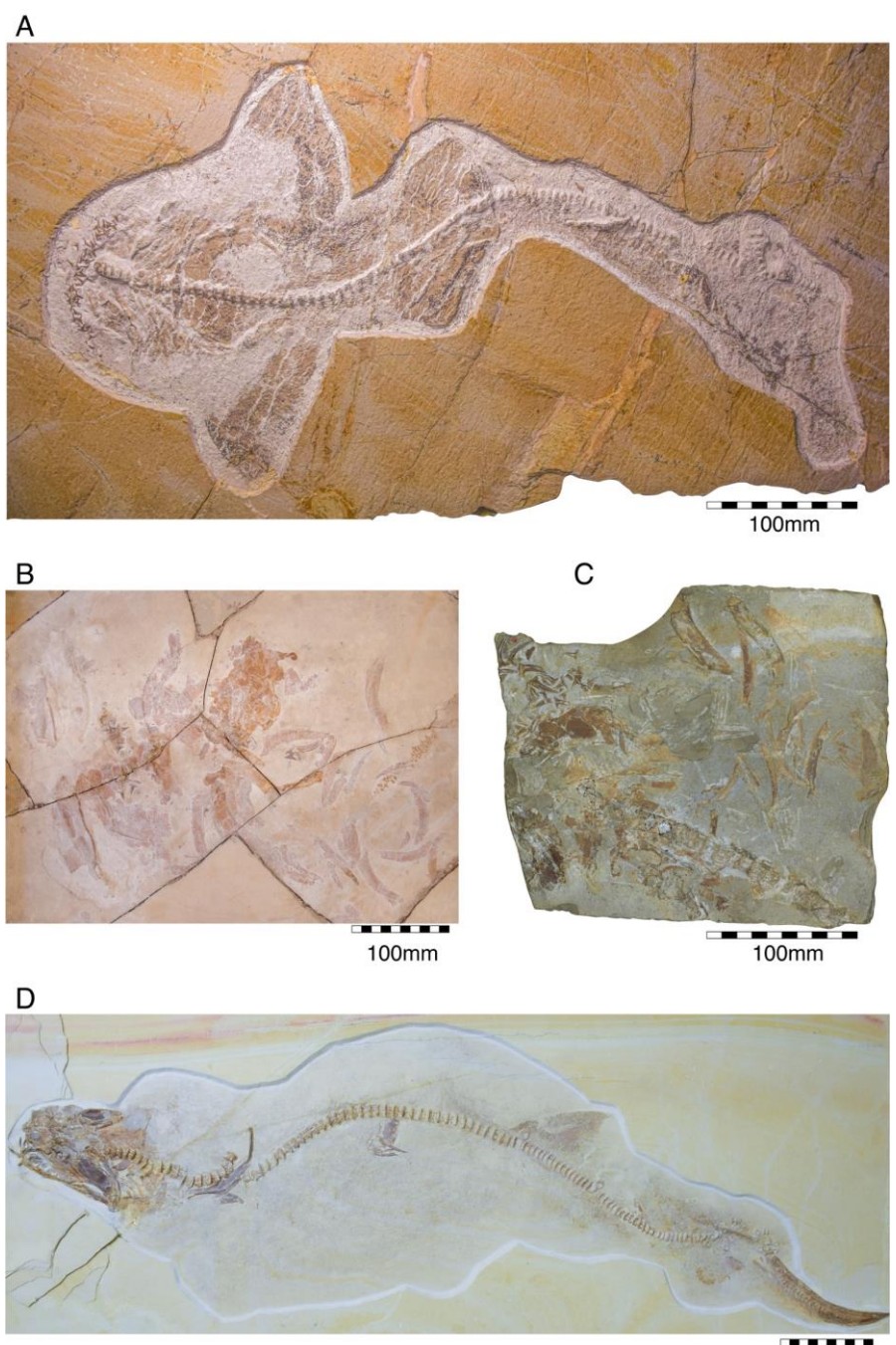

**Figure 13.** Specimens of †*Sphenodus* Agassiz, 1843 [171] found in the Solnhofen Archipelago. (**A**) Holotype specimen of †*S. macer* (Quenstedt, 1851 [106]) (SMNS 80142-44) from the upper Kimmeridgian of Nusplingen. (**B**) Fragmentary specimen of †*S. macer* (MCZ 13389) from the Upper Jurassic of Solnhofen. (**C**) Holotype specimen of †*S. nitidus* Wagner, 1862 [105] (BSGP-AS VII 647) from the Tithonian of Solnhofen. (**D**) †*S. nitidus* (SMNS 96844-7) from the Kimmeridgian of Nusplingen.

Family †Palaeospinacidae Regan, 1906 [140]
Genus †*Synechodus* Woodward, 1888 [178]
†*Synechodus ungeri* Klug, 2009 [70]

†*Synechodus* is one of the species-richest genera of synechodontiforms, currently including ca. 16 nominal species with a stratigraphic range from the Late Triassic to the Palaeocene [24,209]. Initially, skeletal remains of †*Synechodus* only were known from the Upper Cretaceous of England. A detailed revision of potential synechodontiform

sharks by Klug [70] allowed identifying additional new, previously unrecognised and undescribed skeletal remains from the Late Jurassic, which, e.g., include †*Synechodus ungeri* from Nusplingen.

†*Synechodus ungeri* description was originally based on an incomplete disarticulated skeleton from the upper Kimmeridgian of Nusplingen, which is housed in the Staatliches Museum fur Naturkunde Stuttgart under number SMNS 85975/1 (Figure 14A). The disarticulated and incomplete specimen, unfortunately, does not allow deducing detailed conclusions about its morphology. However, isolated teeth collected from the lower Kimmeridgian of Mahlstetten and an additional fairly complete skeleton from the lower Tithonian of the Solnhofen Archipelago (JME SOS 3152) which was recovered from the Plattenkalks near the village of Schernfeld (commune of Birkhof) show that this species was a rather short shark with a large and bulky head with a broadly rounded snout and large pectoral fins (JME SOS 3152) [19,25] (Figure 14B). The multicuspidate teeth indicate a clutching-type dentition.

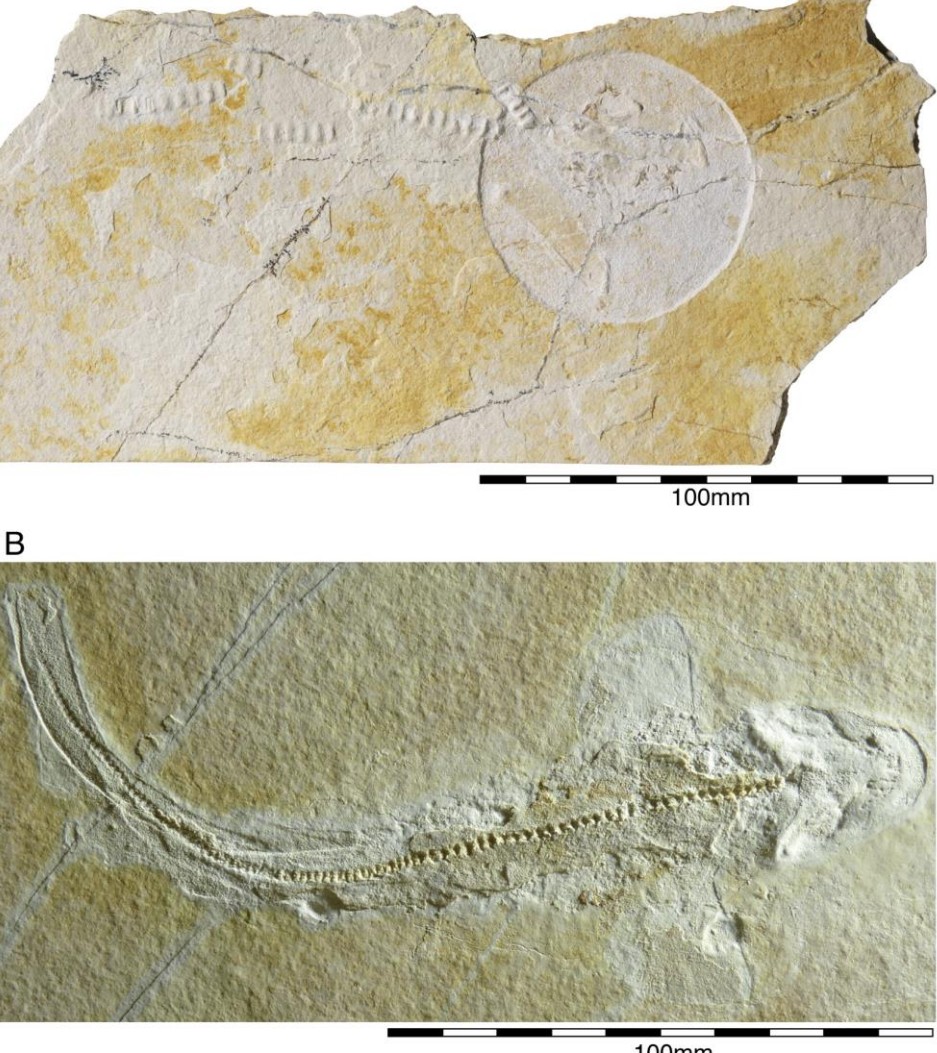

**Figure 14.** †*Synechodus ungeri* Klug, 2009 [70]. (**A**) Holotype specimen (SMNS 85975-1) from the Kimmeridgian of Nusplingen. (**B**) Holomorphic specimen (JME SOS 3152 B) from the Tithonian of Solnhofen.

The presence of two dorsal fins in †*Synechodus ungeri*, as suggested by Kriwet and Klug [19], can neither be confirmed nor disproved due to the limited availability of suitable material. The only available specimen with two dorsal fins, collected from the lower Titho-

nian of Solnhofen, figured and referred to †*Synechodus* sp. by Kriwet and Klug [18] (figure 6b), Klug and Kriwet [210] (figure 3b), and Klug [204] (figure 1d), cannot unequivocally be referred to as †*Synechodus* or any other synechodontiform due to the lack of preserved teeth, as already stated by Klug [56].

Kriwet and Klug [19] assumed that all currently available specimens of †*S. ungeri* represent juveniles, which, however, needs to be established by analysing all skeletal remains in detail.

Genus †*Paraorthacodus* Glikman, 1957 [199]
†*Paraorthacodus jurensis* (Schweizer, 1964 [66])

†*Paraorthacodus jurensis* seemingly is the only species of this genus occurring in the Southern German Plattenkalk deposits. Originally, this species was described from the lower Kimmeridgian lagoonal deposits of Nusplingen. The holotype is incomplete and preserved in ventral view displaying details of its visceral skeleton including the jaws, its anterior vertebral column and its left pectoral girdle without the pectoral fin being preserved (Figure 15A). Recently a more complete specimen was recovered from Nusplingen (SMNS 88987/1) [56] (Figure 15B) and larger specimens from, e.g., the lower Tithonian of Haunsfeld (collection Helmut Leich, Bochum) and the lower Tithonian of Blumenberg near Eichstätt (SNSB-BSPG 1964 XXIII 157; Figure 15C) [19,210] along with several smaller (SNSB-BSPG 1894 X 5 (Schernfeld), SNSB-BSPG 1996 I 31 (Solnhofen); Figure 15D). Specimen SNSB-BSPG 1964 XXIII 157 previously was assumed to represent the hexanchiform †*Notidanoides muensteri* Agassiz, 1843 [171] but subsequently was identified by Kriwet and Klug [19] as a large specimen of †*Paraorthacodus* that still awaits its formal description. Consequently, we assign all †*Paraorthacodus* specimens to the species †*P. jurensis* here.

The two known specimens of †*P. jurensis* from Nusplingen represent adult individuals, while juveniles and adults co-occurred in the Solnhofen Archipelago with no evident size segregation pattern. For instance, the localities of Schernfeld (occurrence of a juvenile specimen) and Blumenberg (occurrence of an adult specimen) are very close and were part of the same depositional area.

Schweizer [66], Duffin [211], and Klug et al. [71] provided detailed accounts of the dental and skeletal morphology of †*P. jurensis*. Based on all available information it is clear that juveniles of this species were rather bulky with a broadly rounded head anteriorly, while adults reached rather large sizes with a fusiform body. The pelvic and anal fins are closely arranged, and the caudal fin is very elongated in both juveniles and adults. Strikingly, a single, far posteriorly placed dorsal fin characterizes this shark (Figure 15D), which is similar to the condition seen in hexanchiform sharks.

The dentition of †*P. jurensis* is of the clutching-type and displays a distinct sexual dimorphism in the morphology of the teeth [71]. Accordingly, the teeth of male specimens always display one pair of lateral cusplets accompanying the main cusp more than females.

The family †Paraorthacodontidae additionally is represented by another shark species in the Solnhofen Archipelago, †*Macrourogaleus hassei*. The holotype of this species is housed in the Bayerische Staatssammlung für Paläontologie und Geologie in Munich (Figure 16A). This species also is known from several very small specimens not reaching more than 16 cm total body length (Figure 16C–E). Hasse [212] originally illustrated this small shark under the name of *Pristiurus* (Bonaparte, 1832 [86]) and Woodward [179] later introduced the species †*P. hassei* (Woodward, 1889 [178]) for these specimens based on a specimen from the lower Tithonian of the Eichstätt area. However, the name *Pristiurus* is a junior synonym of *Galeus* Rafinesque, 1810 [213], which is a deep-water catshark belonging to the Carcharhiniformes. The only character uniting †*Macrourogaleus* and the extant *Galeus* is the presence of a single row of enlarged, almost thorn-like placoid scales on the caudal fin crest (Figure 16B). Otherwise, the Late Jurassic shark differs considerably in its general body shape and some morphological details. Fowler [214] subsequently introduced the genus †*Macrourogaleus* for this fossil shark separating it from *Galeus*. Characteristic features for †*Macrourogaleus* include clearly rounded pectoral and pelvic fins, a very elongated caudal fin, and the presence of a single dorsal fin similar to conditions seen in †*Paraorthacodus*. The

teeth are very small and fragile, which makes a detailed examination very difficult. This also resulted in the long-lasting assumption that its dentition was very similar to †*Palaeoscyllium* Wagner, 1857 [102]. However, a detailed examination revealed that †*Macrourogaleus* is a member of the Paraorthcodontidae because the teeth are mutlicuspidate and have a typical synechodontiform root with a pseudopolyaulacorhize vascularization pattern [56]. Furthermore, †*Macrourogaleus* is easily distinguishable from †*Paraorthacodus* by its slenderer body, the more rounded paired fins, the very elongated anal fin, and the characteristic row of enlarged, thorn-like placoid scales on the dorsal caudal fin crest (Figure 16B). This also indicates that †*Macrourogaleus* most likely does not represent a juvenile of another shark from the Solnhofen Archipelago since larger sharks with a single dorsal fin and combined with an elongated anal fin remain unknown (Figure 16).

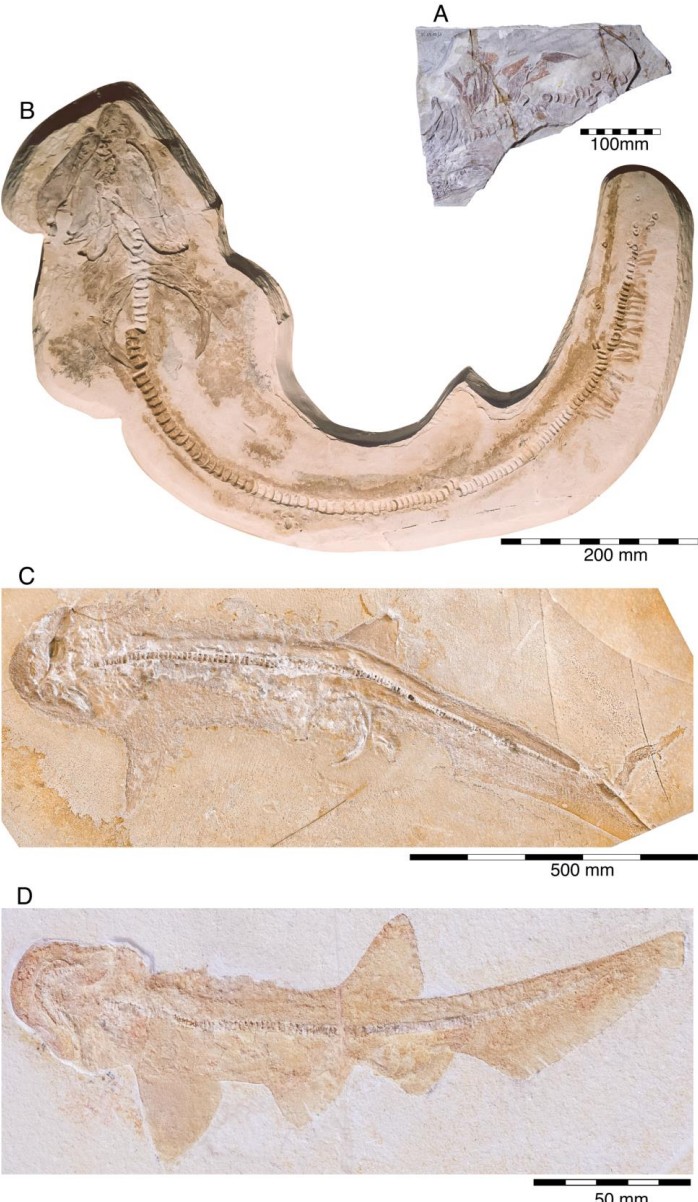

**Figure 15.** †*Paraorthacodus jurensis* (Schweizer, 1964 [66]). (**A**) Holotype specimen of †*P. jurensis* (GPIT 1210/1,) found in the Tithonian of Nusplingen. (**B**) Holomorphic specimen (SMNS 88987-1) found in the Tithonian of Nusplingen. (**C**) Holomorphic specimen (SNSB-BSPG 1964 XXIII 157) from the lower Tithonian of Eichstätt. (**D**) Holomorphic specimen (SNSB-BSPG 1996 I 31) from the lower Tithonian of Eichstätt.

Genus †*Macrourogaleous* Fowler, 1947 [212]
†*Macrourogaleus hassei* (Woodward, 1889 [179])

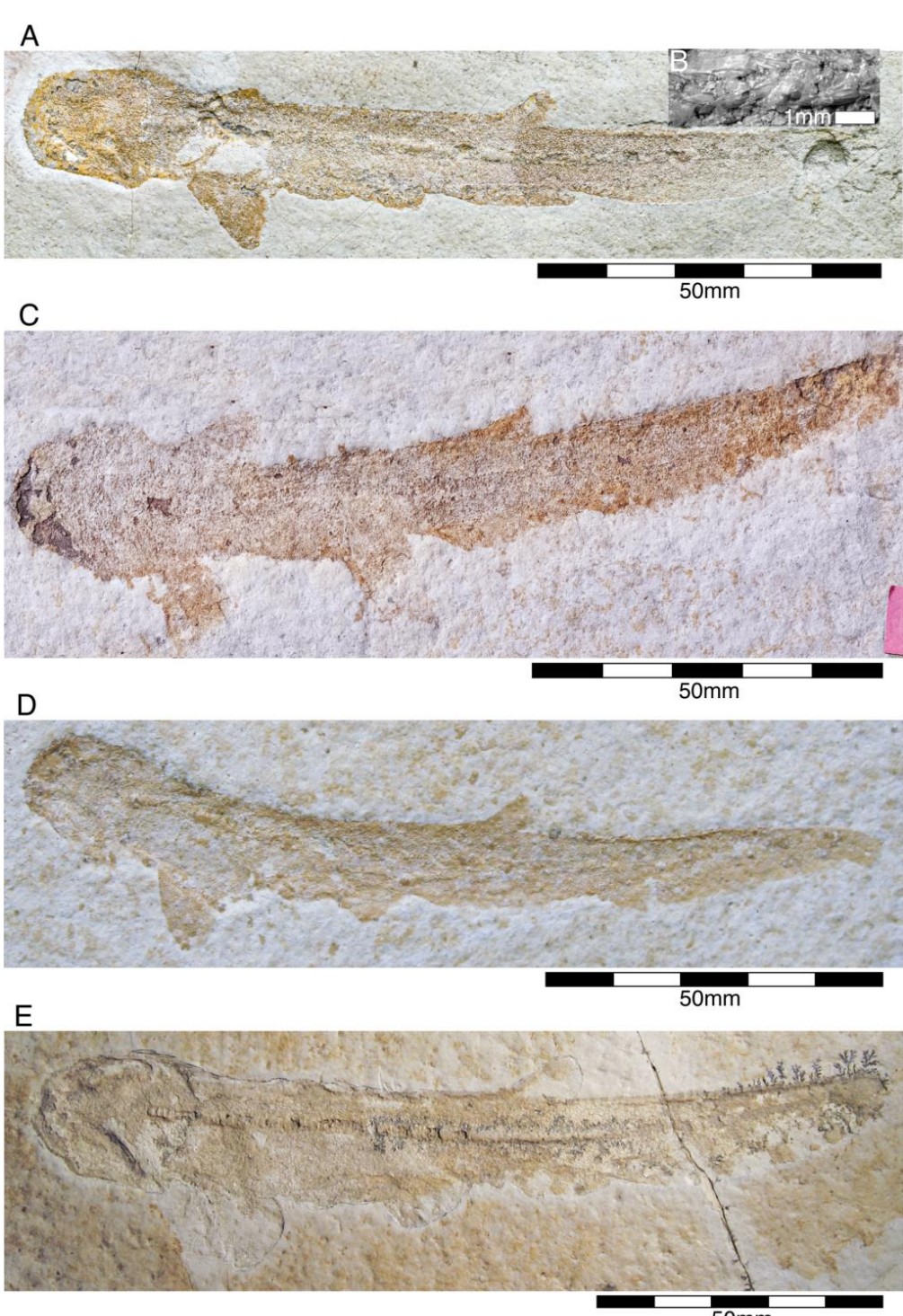

**Figure 16.** †*Macrourogaleus hassei* Fowler, 1947 [211] (**A**) Holotype specimen (SNSB-BSPG AS I 1363) from the lower Tithonian of Eichstätt. (**B**) Close up to the enlarged denticles in the upper lobe of caudal fin. (**C**) Holomorphic specimen (SNSB-BSPG AS I 1362) from the lower Tithonian of Eichstätt. (**D**) Holomorphic specimen (JME SOS 2209) from the lower Tithonian of Eichstätt. (**E**) Holomorphic specimen (AMNH 7498) from the lower Tithonian of Eichstätt.

The body and fin shapes indicate that this small shark was probably strictly benthic, predominantly inhabiting the very structured sponge and microbial reefs surrounding the Plattenkalk basins, where it was hunting in crevices for small soft-bodied prey. The comparably high number of recovered specimens might be related to the fact that these small sharks were either quite abundant or their carcases easily washed into the Solnhofen basins.

4.2.4. Heterodontiformes

Superorder Galeomorphii Compagno, 1973 [20]
Order Heterodontiformes Berg, 1937 [215]

Extant bullhead sharks of the order Heterodontiformes are a small group comprising eight species within a single genus (*Heterodontus* de Blainville, 1816 [216]) that inhabit relatively temperate waters. The oldest fossil record of bullhead sharks comes from the Toarcian (180 Ma) and its Early Jurassic fossil record consists of exclusively of isolated teeth. During the Late Jurassic, only three genera are recognized: *Heterodontus* (†*H. sarstedtensis* Thies, 1892 [217], †*H. semirugosus* Plieninger, 1847 [218] and †*H. zitteli* Eastman, 1911 [219]), †*Paracestracion* (†*P. falcifer* Wagner, 1857 [102], †*P. bellis* Underwood and Ward, 2004 [220], †*P. danieli* Slater, 2016 [221]), and †*Proheterodontus* (†*P. sylvestris* Underwood and Ward, 2004 [220]). Currently, we assume that both †*Paracestracion* and *Heterodontus* to be the only heterodontids co-occurring in the Solnhofen Archipelago, while they are absent from the Nusplingen lagoonal deposits.

Morphological characters for heterodontiforms include a monognathic heterodont dentition comprising molariform lateral and posterior teeth with broad and low cusps, which are pavement-like arranged and smaller anterior teeth with high cusps and lateral cusplets in juveniles, lack of rostral cartilages, prominent supraorbital crests, and antero-posteriorly elongated nasal capsules. The articulation between neurocranium and palatoquadrate is provided by a broad articulation surface situated in the downturn of the basal portion of the neurocranium between the ventral surface of the nasal capsules and part of the suborbital shelf. There are also two strong, unornamented, and enameloid-covered dorsal fin spines supporting the two dorsal fins [18–21,76,102,138–141,215–221].

Family Heterodontidae Gray, 1851 [222]
Genus †*Paracestracion* Koken in Zittel, 1911 [223]

†*Paracestracion* differs from *Heterodontus* most significantly in the position of the pelvic girdle, which is located almost at the level of the first dorsal fin spine, and by its antero-posteriorly shorter and almost rectangular neurocranium. The placement of the pelvic and first dorsal fins suggests a slow swimming, epibenthic lifestyle [224]. Another important morphological feature of †*Paracestracion* is the presence of a root shelf on its teeth and the lack of development of molar teeth in juvenile specimens [59] and the lack of molariform teeth in pre-Kimmeridgian species [220].

†*Paracestracion falcifer* Wagner, 1857 [100]

The type species, †*Paracestracion falcifer*, comes from the early Tithonian (exact locality is unknown) and is represented an almost complete, slightly disarticulated skeleton, which is housed in the Bayerische Staatssammlung für Paläontologie und Geologie, Munich. The specimen is preserved in dorsal view, exposing most of the cranium, the paired pectoral and pelvic as well as the unpaired dorsal fins with their corresponding and the unpaired anal and caudal fins (Figure 17A). This species differentiate from other species within the genus by the labial ornamentation on its anterior teeth and the absence of a distal curvature in parasymphyseal teeth, the position of the first spine over the 23rd–24th vertebral centrum and of the second dorsal fin spine above the 43rd–44th vertebral centrum, position of the pectoral girdle above the tenth vertebral centrum and that of the pelvic girdle over the 24th vertebral centrum (see also [221]) (Figure 17A–C).

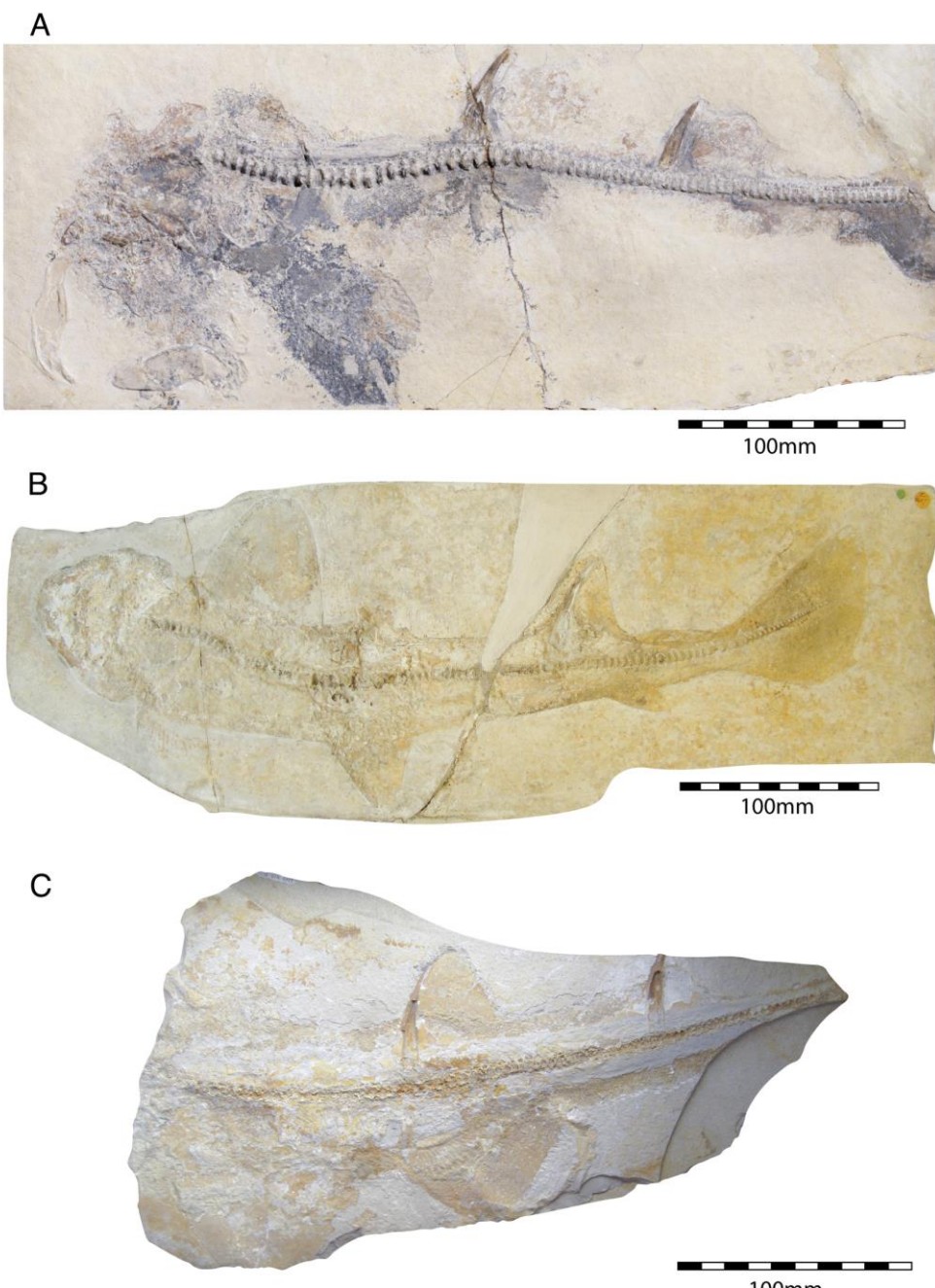

**Figure 17.** †*Paracestracion falcifer* Wagner, 1857 [100]. (**A**) Holotype specimen (SNSB-BSPGM AS VI 505) from the lower Tithonian of Solnhofen. (**B**) Holomorphic specimen (NHMUK P 8657) from the Kimmeridgian of Eichstätt. (**C**) Almost complete specimen (JME SOS 2215) from the Kimmeridgian of Eichstätt.

†*Paracestracion viohli* Kriwet, 2008 [57]

   †*Paracestracion viohli* is represented by a very incomplete species from the upper Kimmeridgian of Schamhaupten, that lacks most of the postcranial skeleton posterior to the pelvic girdle was presented by Kriwet [57]. The type and only specimen known so far is housed in the Jura Museum Eichstätt under collection number JME Scha 728. It is preserved in ventral view (Figure 18). The preserved body portions are densely covered by placoid scales outlining the anterior body portion including most of the head, parts of the jaws with the dentition, the girdles, and the pectoral and pelvic fins. The scales do not differ in their general morphology from those described for other Late Jurassic heterodontids. However, it differs

from †*P. falcifer* in the supposed absence of molariform teeth in adult specimens (see [57] for a detailed description). Surprisingly, all teeth display lateral cusplets, a feature that is supposed to be indicative of juveniles. The overall size compared to the adult specimen of †*P. falcifer*, however, would suggest an adult individual. Additionally, both species occupy different stratigraphic levels in the Solnhofen Archipelago with †*P. viohli* occurring in the late Kimmeridgian and †*P. falcifer* seemingly being restricted to the early Tithonian.

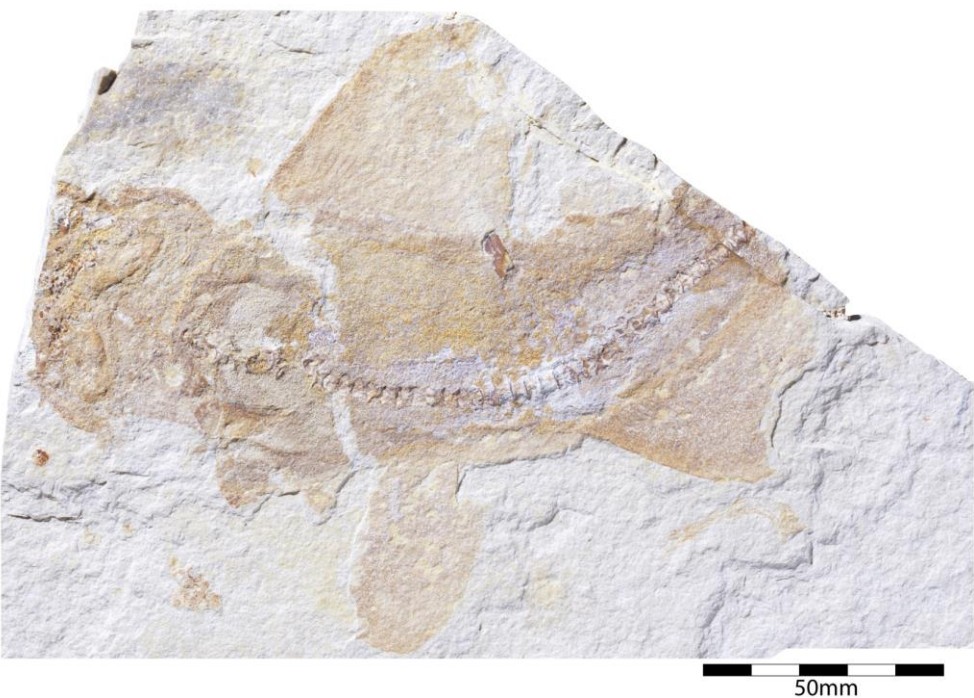

50mm

**Figure 18.** Holotype specimen of †*Paracestracion viohli* Kriwet, 2008 [57] (JME Scha 728) from the upper Kimmeridgian of Schamhaupten.

†*Paracestracion danieli* Slater, 2016 [221]

Recently, Slater [221] identified a third species, †*Paracestracion danieli*, based on a single holomorphic specimen consisting of part and counterpart from the lower Tithonian of Eichstätt while conducting an undergraduate study at the University of Vienna. The type is housed in the Wyoming Dinosaur Center, USA. This specimen is preserved in a dorso-lateral position with its head being bent, displaying the palatoquadrate and Meckel's cartilage in ventral view, the two dorsal fins and their corresponding spines in lateral view, as well as one of the pectoral and pelvic fins, and the anal and caudal fins (Figure 19). This species is characterized by the presence of seven cusps and cusplets on the anterior teeth, the first dorsal fin spine being located between the 32nd–33rd vertebral centra, and the second dorsal fin spine being located between the 62nd–63rd vertebral centrum. The pectoral girdle is located above the 12th vertebral centrum and the pelvic girdle above the 32nd vertebral centrum (see [221] (table 1)).

Currently, the taxonomic status of †*P. viohli* and †*P. danieli* is ambiguous and detailed revisions of all available specimens of the three species is necessary.

Genus *Heterodontus* de Blainville, 1816 [216]
†*Heterodontus zitteli* Eastman, 1911 [219]

This species was originally based on a specimen recovered from the early Tithonian deposits in Eichstätt. The plate of the holotype is housed in the Carnegie Museum of Natural History under collection number CM 4423, while the counterpart of the holotype is deposited in the Natural History Museum of London. The specimen is preserved in dorsal view, displaying features of the skull roof, pectoral and pelvic fins, both dorsal fin

spines as well as anal and caudal fins (Figure 20A). Maisey [225] assigned this species later to *Heterodontus*. However, Hovestadt [226] considers this species to be a *nomen nudum*, because the specimen lacks the dentition completely. In a more recent study, Slater et al. [59] nevertheless identified this species as belonging to *Heterodontus* based on the position of the pelvic girdles between the dorsal fins (Figure 20B–D). Pending further revisions of all Late Jurassic heterodontiforms based on skeletal material might provide a better understanding of their taxonomic composition. There could be several additional specimens of this species currently identified as †*Paracestracion falcifer* (e.g., SNSB-BSPG AS VI 505, SNSB-BSPG 1885 I 12a, b, JME SOS 3153, 3577) based on the relationship between first dorsal fin and pelvic girdle [59].

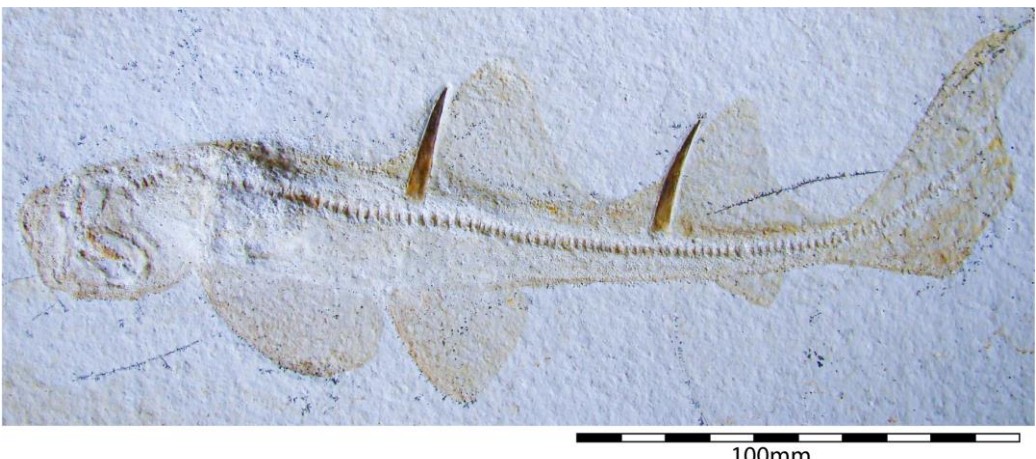

**Figure 19.** Holotype specimen of †*Paracestracion danieli* Slater, 2016 [221] (JPBP-SOL-0005) from the Kimmeridgian of Eichstätt.

4.2.5. Orectolobiformes

Order Orectolobiformes Applegate, 1974 [227]

This order comprises seven extant families with 13 genera and 45 species [228]. Orectolobiforms generally are relatively rare in the Late Jurassic of Europe. Currently, only two orectolobiforms, †*Phorcynis catulina* Thiollière, 1852 [229] and †*Palaeorectolobus agomphius* Kriwet, 2008 [58] are known by skeletal remains and holomorphic specimens from the Upper Jurassic Plattenkalk Lagerstätten of Southern Germany. Cappetta [230] also assigned the carcharhiniforms †*Corysodon* de Saint-Seine, 1949 [80] and †*Palaeoscyllium* Wagner, 1857 [102] erroneously to the Orectolobiformes (see also below).

In Late Jurassic chondrichthyan faunas that are represented by isolated teeth only, such as Mahlstetten, orectolobiforms, conversely, are more diverse including known taxa (e.g., †*Palaeobrachaelurus* Thies, 1982 [217]), but also several still undescribed taxa. The systematic position and interrelationships of the Plattenkalk orectolobiforms still are unresolved pending further analyses.

*Incertae familiae*
Genus †*Phorcynis* Thiollière, 1852 [229]
†*Phorcynis catulina* Thiollière, 1852 [229]

The orectolobiform †*Phorcynis catulina* originally was described by Thiollière [229] (381, pl. 8, figure 2) from late Kimmeridgian Plattenkalks near of Le Bugey, close to Cerin in the Department of Ain (France). The holotype, which originally was part of the palaeoichthyological collection of the Muséum d'histoire naturelle de Lyon (1772–1978), is housed today in the Musée des Confluences in Lyon under the collection number MHNL 15 293.

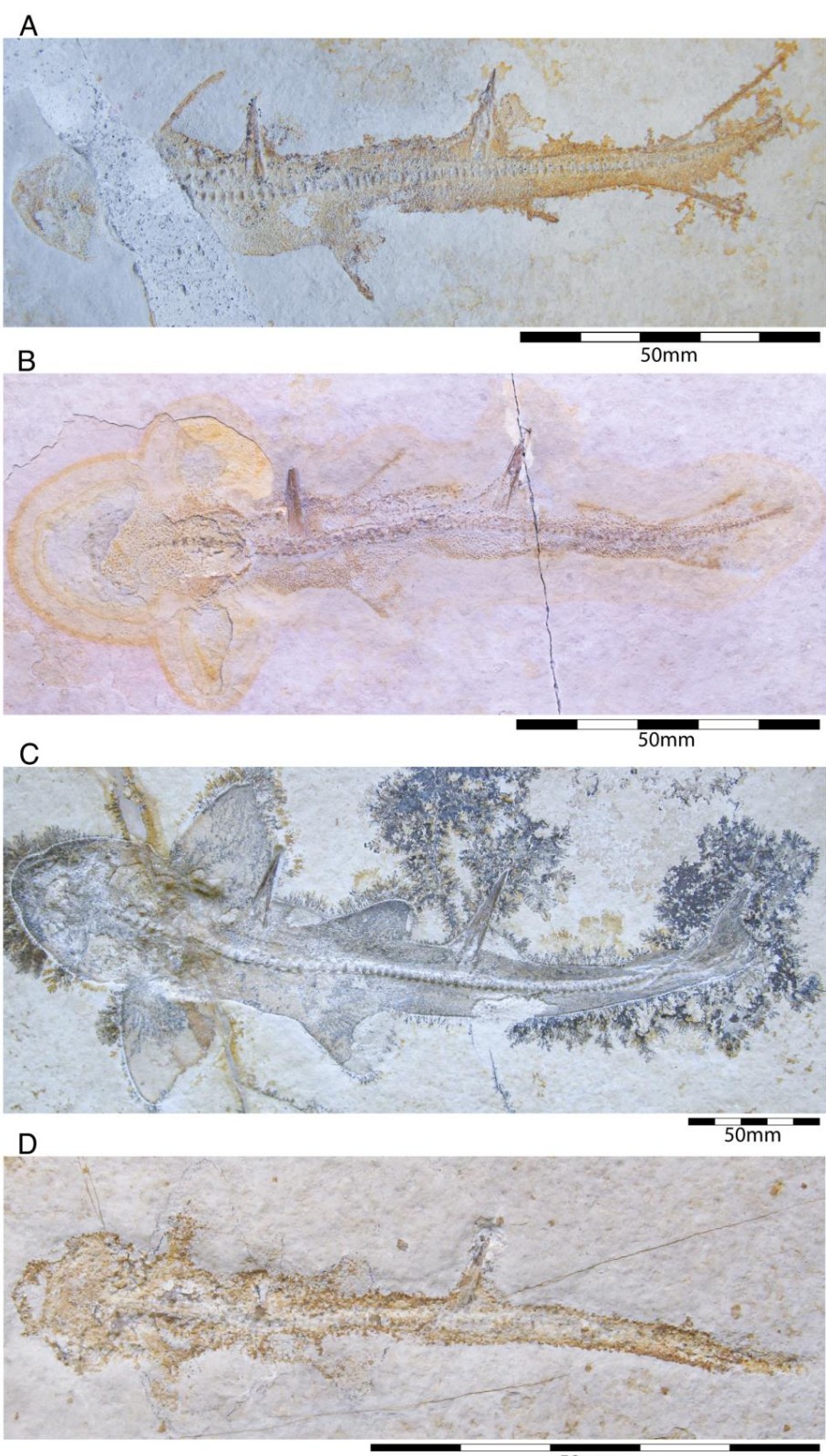

**Figure 20.** †*Heterodontus zitteli* Eastman, 1911 [219]. (**A**) Holotype specimen (NHMUK P 6938) from the Kimmeridgian of Eichstätt. (**B**) Holomorphic specimen (SNSB-BSPG 1885 I 12a) from the Tithonian of Eichstätt. (**C**) Holomorphic specimen (JME SOS 3577) from the lower Tithonian between Wintershof and Workerszell. (**D**) Holomorphic specimen (JME SOS 3153) from the lower Tithonian from Blumenberg.

†*Palaeoscyllium minus* (Woodward 1889) [179] was introduced from a small holomorphic shark specimen from the Eichstätt area, which was acquired by exchange from the German mineral and fossil dealer Bernard Stürtz (1845–1928) in June 1888 and which is housed in the Natural History Museum London under catalogue number P.5541. The label indicates that the specimen comes from the Kimmeridgian; however, it is more likely that it derived from the early Tithonian of Solnhofen. White [231] noticed considerable morphological differences of this specimen to the type species of †*Palaeoscyllium*, †*P. formosum* Wagner, 1857 [100], such as pelvic fins that are positioned more anteriorly in relation to the dorsal fin and a smaller anal fin that is more closely positioned to the caudal fin.

Later a new taxon †*Crossorhinus jurassicus* (Woodward, 1918 [232]) from the Plattenkalks of Solnhofen was introduced, the specimen was purchased from the German 'Rheinisches Mineralien-Kontor Dr. F. Krantz' in October 1913 and is housed in the Natural History Museum London under collection number NHMUK P 11211. Most likely, this specimen also was recovered from the early Tithonian of Solnhofen. Fowler [212] transferred †*C. jurassicus* to the new genus †*Palaeocrossorhinus* (Fowler, 1947 [212]). The single specimen of this taxon, however, is almost impossible to separate from †*Palaeoscyllium minus* and the slight differences are related to the preservation of the two specimens. Cappetta [230] was the first to note that neither the general morphology nor the dentition separates these species from †*Phorcynis catulina*.

The overall morphology of †*Phorcynis catulina* (e.g., NHMUK P 5541, SNSB-BSPG 1990 XVIII 51, JME SOS 3150; Figure 21A–D) resembles that of Orectolobiformes in general with a broad head and a short snout (wide orbital cavity and a short otic region and a reduced rostrum), trumpet-shaped nasal capsules (Figure 21C), and large and rounded pectoral and pelvic fins (Figure 21). As in modern orectolobiforms, both dorsal fins lack fin spines and are placed well posterior on the body. The pelvic fins lack an apical angle and are located in front of the first dorsal fin. The caudal fin is elongated with a subterminal notch. The dorsal fins are similar in size, triangular and the second dorsal fin is closely situated to the caudal fin. The anal fin is small and rounded. The heterocercal caudal fin presents a well-differentiated apical lobe with a subterminal notch, but no lower lobe. The anal fin, which is not observable in the type specimen from Cerin, but is present in other specimens (see [233]) is small and rounded similar to those of extant parascylliids and located very near the caudal. The teeth are typical orectolobiform with a broad labial apron that overhangs the hemiaulacorhize root [24].

*Incertae familiae*
Genus †*Palaeorectolobus* Kriwet, 2008 [58]
†*Palaeorectolobus agomphius* Kriwet, 2008 [58]

The only, very incomplete specimen of this orectolobiform with uncertain affinities comes from the lower Tithonian Plattenkalks of Kelheim in the Solnhofen Archipelago. The specimen, which is housed in the Museum für Naturkunde, Berlin and consists of only parts of the cephalic region and a partially preserved pectoral fin (Kriwet [58]; see also Kriwet and Klug [18] (as †*Corysodon* de Saint-Seine, 1949 [80]); Kriwet and Klug [19]).

†*Palaeorectolobus agomphius* seemingly had a dorso-ventrally flattened body, with several heavily branched dermal lobes along the lateral margin of the head anterior to the mouth gape and between the mouth and branchial chamber. The pectoral fins seemingly were large with three basal elements, of which the propterygium is the smallest (Figure 22). The teeth lack the labial apron, which is characteristic for orectolobiforms but displays a hemiaulacorhize root supporting its inclusion in the Orectolobiformes, as do the dermal cephalic lobes (see [58] for complete description).

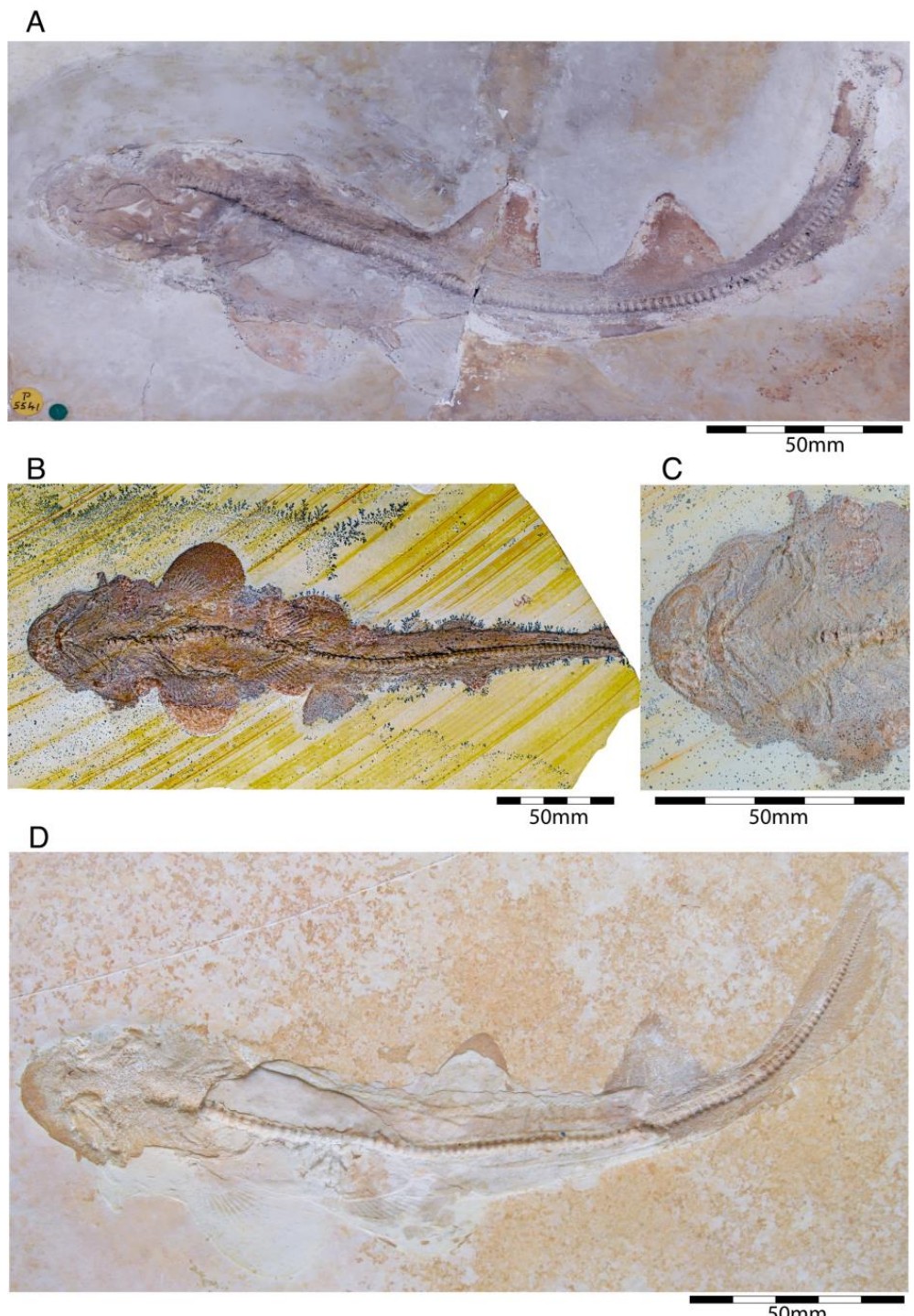

**Figure 21.** †*Phorcynis catulina* Thiollière, 1852 [229]. (**A**) Holotype specimen (NHMUK P 5541) from the lower Tithonian of Eichstätt. (**B**) Almost complete specimen (SNSB-BSPG 1990 XVIII 51) from the lower Tithonian of Zandt. (**C**) Close up to the cephalic region. (**D**) Holomorphic specimen (JME SOS 3150) from the lower Tithonian of Blumenberg.

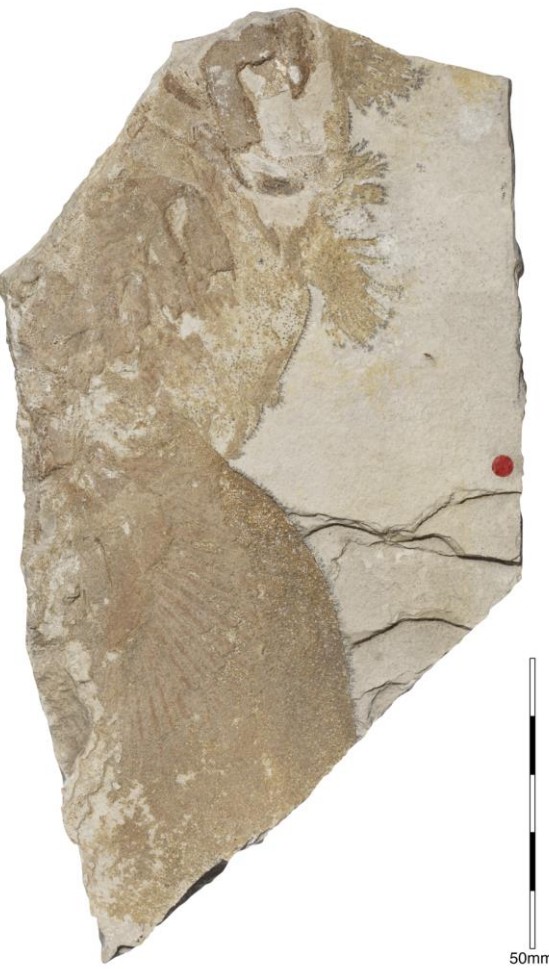

**Figure 22.** Holotype of the species †*Palaeorectolobus agomphius* Kriwet, 2008 [58] (MB.f. 11191) from the lower Tithonian of Kelheim.

4.2.6. Carcharhiniformes

Order Carcharhiniformes Compagno, 1977 [21]

Today, this group represents the most speciose and widespread clade of sharks, including at least 291 extant species in 10 families and 54 genera [186,233]. The earliest records of this group are from the Middle Jurassic (Bathonian), from which several genera have been described based on isolated teeth [24,219]. During the Cretaceous, the abundance and diversity of carcharhiniform sharks increased [234], but the highest diversification rates of carcharhiniforms occurred in the Cenozoic, especially in the last 30 Ma [235].

Two carcharhiniform taxa, †*Bavariscyllium tischlingeri* Thies, 2005 [55], and †*Palaeoscyllium formosum* Wagner, 1857 [102] *non* Marck, 1863 [236], have been recognized in the Upper Jurassic Solnhofen Archipelago and the Nusplingen lagoon up to now. Both were assigned to Scyliorhinidae (catsharks), but no sophisticated phylogenetic analyses have been conducted so far to support such an assignment. The phylogenetic relationships of these carcharhiniform taxa therefore remain controversial as several studies present contrasting hypotheses for living scyliorhinids. Vélez-Zuazo and Agnarsson [237] and Naylor et al. [193], e.g., proposed scyliorhinids to be paraphyletic based on molecular data, splitting them into three not closely related groups. These three groups were previously not detected morphologically, but recent morphological studies have started to characterise them (see [238]). However, more morphological studies on scyliorhinids are necessary, as it remains unclear whether the Jurassic catsharks represent the oldest modern catsharks or whether they represent a convergently evolved extinct group. Additionally, there are still several undescribed small

shark specimens with dentitions suggesting that this group could have been more diverse than expected during the Late Jurassic in Europe.

Family Scyliorhinidae Gill, 1862 [239]
Genus †*Bavariscyllium* Thies, 2005 [55]
†*Bavariscyllium tischlingeri* Thies 2005 [55]

†*Bavariscyllium tischlingeri* is the only known species within this genus so far. The holotype is a holomorphic specimen, housed in the Jura Museum Eichstätt, which most probably comes from the lower Tithonian of the Eichstätt region. Additional holomorphic specimens from the Solnhofen Archipelago are stored in the collections of the Bayerische Staatssammlung für Geologie und Paläontologie, Munich (SNSB-BSPG 1878 VI 6), Staatliches Museum für Naturkunde Stuttgart, Germany (SMNS 96086), and Senckenberg Naturmuseum, Frankfurt, Germany (SMF P272). Isolated teeth of this genus also were recovered from Kimmeridgian deposits of Northern Germany [240].

Characterised by a slender body, this small shark presents rounded pectoral and pelvic fins and a caudal fin without a distinct ventral lobe. The anal fin is very low and elongated, extending from the anterior edge of the first dorsal fin almost to the posterior edge of the second dorsal fin. The dorsal fins are triangular and rounded, and the base of the first is located posterior to the pelvic fins (Figure 23). Both features (shape and position) are considered typical characters for the catshark family Scyliorhinidae [55]. The dental morphology of †*Bavariscyllium tischlingeri* is well established based on the holotype and isolated teeth from the Kimmeridgian of North Germany. Accordingly, the tooth crown and root display the characteristic morphology of scyliorhinids [55,240]. Both the skeletal morphology and dental characteristics of †*Bavariscyllium tischlingeri* thus strongly reinforce its assignment to Scyliorhinidae, making †*Bavariscyllium* one of the oldest holomorphic catshark species. However, the exact systematic position within carchrhiniforms need to be established in the future employing strict cladistics principles.

Genus †*Palaeoscyllium* Wagner, 1857 [102]

†*Palaeoscyllium* is a widespread taxon with a fossil record from the Middle Jurassic to the Early Cretaceous in Europe. Up to now, three species of †*Palaeoscyllium* have been identified: †*P. formosum* Wagner 1857 [102], †*P. tenuidens* Underwood and Ward, 2004 [220], and †*P. reticularis* Underwood and Mitchell, 1999 [241]. The earliest records of †*Palaeoscyllium* so far are based on isolated teeth referred to †*Palaeoscyllium* sp. and †*Palaeoscyllium tenuidens* from the Bathonian of England [220,234]. In the Solnhofen Archipelago, only †*Palaeoscyllium formosum* can be undoubtedly confirmed, which is represented by several well preserved and holomorphic specimens.

Based on its overall fin morphology, size, and position, Cappetta [230] synonymized †*Palaeoscyllium* with †*Corysodon* from the early Kimmeridgian of Cerin (France), attributing minor morphological differences to taphonomic processes, and placed the species within Orectolobiformes as *incertae familiae*. However, several authors subsequently rejected the merging of the two genera and their placement within the Orectolobiformes and supported conversely their inclusion within Carcharhiniformes (e.g., [241,242]). Interestingly, Thies and Candoni [242] used the body morphology and the position and size of the dorsal fins as characters to separate both taxa, contrary to what was established by Cappetta [230]. Leidner and Thies [243] agreed with Thies and Candoni [242] regarding the validity of both taxa and its assignment to Carcharhiniformes based on their revision of placoid scale morphologies. Currently, †*Palaeoscyllium* is considered a member of Scyliorhinidae, mostly because the tooth morphology resembles that of scyliorhinids [24]. However, the systematic position of this genus within Carcharhiniformes still is unclear as the position of the fins does not match that of true catsharks, as scyliorhinids are characterized by the midpoint position of the first dorsal fin base being posterior to the origin of the pelvic fins. However, in †*Palaeoscyllium*, the first dorsal fin base is above the origin of the pelvic fin. Moreover, the origin of the anal fin is usually anterior to the origin of the second dorsal fin in true catsharks, while in †*Palaeoscyllium*, it is posterior.

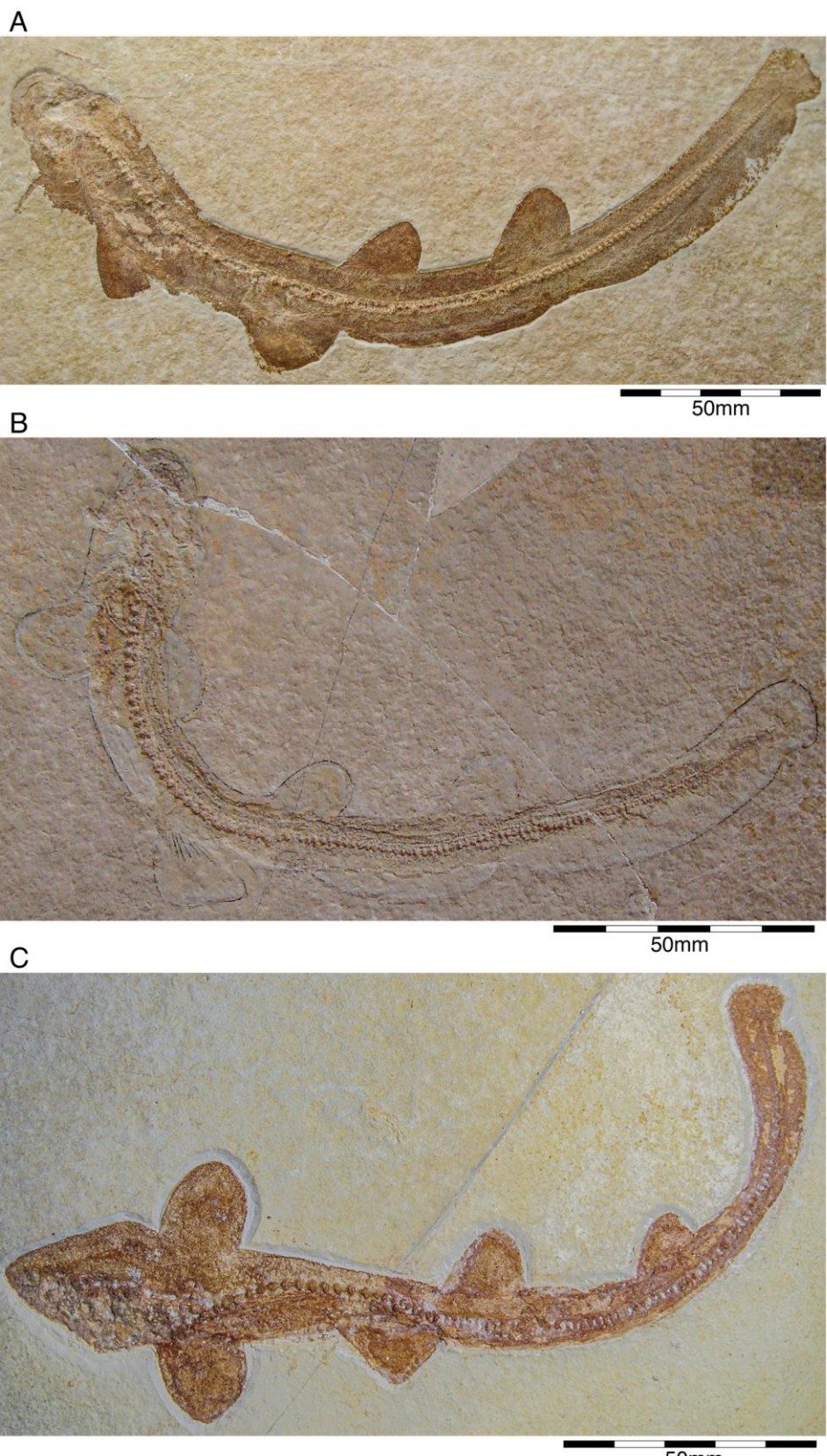

**Figure 23.** †*Bavariscyllium tischlingeri* Thies, 2005 [55] (**A**) Holotype specimen (JME SOS 4124) from the lower Tithonian of Eichstätt. (**B**) Holomorphic specimen (SMF P272) from the lower Tithonian of Eichstätt. (**C**) Holomorphic specimen (SMNS 96086) from the lower Tithonian of Eichstätt.

†*Palaeoscyllium formosum* Wagner, 1857 [102], *non* Marck, 1863 [236]

Based on a specimen with a very crushed and incompletely preserved cranial region from the lower Tithonian of Solnhofen, which is housed in the Bayerische Staatssammlung für Paläontologie und Geologie in Munich (SNSB-BSPG AS I 1365) (Figure 24A). Additional specimens of this species from the Solnhofen Archipelago are also stored in the same collection (SNSB-BSPG AS I 589a and SNSB-BSPG AS I 589b, part and counterpart) (Figure 24B), but also in the Jura-Museum Eichstätt (JME SOS 3151 (Figure 24C), JME SOS 7885), in the Staatliches Museum für Naturkunde Karlsruhe (SMNK, no catalogue number), in the Senckenberg Naturmusem Frankfurt (SMF P.171), but also in private collections (see [133]). Schweizer [66] presented an incomplete vertebral column from the upper Kimmeridgian of Nusplingen, stored in the Staatliches Museum für Naturkunde Stuttgart, Germany, which he identified as belonging to †*Palaeoscyllium* (SMNS 3695/9). Another possible specimen of †*Palaeoscyllium* from the lower Kimmeridgian of Cerin is housed in the fossil collections of the Musée des Confluences in Lyon under number MHNL 15 202 (coll. Thiollière). Additionally, †*Palaeoscyllium formosum* has been reported from the Oxfordian of Northern Germany [240], from the Kimmeridgian of Western France (as †*Parasymbolus octevillensis* Candoni, 1993 [244,245]), and England [246], and from the Kimmeridgian of Switzerland (as †*Palaeoscyllium* cf. *formosum* [82]). Isolated scales and teeth of †*Palaeoscyllium* sp. are known from the Kimmeridgian of Spain ([247], see also [248]).

The fossil record of †*Palaeoscyllium* also extends into the Cretaceous based exclusively on teeth from the Valanginian of Poland (†*Palaeoscyllium* sp. [249]), the Barremian of England (†*Palaeoscyllium* aff. *formosum* [250], †*Palaeoscyllium* sp. [234]), and from the Albian of England (†*Palaeoscyllium reticularis* as †*Parasymbolus reticularis* Underwood and Mitchell, 1999 [241]).

†*Palaeoscyllium formosum* occurs in the Solnhofen Archipelago in the upper Kimmeridgian and lower Tithonian [18,19,25,242,243].

The dorsal fins of †*Palaeoscyllium formosum* are rounded and triangular, and the base of the pelvic fins is ventral to the first dorsal fin. The well-developed anal fin is located below the second dorsal fin and is separated from the caudal fin, which displays a distinct ventral lobe [107] (Figure 24). The teeth are of general scyliorhinid shape, with a high central cusp and one or two lateral cusplets in mesial and distal positioned teeth, respectively. Longitudinal ridges are present on the tooth crown's labial and lingual face but more distinctly developed on the labial face. The root possesses a conspicuous lingual protuberance and expanded lobes [25]. Underwood and Ward [220] consider †*Palaeoscyllium* to be a member of Scyliorhinidae (true catsharks) based on the tooth morphology (see [238] for further morphological characters). However, the dental morphology of †*Palaeoscyllium* probably is more characteristic for basal ground sharks than they are for true catsharks pending further detailed phylogenetic analyses.

Egg Capsules of Carcharhiniformes

Egg capsules of extant scyliorhinids have a vase-shaped fusiform body with a slightly constricted waist, lateral extremely reduced to absent flanges as well as anterior and posterior pairs of horns of different length. The posterior horns are strongly curved inward tending to form a semi-circular edge. Capsule surface can be covered with longitudinal striation [128]. The horns may merge into coiled tendrils. Fossil remains, summarised under the parataxonomic ichnogenus †*Scyliorhinotheca* Kiel et al., 2013 [251], are reported from Cenozoic marine deep-water sediments of the USA and New Zealand. So far, one specimen of a catshark egg capsule is known from the Upper Jurassic Plattenkalks of the Solnhofen Archipelago. It is part of the collection of U. Resch and currently under description. The tiny fossil shows the typical vase-shaped outline with the constricted waist (Figure 25). This egg capsule is the oldest representative of †*Scyliorhinotheca* known so far.

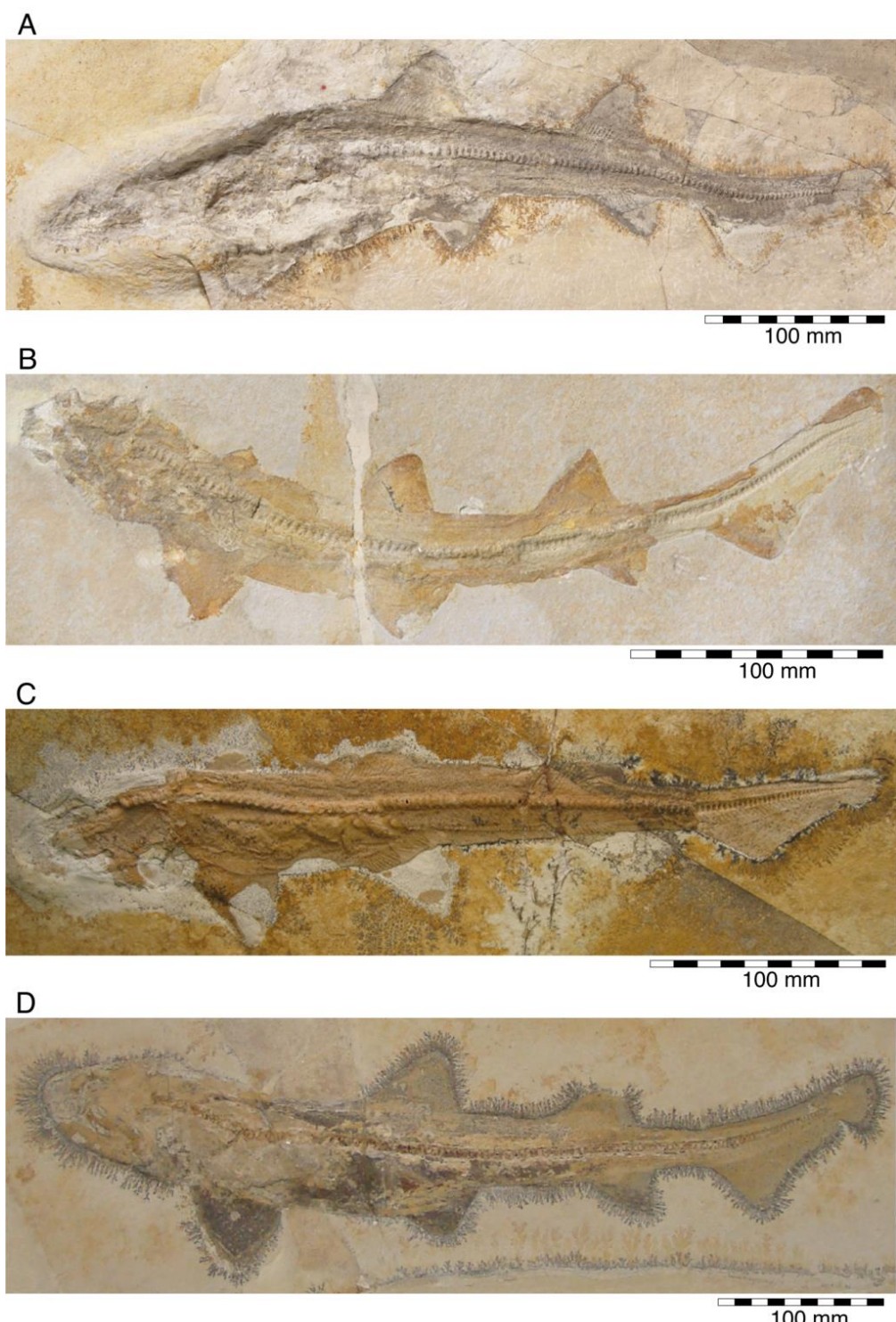

**Figure 24.** Specimens of †*Palaeoscyllium formosum* Wagner, 1857 [102]. (**A**) Holotype specimen (SNSB-BSPG AS I 1365) from the lower Tithonian of Solnhofen. (**B**) Holomorphic specimen (SNSB-BSPG AS I 589 A) from the lower Tithonian of Solnhofen. (**C**) Holomorphic specimen (JME SOS 3151) from the lower Tithonian of Solnhofen. (**D**) Holomorphic specimen (SMNK, without number).

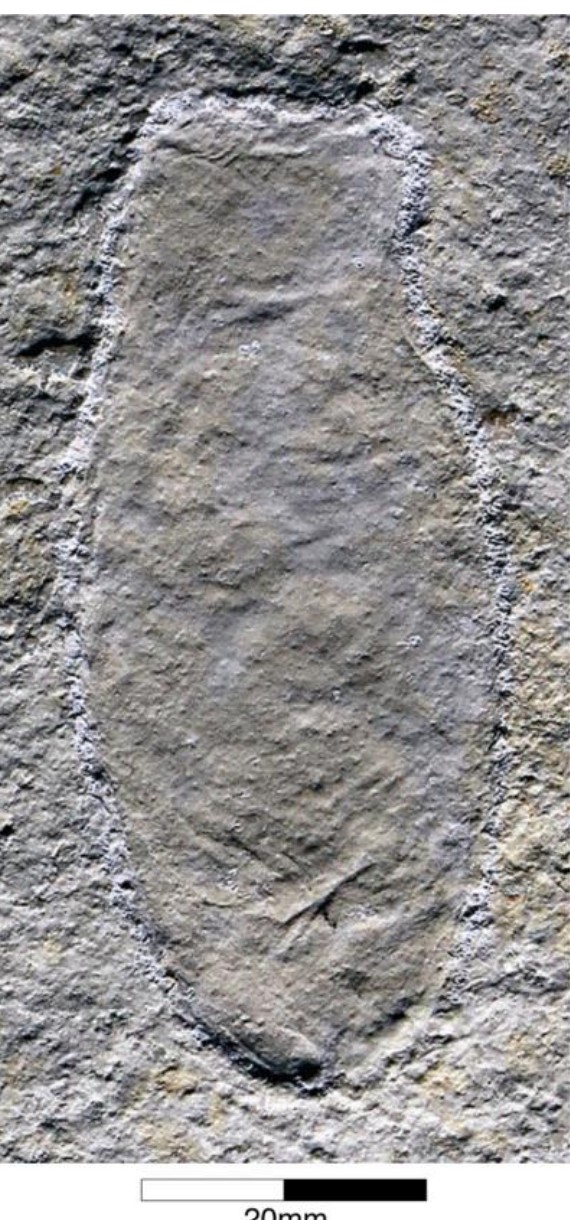

**Figure 25.** Scyliorhinid type egg capsule. †*Scyliorhinotheca* Kiel et al., 2013 [251] (collection Resch, JMS 456) from the Solnhofen Archipelago.

4.2.7. Lamniformes

Order Lamniformes Berg, 1958 [252]

The order Lamniformes currently comprises 15 extant species in 11 genera and 8 families [186]. Although lamniform sharks are well known from Early Cretaceous strata [249,251–254], their first occurrence in the fossil record is still heavily debated, with several authors proposing a Jurassic origin placing †*Palaeocarcharias* de Beaumont 1960 [51] as the oldest representative of this group [51,53,255] (but see Applegate [256], who considered this fossil shark to be a transitional species between orectolobiforms and lamniforms).

*Incertae familiae*
Genus †*Paleaocarcharias* de Beaumont, 1960 [51]

The monotypic genus †*Palaeocarcharias* is well known from several articulated skeletons from the Late Jurassic [18,19,51,257]. However, its first occurrence in the fossil record can be traced back to the Bathonian (Middle Jurassic) of England, based on a single but

well-preserved tooth of a yet unidentified species of †*Palaeocarcharias* [258]. Additionally, three isolated teeth from the Forest Marble Formation at Waton Cliff and the Taynton Limestone at Huntsman's Quarry (England) might also be affiliated with this genus, providing further evidence for the presence of the genus in the Bathonian. However, these teeth are heavily abraded, hampering a specific identification, which were thus only tentatively assigned as †*Palaeocarcharias* sp. by Underwood and Ward [220], also see [259].

Small lamniform shark teeth from the lowermost Berriasian (Early Cretaceous) of the Czech Republic also were assigned to †*Palaeocarcharias* and thus would indicate that †*Palaeocarcharias* crossed the Jurassic-Cretaceous boundary [260]. However, the material from Kurovice Quarry is poorly preserved and does not exhibit any diagnostic features typical for †*Palaeocarcharias* (see above, but also [25,261]). On the contrary, it shows several features that contrast with what is known from †*Palaeocarcharias stromeri* (e.g., broad, and rather stout crown lacking a sigmoidal profile in lateral view, the presence of lateral cusplets, and lingual crown face with long vertical folds) and, therefore, do not warrant the assignment of this material to †*Palaeocarcharias*. Thus, the occurrence of this group appears to be restricted to the Jurassic only.

†*Palaeocarcharias stromeri* de Beaumont, 1960 [51]

†*Palaeocarcharias stromeri* is the only nominal species described for the genus, and it is known from several well-preserved skeletons and isolated teeth from the early Tithonian of Germany and France [4,18,19,257]). The species was first described by de Beaumont [51] based on three specimens from the Solnhofen Archipelago including the holotype and a paratype, which are currently housed in the Jura-Museum Eichstätt (holotype JME SOS 2294; paratype 2, uncatalogued) (Figure 26), while the Naturwissenschaftliche Sammlungen der Philosophisch-Theologischen Hochschule, Eichstätt houses paratype 1 (also uncatalogued). Additional specimens from the Solnhofen archipelago also are stored in the Jura Museum Eichstätt (JME SOS 2216 a and b, part and counterpart), the Bayerische Staatssammlung für Paläontologie und Geologie, Munich (SNSB-BSPG 1964 XXIII 156), the Bürgermeister Müller Museum, Solnhofen (uncatalogued), the Fossilien und Steindruck-Museum, Gunzenhausen (FSM 719), Urweltmuseum Neiderhell (uncatalogued), and in several private collections.

†*Palaeocarcharias stromeri* was a small to medium-sized shark, reaching a total length of about 1 m. The body plan of this species is fusiform, considered to be rather dorso-ventrally flattened and poorly streamlined, similar to extant carpet sharks (*Orectolobus* spp.). The pectoral fins are rather large and rounded. Two almost equally sized dorsal fins are inserted on the level of the pelvic fin and slightly anterior to the anal fin, respectively. The anal fin is small and placed close to the caudal fin. The heterocercal tail is elongated, with a narrow and strip-like dorsal lobe and a semi-oval ventral lobe. The head is broad and round and bears barbels and cephalic lobes, a character regarded as an autapomorphy of recent sharks of the family Orectolobidae [257] (Figure 26). The subterminal mouth bears teeth resembling the lamniform tooth morphology with a linear gradient monognathic heterodonty. The anterior teeth are narrow, high-crowned, and show a sigmoidal profile in lateral view. The cusp becomes gradually lower and broader, and the profile less sigmoidal in more posterior teeth. Overall, its teeth have long, narrow, straight mesial and distal shoulders, which nearly reach the base of the root lobes on the labial side. Lateral cusplets are absent. Both the labial and the lingual faces of the crown are convex. The labial face is smooth in anterior teeth, whereas the lingual face exhibits short vertical folds along the basal margin. In lateral teeth, vertical folds are on both the lingual and labial faces and can reach up to the apex of the crown. The crown is separated from the root by a high neck on the lingual side. The root is bifid with two well-developed root lobes and a V-shaped interlobe area. The lingual protuberance of the root bears a nutritive furrow, and a row of small foramina is parallel to the crown-root junction.

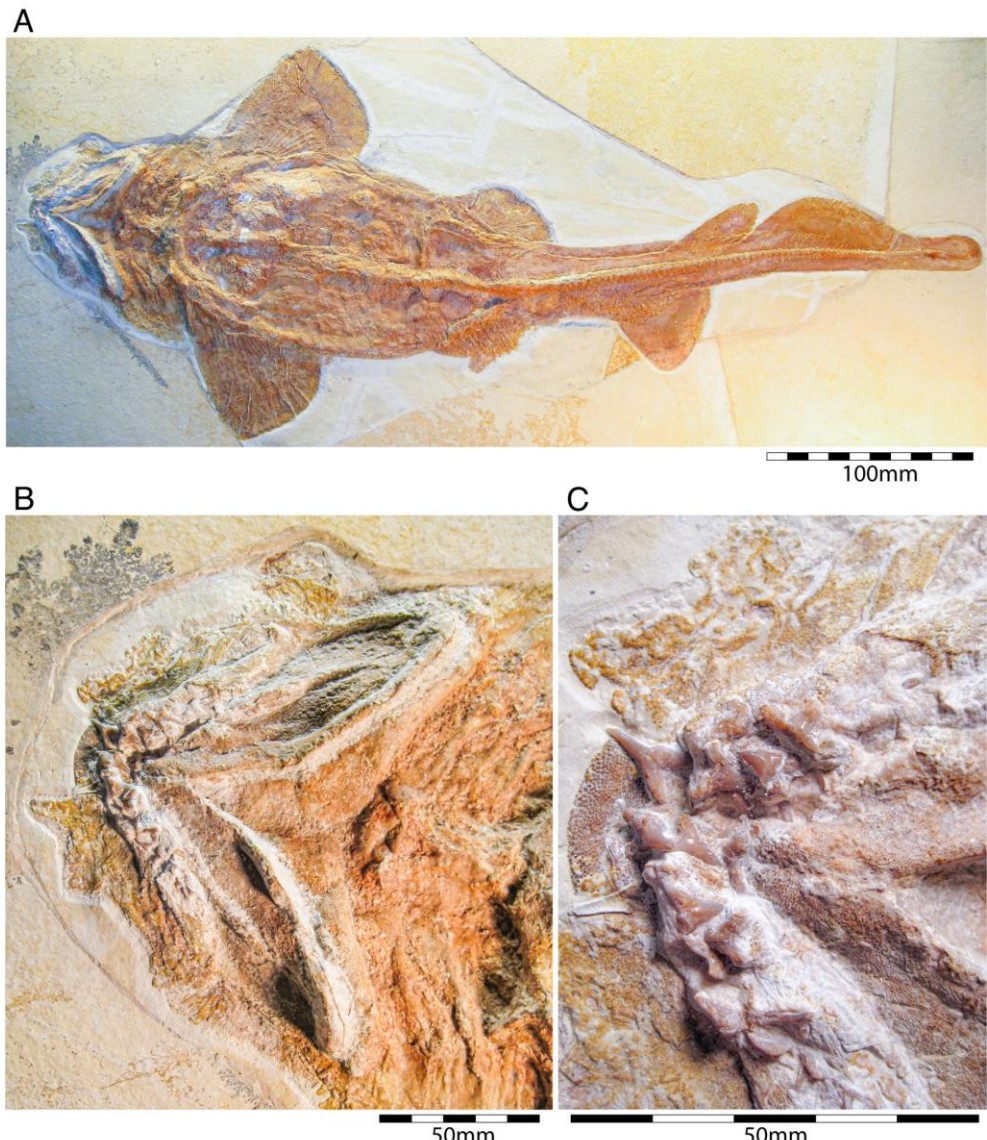

**Figure 26.** †*Palaeocarcharias stromeri* de Beaumont, 1960 [51], (**A**) Holotype specimen (JME SOS 2294) from the lower Tithonian of Eichstätt. (**B**) Close up of the cephalic regions. (**C**) Close up of the anterior portion of the jaws.

The presence of both orectolobiform and lamniform characters has led to an ongoing debate about the phylogenetic position of this taxon since its first description. In the original description, †*Palaeocarcharias stromeri* was regarded as a basal lamniform shark, representing an intermediate between Jurassic orectolobiforms and modern lamniforms [51]. Compagno [20], however, assumed that †*Palaeocarcharias* was not a basal lamniform shark but an orectolobiform based on its body shape, body size, and position of its fins. In a very detailed study on the tooth morphology of †*Palaeocarcharias stromeri*, Duffin [53] acknowledged the similarities in overall body form between Orectolobiformes and †*Palaeocarcharias* but emphasized the similarities in tooth morphology between †*Palaeocarcharias* and Lamniformes, concluding that †*Palaeocarcharias* was indeed a basal lamniform shark that descended from Orectolobiformes. In a conference contribution, Applegate [256] also regarded †*Palaeocarcharias* as an intermediate between Orectolobiformes and Lamniformes but suggested it to be a member of its own order †Palaeocarchariformes.

The first cladistic analysis conducted on †*Palaeocarcharias* supported this assumption, proposing that †*Palaeocarcharias* was the sister group to a clade consisting of Carcharhiniformes and Lamniformes, and thus should be put in its own family and order [257].

However, this phylogenetic analysis was shown to be inconclusive, as a significant number of phylogenetically uninformative characters was included in the analysis. Consequently, the analysis failed to detect synapomorphies uniting †*Palaeocarcharias* to any of the other galeomorph shark groups [255]. However, a unique tooth histology pattern shared by lamniform sharks and †*Palaeocarcharias* reinforces previous interpretations that †*Palaeocarcharias* is the oldest known representative of the order Lamniformes [255,260]. At this time, the phylogenetic interrelationships of †*Palaeocarcharias stromeri* remain ambiguous and cladistic analyses with additional galeomorph characters will be needed to address this issue.

4.2.8. Hexanchiformes

Superorder Squalomorphii Compagno, 1977 [21]
Order Hexanchiformes de Buen, 1926 [262]

Hexanchiformes is traditionally considered a primitive clade of sharks, as this group presents features that also occur in more ancient groups, including an otic process on the palatoquadrate that articulates with the postorbital process (see [263]), by the lack of fusion between the right and left pectoral fin halves, the little hyomandibular support of the lower jaws, and the reduced calcification of the vertebral centra. However, the fossil record of this group traces back only 190 Ma ago [24,264], which contradicts their ancient status.

Two extant monophyletic groups within Hexanchiformes are currently recognized, chlamydosleachids and hexanchids. Within hexanchids, only three genera are accepted as valid, *Hexanchus* Rafinesque, 1810 [265], *Notorynchus* Ayres, 1855 [266], and *Heptranchias* Rafinesque, 1810 [265], which show slight anatomical variations including the number of gill slits and cusplets or serrations on the lateral teeth. The fossil record includes many extinct genera, especially from the Cretaceous and Cenozoic that generally can be included in any of the three extant families. Jurassic taxa such as †*Crassodontidanus* Kriwet and Klug, 2011 [267], †*Notidanoides* Maisey, 1986 [268], and †*Pachyhexanchus* Cappetta, 1980 [269] with a stratigraphic range from the early Pliensbachian (Early Jurassic) to the early Tithonian (Late Jurassic), conversely, were assigned to the family †Crassodontidanidae [54,267].

Family †Crassodontidanidae Kriwet and Klug, 2011 [267]
Genus †*Crassodontidanus* Kriwet and Klug, 2011 [267]
†*Crassodontidanus serranus* Kriwet and Klug, 2011 [267]

This extinct hexanchiform species is known from the Nusplingen deposits and its based only on dental remains [267]. The holotype is housed in the Staatliches Museum fur Naturkunde Stuttgart under catalogue number SMNS 3695/10 (Figure 27D,E). Teeth of †*Crassodontidanus serranus* are characterized by the combination of a main cusp with a serrated mesial cutting edge and deep and rounded root with oblique basal face. Overall, the root morphology is similar to that of †*Notidanoides* presenting a pseudopolyaulacorhize vascularization (Figure 27).

Genus †*Notidanoides* Maisey, 1986 [268]
†*Notidanoides muensteri* (Agassiz, 1843 [171])

This species is the only representative of extinct crassodontidanids known from skeletal remains in the Upper Jurassic Plattenkalks. Described by Agassiz [171] based on an isolated tooth from the Kimmeridgian deposits of Streitberg in Franconia (Bavaria). The oldest known fossil records come from the Sinemurian (Lower Jurassic) of Switzerland (Ticino) and were described as '*Notidanus*' *arzoensis* by de Beaumont [51].

Up to now, only two more or less complete skeletons of this species have been recovered. The first specimen originated most probably from early Tithonian of the Eichstätt areas in the Solnhofen Archipelago and was figured by Beyrich et al. [270] (see also [54]). For this specimen, Wagner [103] introduced the new species †*N. eximius*, to which he also assigned an isolated tooth from the lower Tithonian of Daiting (Bavaria). Unfortunately, this almost complete specimen was lost during the World War II. Schweizer [66] described in detail a second, slightly disarticulated specimen from the Kimmeridgian (Upper Jurassic) Plattenkalks of Nusplingen (Baden-Württemberg) that is housed in the Paläontologische

Sammlung Universität Tübingen. He assigned it to '*Notidanus*' (Hexanchidae) based on the characteristic tooth morphology.

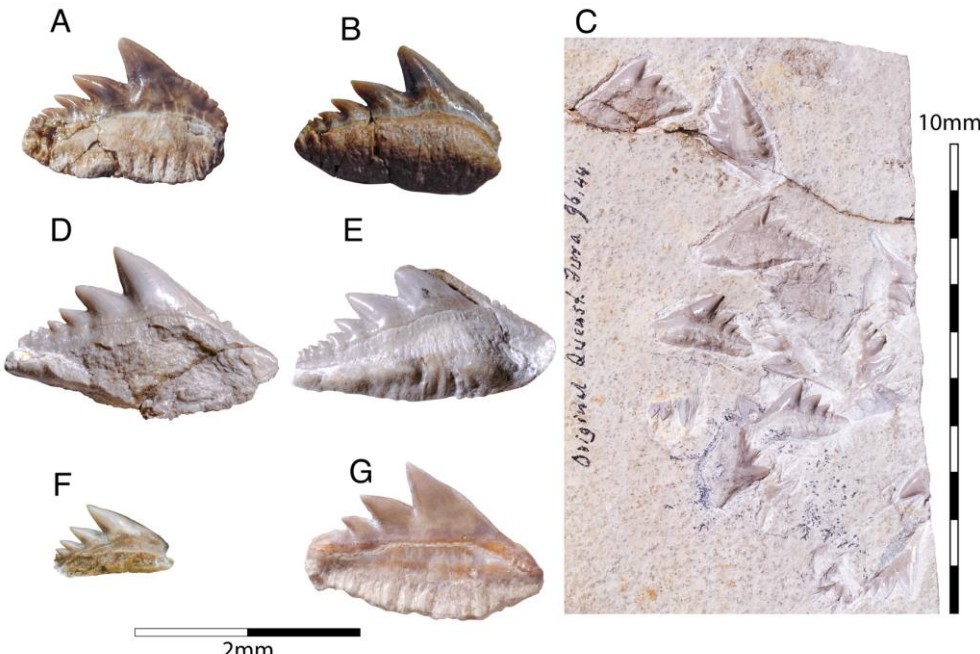

**Figure 27.** †*Crassodontidanus serranus* Kriwet and Klug, 2011 [267]. (**A**) Holotype specimen labial view (SMNS 3695/10) from the late Kimmeridgian of Nusplingen. (**B**) Lingual view. (**C**) Disarticulated teeth (GPIT 81512) from the Kimmeridgian of Nusplingen. (**D**) Close up of a tooth in labial view. (**E**) Close up of a tooth in lingual view. †*Notidanoides muensteri* (Agassiz, 1843 [171]). (**F**) Teeth of labial view (JME SOS 2213) from the Tithonian of Eichstätt. (**G**) Teeth in lingual view (SNSB-BSPG 1989 X 12) from the Kimmeridgian of Daiting.

Pfeil [271] placed all Jurassic combtooth sharks (hexanchiforms) into the genus †*Eonotidanus* (Pfeil, 1983 [271]), which was established for †*Notidanus contrarius* Münster, 1843 [272]. However, the holotype of †*N. contrarius* must be considered lost and the genus †*Eonotidanus* thus cannot be assumed to be valid. Because of this, the hexanchiform genus †*Paranotidanus* Ward and Thies, 1987 [273] was proposed for the Jurassic combtooth, teeth, which is currently considered invalid (*nomen nudum*) see [19].

Subsequently, Maisey and Wolfram [274] and Maisey [268] presented detailed description of its cranial morphology based on the Nusplingen specimen and compared it with other hexanchiforms (Figure 28), introducing the genus †*Notidonoides* Maisey, 1986 [268] for this fossil since *Notidanus* (Cuvier, 1817 [275]) represents a junior synonym of *Hexanchus*.

†*Notidonoides* displays a shorter distance between the pectoral and pelvic fins, which is not more than twice the distance between pelvic and anal fins according to the figure provided by Beyrich et al. [270] (see also [274] (figure 2G)). Furthermore, the teeth lack serrations on its anterior teeth, along with an overall reduction in serrations in all teeth compared to teeth of *Notorynchus* (four to five) and *Hexanchus* and *Heptranchias* (eight or more), and less labio-lingual compression [24]. †*Notidanoides* also is easily distinguished from extant hexanchoids by a larger number of lower lateral tooth rows, lack of synchronized tooth replacement, and the present strongly calcified vertebra centra (Figure 28). Additionally, the dorsal fin of †*Notidanoides* is inserted above the mid-region of the anal fin while it is located posterior to the pelvic fins in extant species [274].

As with modern species, the positions of the internal carotid arteries are marked by shallow transverse grooves. Unfortunately, it is impossible to determine whether the internal carotid had separate openings or shared a common entrance (extant hexanchids present two foramina slightly posterior to the 'basal angle' and the articular surface for

the palatoquadrate orbital process) due to incomplete preservation. The orbital process is poorly preserved, although its position is suggested by a bump posterior to the symphysis and the orbital articulation probably was positioned at the centre of the orbit compared to living hexanchids [268]. This and the presence of a process in the quadrate portion of the palatoquadrate suggest that the jaw suspension in †*Notidanoides* most closely resembled that of *Hexanchus* and *Notorynchus* (Figure 28B).

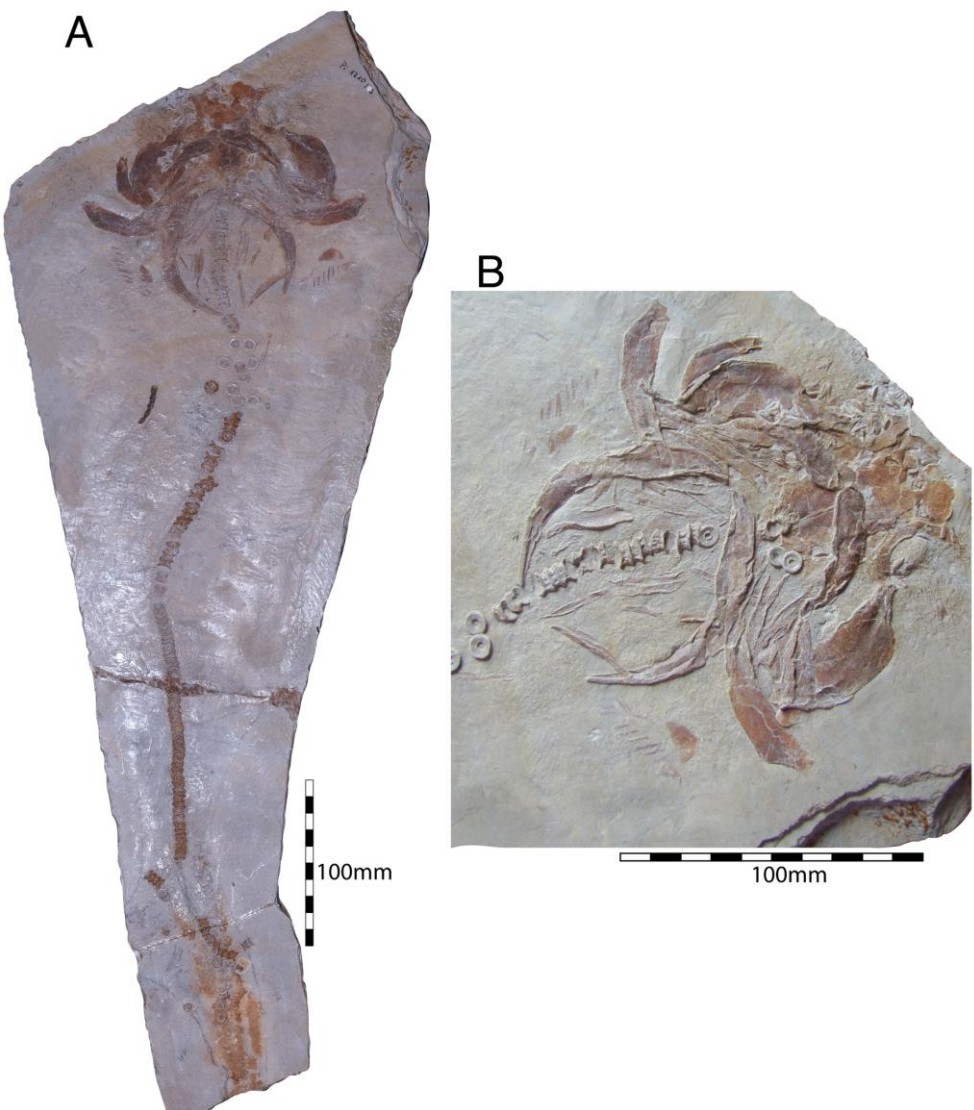

**Figure 28.** †*Notidanoides muensteri* (Agassiz, 1843 [170]). (**A**) Incomplete specimen (GPIT Pi 1210/3) from the Kimmeridgian of Nusplingen. (**B**) Close up of the cephalic region.

4.2.9. Squatiniformes

Order Squatiniformes de Buen, 1926 [262]

Presently this order includes a single family (Squatinidae) and genus (*Squatina*) with 23 species considered valid. This group of sharks is characterized by a dorso-ventrally flattened body, dorsally located eyes, and laterally placed gill slits that expand ventrally. Furthermore, their cervical vertebrae present an expansion of the basiventral processes, and they exhibit well-developed pectoral fins that are detached from the head and overlap with their pelvic fins and the lack of spines. The fossil record of squatiniforms extends well into the Jurassic. De Carvalho et al. [69] assigned the oldest skeletal and holomorphic specimens, which are known from the Upper Jurassic Plattenkalks of Southern Germany

(Nusplingen and Solnhofen Archipelago), to a different family, †Pseudorhinidae based on several dental and skeletal features. This group most likely represents stem group members of Squatiniformes pending further detailed phylogenetic analyses.

Family †Pseudorhinidae de Carvalho et al., 2008 [69]
Genus †*Pseudorhina* Jaekel, 1989 [276]

Complete skeletons of this extinct group are relatively abundant in the lithographic limestones of Southern Germany (Figure 29), especially in the deposits of the Nusplingen lagoon. Additionally, more rare specimens that generally are smaller than those from Nusplingen, also occur in the Solnhofen Archipelago. Currently, two species, †*Pseudorhina alifera* (Münster, 1842 [50]) (Figure 29A) and †*Pseudorhina acanthoderma* (Fraas, 1854 [64]) (Figure 29B–D) are considered valid. While both species probably occurred in the Solnhofen Archipelago, †*P. acanthoderma* seemingly was the only angel shark inhabiting the Nusplingen lagoon. A third species, †*P. speciosa* von Meyer, 1856 [277], currently is considered synonymous with †*P. alifera* (see [69,240]).

Both species differ in size, body proportions, and dental morphologies according to our current knowledge. However, it is necessary that all small specimens from the Late Jurassic Plattenkalks previously attributed to †*P. speciosa*, but also all other specimens should be re-examined to better understand the distribution of †*Pseudorhina* species in the different localities.

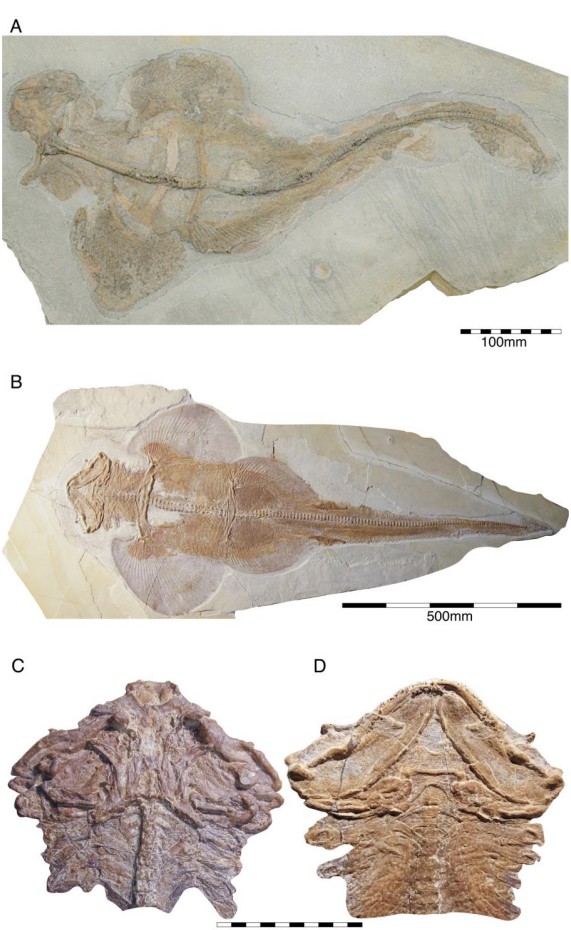

**Figure 29.** Specimens of †*Pseudorhina* Jaekel, 1989 [276]. (**A**) Holotype specimen of †*P. alifera* (Münster, 1842 [50]) (SNSB-BSPG AS VII 3) from the lower Tithonian of Eichstätt. (**B**) †*Pseudorhina acanthoderma* (Fraas, 1854 [64]) (SMNS 80431/20) from the upper Kimmeridgian of Nusplingen. (**C**) Close up of the cephalic region in dorsal view of †*P. acanthoderma* (SMNS 86214/41) from the Kimmeridgian from Nusplingen. (**D**) Close up of the cephalic region in ventral view of †*P. acanthoderma* (SMNS 80431/20).

Originally, all Late Jurassic Plattenkalk angel sharks were assigned either to *Thaumas* (Münster, 1842 [50]) or *Squatina* Dumeril, 1806 [278]. The name *Thaumas*, however, is pre-occupied by a lepidopteran and a protist, subsequently considered synonymous with †*Pseudorhina* [24]. Jaekel [276] introduced the genus †*Pseudorhina* for the Plattenkalk squatiniforms, which, in the following, however, was a junior synonym of *Squatina* (e.g., Cappetta [279]). However, both genera present numerous morphological differences supporting their taxonomic validity. In †*Pseudorhina*, e.g., the antero-posterior length of the first basiventral is equal to that of the second (Figure 29C,D), the postorbital process is directed laterally, the orbital process is nearly vertical, and the anterior spool of the first vertebra centrum is not reduced, see [69,280] (Figure 29C,D). Furthermore, the teeth of *Squatina* present higher cups, which are separated from the heels and a narrower apron separated from the basal margin of the crown, whereas teeth of †*Pseudorhina* display broader triangular cusps and very oblique and short heels, with a broad apron united to the basal labial margin of the crown [24]. It is noteworthy that both species are very similar in their skeletal anatomy and no skeletal characters have been identified to differentiate them (Figure 29) (see [69]). Differentiation of both species currently is based mainly on dental features.

### 4.2.10. †*Protospinax*

*Incerti ordinis*
Family †Protospinacidae Woodward, 1918 [232]
Genus †*Protospinax* Woodward, 1918 [232]

†*Protospinax* is one of the most enigmatic taxa in regards to its systematic position within elasmobranch fishes. Isolated teeth are known from several Middle to Late Jurassic deposits of Europe, including Germany [211,217], France [281], Luxembourg [282], Poland ([283–285], Spain [247], Switzerland [161], and Southern England [214,220,256], with the oldest record dating back to the Toarcian (Early Jurassic) [282,286]. Its synonymy with the Cretaceous taxon †*Pseudospinax* Müller and Diedrich, 1991 [287], extending the stratigraphic range of this genus into the Late Cretaceous (see [24,288,289]). However, the similarities between both genera probably are only superficial, and some authors still consider the genus †*Pseudospinax* to be valid [197,290]. Therefore, it remains unclear whether †*Protospinax* crossed the Jurassic-Cretaceous boundary.

†*Protospinax annectans* Woodward, 1918 [232]

Several holomorphic specimens of †*Protospinax annectans* have been recovered from the lower Tithonian of the Solnhofen Archipelago [18,19,232,291,292]. In addition to the holotype and the paratype, which are both housed in the Natural History Museum of London under collection numbers NHMUK PV P 8775 (Figure 30) and NHMUK PV P 37014, respectively, several additional holomorphic specimens are stored in the Bayerische Staatssammlung für Paläontologie und Geologie, Munich (SNSB-BSPG 1963 I 19) (Figure 30B), the Jura-Museum Eichstätt (JME SOS 3386), the Museum Bergér, Eichstätt (MB 14-12-22-1), Fossilien- und Steindruckmuseum Gunzenhausen (FSM 727a,b), the Harvard Museum for Comparative Zoology, USA (MCZ 278; MCZ 6394), Urweltmuseum Neiderhell (uncatalogued), and in several private collections [133,176].

†*Protospinax annectans* is a very dorso-ventrally flattened, medium-sized shark reaching a total length of up to 1.5 m. It has two large and rounded pectoral fins, which are not fused with the head (Figure 30). Like modern squaliform and heterodontiform sharks, it exhibits two dorsal fin spines supporting each of the two dorsal fins. Both dorsal fins are almost equal in size and are placed far posterior on the body caudally to the pectoral girdle. The anal fin is missing. The teeth of †*Protospinax* are small, not reaching more than 3 mm in size and are typical for a crushing-type dentition [24,230]. They closely resemble batomorph teeth but can be distinguished from them by a lingually displaced root [202].

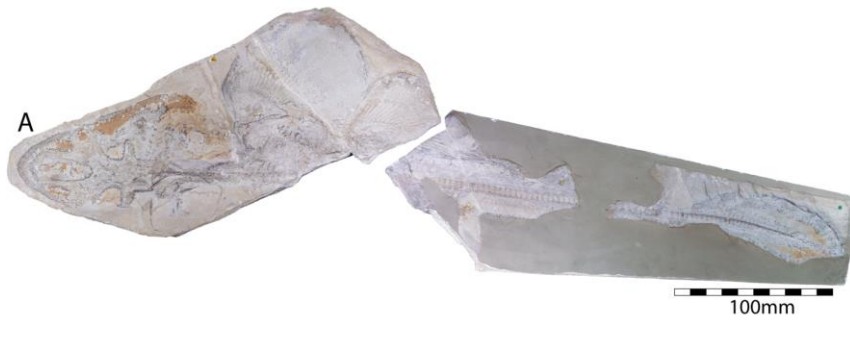

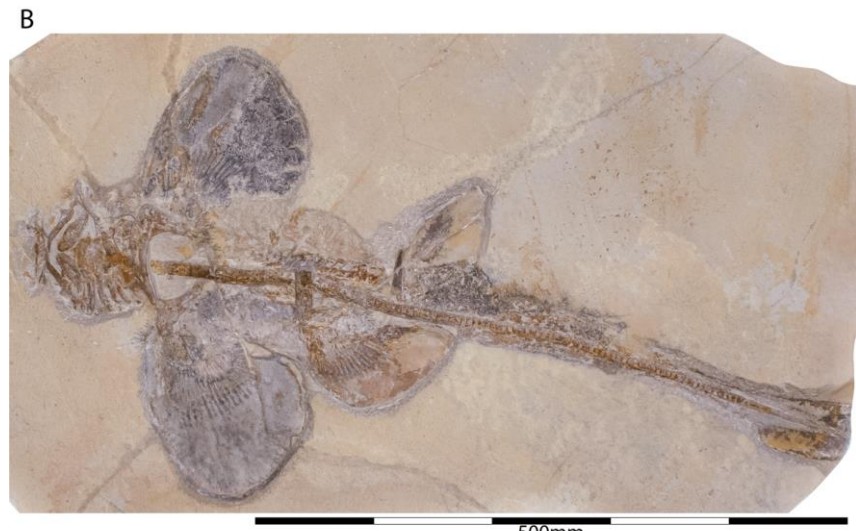

**Figure 30.** †*Protospinax annectans* Woodward, 1918 [232]. (**A**) Holotype specimen (NHMUK PV P 8775) from the lower Tithonian from Solnhofen. (**B**) Holomorphic specimen preserved in ventral view (SNSB-BSPG 1963 I 19) from the lower Tithonian of Eichstätt.

Since its original description in the early 20th century, the relationships of †*Protospinax annectans* within other sharks, but also batomorphs, have been controversially debated. Woodward [232] put it in its own family (†Protospinacidae) and noted its close relationship to squaliform and echinorhiniform sharks ('Spinacidae'), within a clade that also comprised all batomorphs ('Tectospondyli'). Since then, †*Protospinax* has been considered the ancestor of batomorphds, squalomorph sharks, or some galeomorph sharks [293], to be the sister group of all extant and extinct batomorphs [80], an intermediate group between squalomorph sharks and batomorphs [20], a stem group Squalomorphii [294], a sister group to squaliform sharks [295], or a squaliform shark [217].

De Carvalho and Maisey [292] were the first to examine the phylogenetic interrelationship of †*Protospinax annectans* based on morphological data using modern cladistic approaches. Their analysis resolved †*Protospinax* as a highly derived squalomorph shark that represents the sister group to recent 'hypnosqualeans', a clade comprising angel sharks, saw sharks, and batomorphs. However, recent molecular analyses reject the Hypnosqualea hypothesis and resolve batomorphs as the sister group to a clade consisting of the sister groups squalomorphs and galeomorphs (Selachimorpha), rather than batomorphs being highly derived squalomorph sharks [192,237,296,297]. In an attempt to calibrate the molecular clock and date divergence times for squalomorph sharks, Flammensbeck et al. [298] combined molecular and morphological data (dental characters) of 85 extant and fossil taxa, including several species of †*Protospinax*. The analysis recovered the group as a paraphyletic clade, appearing in close relation to squatiniform sharks. However, the support values for this node were low, and the phylogenetic placement of †*Protospianx* continues to be ambiguous.

Besides all the controversy about its phylogenetic position, Maisey [291] added further taxonomic confusion to †*Protospinax*, by transferring the holotype and the paratype to two separate genera. He placed the holotype within the batomorph genus †*Belemnobatis* Thiollière, 1852 [229], while he erected a new genus and species †*Squalogaleus woodwardi* (Maisey, 1976 [291]) for the paratype.

Initially, Maisey [291] considered †*Squalogaleus* to be a small, spinate galeomorph shark based on the loss of the palatoquadrate otic process. However, it was later allocated to squalomorph sharks based on the presence of an orbitostylic jaw articulation [299]. Maisey [291] and Cappetta [230] illustrated the teeth of †*Protospinax annectans* and †*Squalogaleus woodwardi*, pointing out morphological differences between both taxa. Primarily, the root vascularization is assumed to differ significantly between both species, i.e., teeth of †*Protospinax annectans* exhibit a nutritive root groove on the basal face of the crown that is limited to the lingual part of the root. Teeth of †*Squalogaleus woodwardi*, conversely, exhibit an open nutritive grove with a central foramen in antero-lateral teeth, whereas the groove is only partially open in more lateral teeth. Underwood and Ward [220] studied several thousand teeth of †*Protospinax* from the Bathonian of England and erected three new species. They noted that the tooth morphology, especially the root vascularization, shows differences between different species but can even vary within the same species, indicating the presence of a certain degree of heterodonty in tooth root morphology [220]. Thies [217] regarded †*Squalogaleus woodwardi* as a juvenile †*Protospinax annectans*, a broadly accepted conclusion nowadays [18,19,24,292]. Further studies on the in situ dentitions of holomorphic specimens are needed to provide more information about the heterodonty found in †*Protospinax* (e.g., dignathic heterodonty, ontogenetic heterodonty) and might allow the identification of more species or maybe even genera.

The similarities between †*Protospinax* and batomorphs certainly are the result of convergent evolution due similar ecological adaptations rather than being indicative of a shared origin [18,19,227]. Nevertheless, many controversies still surround this enigmatic Mesozoic elasmobranch and its phylogenetic placement [300]. Nonetheless, †*Protospinax* still is widely used as an important key taxon to calibrate phylogenetic trees and make inferences about the evolutionary history of cartilaginous fishes [298,301]. Therefore, it is important to establish its systematic position beyond any doubt to resolve its phylogenetic placement within the chondrichthyan tree of life and to re-evaluate the evolutionary history of squalomorph sharks and elasmobranchs in general.

### 4.2.11. Batomorphii

Superorder Batomorphii Cappetta, 1980 [302]

Batomorphs represent a monophyletic group placed within the neoselachians [20,21]. However, numerous phylogenetic uncertainties persist. Perhaps the most important issue is their relationship with sharks, which remains unresolved, as a morphology-based analysis suggests that batomorphs are derived from sharks closely related to pristiophorids and squatinids forming the Hypnosqualea [296,303]. Molecular analyses, on the other hand, recover a mutual monophyletic arrangement between both groups in a sister relation, rejecting the Hypnosqualea hypothesis [192,296,304–308].

Overall, this group of flattened cartilaginous fishes with gill openings in ventral positions, greatly enlarged pectoral fins attached to the sides of the head and a fusion of the anterior vertebrae into a synarcual are the most diverse group of extant neoselachians, with about 665 described species [233]. However, this was not always the case; with a fossil history that can be traced back to the late Early Jurassic [202], the diversity and assemblage of batomorphs has changed throughout their evolutionary history. During the Jurassic, batomorph diversity was low and included only a few genera, of which at least four are known from skeletal remains [24,309]: †*Asterodermus* Agassiz, 1936 [171] from Southern Germany, †*Spathobatis* Thiollière, 1852 [229] from Southern Germany and France [19,25,132,175,228,308–311], †*Belemnobatis* Thiollière, 1852 [229] from France, †*Kimmerobatis* Underwood and Claeson, 2017 [312] from England, and an un-

named batomorph from Argentina [313]. Of these genera, at least two, †*Asterodermus* and †*Spathobatis*, are present in the Solnhofen Archipelago. Despite the relatively good fossil record of these genera, most diagnostic features are based on their similar tooth morphologies so far.

*Incerti ordinis*
*Incertae familiae*
Genus †*Asterodermus* Agassiz, 1936 [171]
†*Asterodermus platypterus* Agassiz, 1936 [171]

†*Asterodermus* is a monotypic batomorph genus that was created by Agassiz [171] based on a small fossil ray specimen from the lower Tithonian of Kelheim, which is housed in the Natural History Museum London under collection number NHMUK P 12067 (Figure 31A). This specimen lacks the complete cephalic region. Already in 1836, Agassiz [171] introduced †*Asterodermus platypterus* for this head-less specimen using the star-like outline of the placoid scales, the presence of slender ribs, and the structure of the pectoral and pelvic girdles to characterize the taxon. The placoid scales, however, are embedded apically in the sediment so that only the star-like bases of the roots are visible, which therefore does not represent any useful morphological feature as the placoid scale basis in other batomorphs, such as †*Spathobatis* Thiollière, 1852 [229] also is star-like in shape [243]. Because of this and the lacking cranium including the dentition, there are thus currently no unambiguous characters that differentiate †*Asterodermus platypterus* from other Jurassic batomorphs, causing uncertainties and debates.

The specimen exhibits a well-developed pectoral girdle and crescent-shaped, broad metapterygia. The incompletely preserved pelvic girdle is narrow and relatively straight. Eight pairs of ribs are present. A small dorsal fin spine is visible, but the dorsal fins are not preserved in the holotype.

Leidner and Thies [243] depicted a tooth of a putative †*Asterodermus* specimen (SNSB-BSPG 1960 XVIII 56) from the lower Tithonian of Zandt near Denkendorf assuming that all Upper Jurassic batomorphs from Southern Germany belong to this genus based on placoid scale morphology, an opinion followed by Kriwet and Klug [18]. However, as mentioned above, the placoid scale morphology might not be a good character to identify fossil batomorphs from the Late Jurassic. Kriwet and Klug [19] reconsidered their previous interpretation and accepted the occurrence of both †*Asterodermus* and †*Spathobatis* in the Solnhofen Archipelago. Thies and Leidner [25] described further teeth of putative †*Asterodermus* specimens, although they noted that the genus affiliation remains unclear due to the incomplete holotype lacking the dentition, resulting in further ambiguities related to this genus.

The teeth figured by Thies and Leidner [25] are mesio-distally expanded, possess a low central cusp, a sparsely developed apron, and a distinct uvula. A mesiodistal ridge is separating the labial and lingual faces of the crown. The root is holaulacorhize. Up to now, the tooth morphology is the primary means to distinguish between Late Jurassic batomorphs, making a clear identification of †*Asterodermus* impossible as these features are missing in the holotype. Underwood and Rees [314] therefore suggested to consider †*Asterodermus* a *nomen dubium* until diagnostic skeletal characters are described.

Based on an study in progress (Türtscher et al., in prep.), in addition to the holotype stored in the Natural History Museum London (NHMUK P 12067), there are other specimens possibly belonging to †*Asterodermus platypterus* (Figure 31B–E), housed in the Bayerische Staatssammlung für Paläontologie und Geologie, Munich (SNSB-BSPG 1952 I 82, SNSB-BSPG 1960 XVIII 56, SNSB-BSPG 1964 XXIII 577, SNSB-BSPG AS I 1377 (see also [25]) and SNSB-BSPG AS I 1378 (part and counterpart), and SNSB-BSPG AS XIX 502), the Jura-Museum Eichstätt (JME SOS 3647 (as †*Asterodermus* in Frickhinger [133,176] and Kriwet and Klug [18], but as †*Spathobatis* in Kriwet and Klug [19]), JME SOS 2212 (see also [25]), JME SOS 2212a (J.T., pers. obser.)), and the Museum Bergér in Eichstätt (two specimens, one as part and counterpart (no catalogue numbers; J.T., pers. obser.); one figured by Frickhinger [133] as †*Aellopos*/†*Spathobatis*. A nearly complete specimen is on

display in the Tierpark and Fossilium Bochum (labelled as †*Asterodermus*; no catalogue number (J.T., pers. obser.)), and several additional specimens are stored in diverse private collections [175].

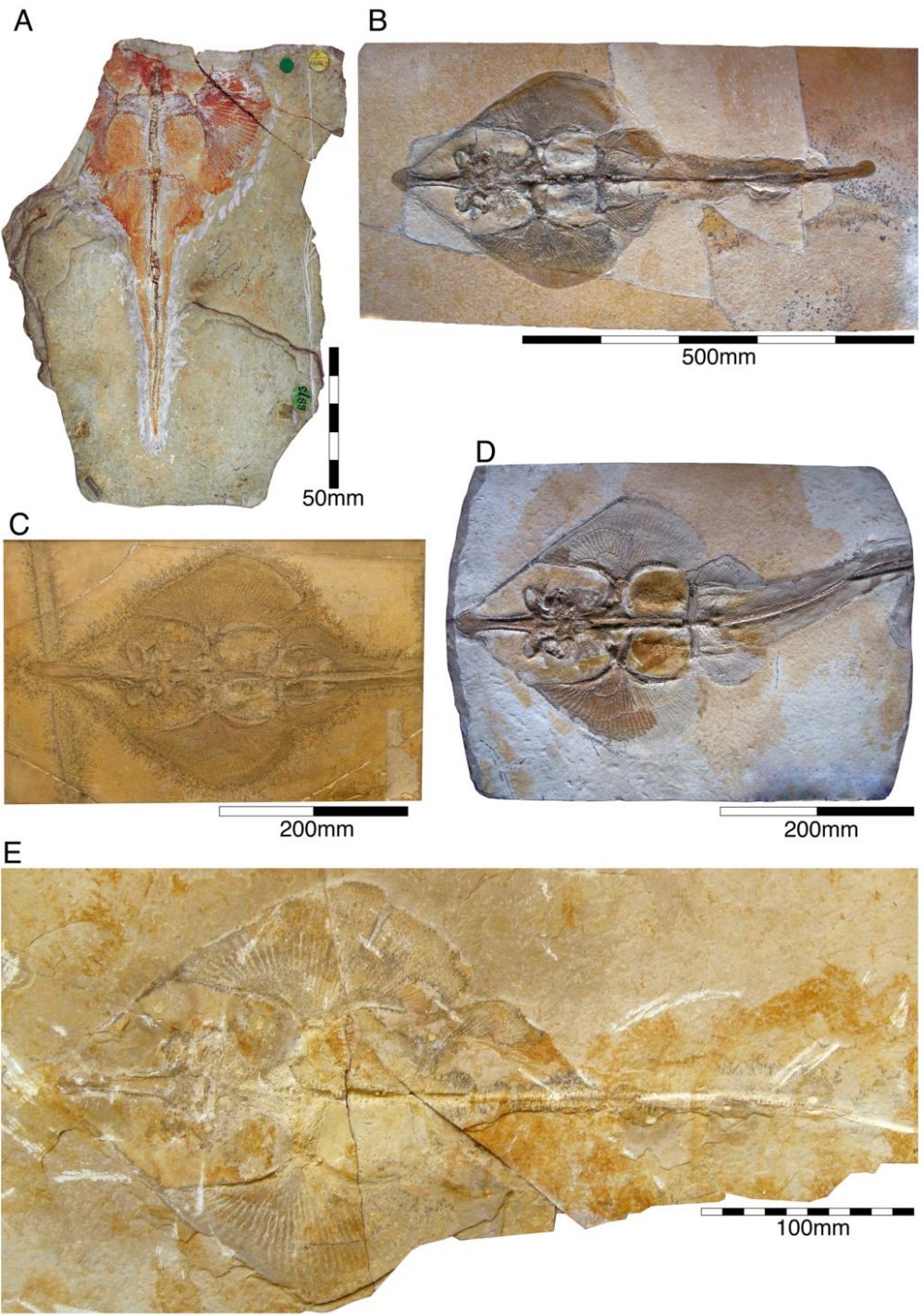

**Figure 31.** †*Asterodermus platypterus* Agassiz, 1936 [171]. (**A**) Holotype specimen (NHMUK P 12067) from the lower Tithonian of Kelheim. (**B**) Holomorphic specimen (JME SOS 2212 A) from the lower Tithonian of Blumenberg. (**C**) Partial specimen (SNSB-BSPG AS I 1378) from the Tithonian of Kelheim. (**D**) Partial specimen (JME SOS 3647) from the Tithonian of Birkhof. (**E**) Holomorphic specimen (SNSB-BSPG 1964 XXIII 577) from the Tithonian of Solnhofen.

Genus †*Spathobatis* Thiollière, 1852 [229]

Remains of the batomorph †*Spathobatis* have been described from Early-Middle Jurassic to Early Cretaceous deposits of Europe [201,217,220,229,247,281,290,315–317]. It seemingly is a speciose genus with many species described on the basis of isolated teeth. However, the validity of many species within this genus remains obscure, e.g., †*Spathobatis moorbergensis* Thies, 1982 [216], which is from the Toarcian and Aalenian of northern Germany representing the putative earliest record of †*Spathobatis*. The generic affinity of this species, however, is not clarified as it possibly belongs to †*Belemnobatis* [203,220,248,311,318,319].

†*Spathobatis bugesiacus* Thiollière, 1852 [229]

The holotype of †*Spathobatis bugesiacus* from the upper Kimmeridgian of Cerin (France) is a holomorphic specimen that is very similar to that of †*Asterodermus platypterus*, with a broad pectoral girdle, crescent-shaped metapterygia, a narrow puboischiadic bar that is anteriorly arched and possesses postpelvic processes (Figure 32). Both species also exhibit ribs, star-shaped placoid scales in basal view, and small dorsal fin spines. The nasal capsules are oval and are set off from the rostrum. The rostrum is elongated and has a spatula-shaped rostral appendix.

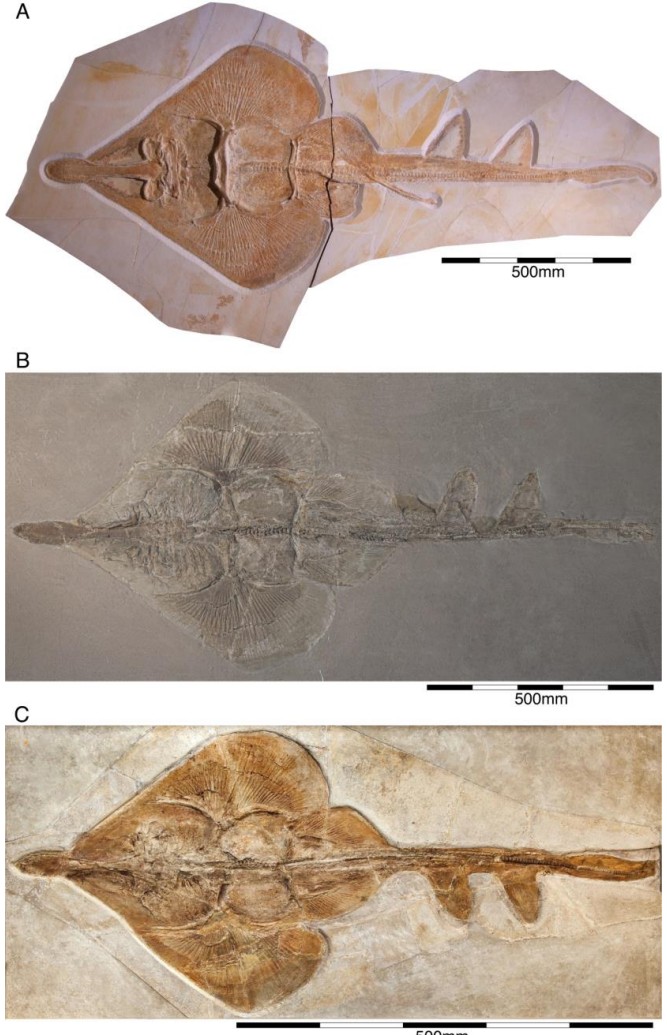

**Figure 32.** †*Spathobatis bugesiacus* Thiollière, 1852 [229]. (**A**) Holomorphic specimen (BMMS BK 3-1) from the lower Tithonian of Eichstätt. (**B**) Holomorphic specimen (SNSB-BSPG AS I 505) from the lower Tithonian of Eichstätt. (**C**) Holomorphic specimen (NHMUK PV-P-6010) from the Tithonian of Solnhofen.

The teeth of †*Spathobatis bugesiacus* also resemble those of †*Asterodermus platypterus* to a high degree. They are wider than high and have a broad and well-developed lingual uvula and a mesiodistally running ridge through the low apex separating the crown's labial and lingual faces. A distinct, peg-like apron is present, and the root is holaulacorhize [25]. However, †*Spathobatis bugesiacus* exhibits dental intraspecific variations, as teeth with a well-defined median cusp and a massive uvula are also known from this species [24,82,246,311]. This species also is known from holomorphic specimens in the Solnhofen Archipelago (see below). Isolated teeth of this species also were reported from the Kimmeridgian of England [246] and the Kimmeridgian of Switzerland [82].

A second species, †*Spathobatis morinicus* Sauvage [310], is known from a partially preserved skeleton from the Tithonian of northern France [311]. Otherwise, all other species only are known from isolated teeth from the Bathonian of England (†*Spathobatis delsatei* Underwood and Ward, 2004 [220]), from the Oxfordian of Spain (†*Spathobatis* sp.; [247]), from the Kimmeridgian of northern Germany (†*Spathobatis uppensis* Thies, 1982 [217]), and from the Tithonian of France (as †*Spathobatis uppensis* and †*Spathobatis* sp.; [281]). Cretaceous records based on teeth are known from the Valanginian of France (†*Spathobatis* sp.; [201]), from the Hauterivian of England (as †*Spathobatis rugosus* Underwood et al., 1999 [290]), from the Barremian of France (†*Spathobatis halteri* Biddle and Landemaine, 1988 [320]) and Spain (†*Spathobatis halteri* and †*Spathobatis* sp.; [316,317]), and the Albian of France (†*Spathobatis halteri*; [315]).

More or less well preserved specimens of †*Spathobatis bugesiacus* (Figure 32A–C) from the Solnhofen Archipelago providing abundant morphological features for future taxonomic and systematic analyses are housed in the Natural History Museum of London (NHMUK PV P 6010, NHMUK PV P 10934, and the cast NHMUK P 49149), the Bayerische Staatssammlung für Paläontologie und Geologie, Munich (SNSB-BSPG 1959 I 434, SNSB-BSPG AS I 505, SNSB-BSPG AS VII 1170, as well as a specimen without catalogue number), the Museum Bérger in Eichstätt (uncatalogued), and the Staatliches Museum für Naturkunde in Karlsruhe (uncatalogued). A juvenile specimen is housed in the American Museum of Natural History (AMNH 7506).

An exquisitely preserved adult specimen is housed in the Bürgermeister-Müller-Museum in Solnhofen (BMMS BK 3-1) and several more specimens are stored in diverse private collections [176].

Egg Capsules of Batomorphs

Egg capsules of extant batomorphs are characterized by a dorso-ventrally flattened, rectangular body outline with concave anterior and posterior margins, and inwardly directed horns at the anterior and posterior ends [128]. Length ratios and curvature of anterior and posterior horns are diagnostic for the genus. So far, three distinguishable species have been described from shallow marine Cenozoic deposits of Europe [321]. They are summarised under the parataxonomic ichnogenus †*Rajitheca* Steininger, 1966 [322]. Egg capsules of presumed batomorph origin are scarce in the fossil record from the Upper Jurassic Plattenkalks of the Solnhofen Archipelago. Just a few specimens are available in private collections until now [19]. All capsules have an overall length up to 135 mm and a width of about 60 mm (J.F., pers. obser.), which makes them, like the Plattenkalk holocephalian capsule †*Chimaerotheca schernfeldensis*, significantly larger than all other known fossil †*Rajitheca* species [321]. The horns are relatively short and curved inward. Conspicuous on all specimens are two clear bilateral constrictions in the overall rectangular body (Figure 33).

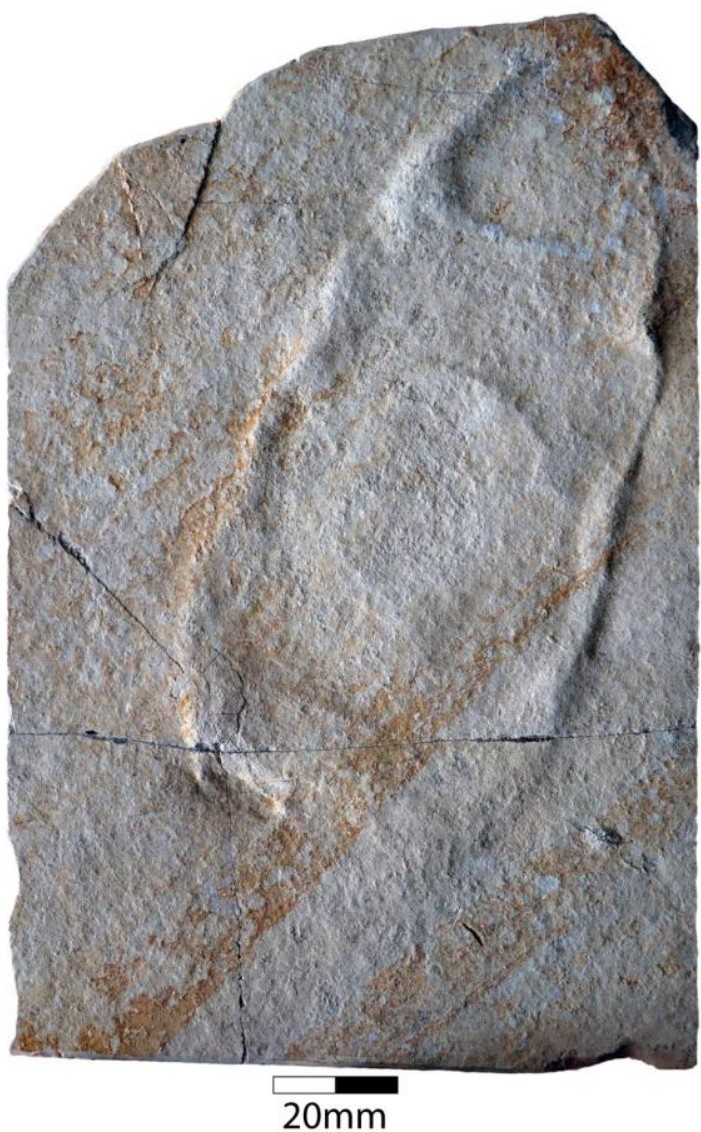

20mm

**Figure 33.** Putative batomorph type egg capsule from the Solnhofen Archipelago. †*Rajitheca* Steininger, 1966 [322] (private collection).

### 4.3. Faunal Relationships of the Solnhofen Archipelago Chondrichthyan Fauna

Considering that tropical reef environments accumulate enormous proportions of modern global biodiversity (e.g., [323]) due to their dynamic nature as barriers of dispersal (e.g., [324]), the Solnhofen Archipelago may have had represented a significant marine biogeographical region during the Late Jurassic. With more than 1600 species of plants and animals reported from these deposits [325], the Solnhofen Archipelago is one of the most speciose and well-studied fossil assemblages from the pre-Quaternary fossil record [326]. Despite this the faunal relationships of chondrichthyan fishes from the Late Jurassic Solnhofen Archipelago have remained poorly understood [327]. Here, we used the taxonomic and stratigraphic information gathered during the last decades to establish possible relationships of the Solnhofen Archipelago chondrichthyan assemblage with other Late Jurassic associations from Europe.

The non-metric multidimensional scaling (MDS) plot (Figure 34) indicates that the oldest locality from the Oxfordian (Moneva in NE Spain [245]) is the most dissimilar when compared to the rest of localities. In the present analysis, the Kimmeridgian localities are found to be widely scattered. The French locality of Cerin appears to be the most dissimilar one within this stage, presenting a reduced chondrichthyan fauna that consists of four

genera and only two (†*Spathobatis* and †*Phorcynis*) shared with the Solnhofen Archipelago. The localities of Porrentruy (Switzerland) and Octeville (France) are more similar to the Kimmeridgian–Tithonian Solnhofen Archipelago and Kimmeridge Clay. The Tithonian locality of Chassiron (France) appears to be closer to the Kimmeridgian localities of Octeville and Porrentruy than to the Solnhofen Archipelago and Kimmeridge Clay, although this may result from a lack of study (monographic effect) as there are only two studies that report on the chondrichthyan fauna from Chassiron [169,328]. The Solnhofen Archipelago and Kimmeridge Clay are found to be grouped closely together, but this may well be the result of a Lagerstätten effect (see, e.g., [329–331]). To further assess these faunal associations, we made a cluster dendrogram to observe the association between the localities (Figure 35).

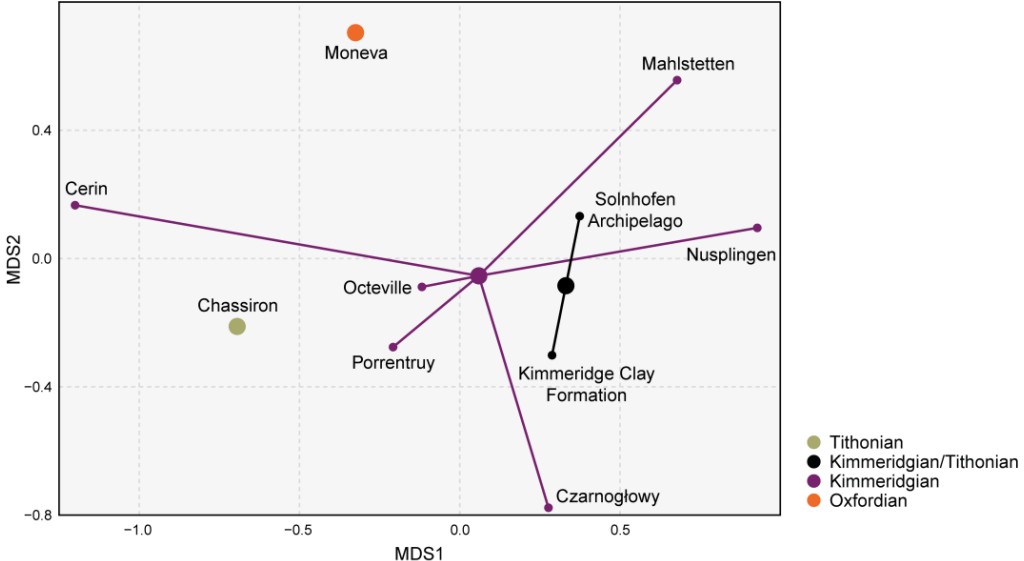

**Figure 34.** Multidimensional scaling dispersion plot, showing the dissimilarity relation between localities and geological times.

With the cluster dendrogram, and after estimating the number of clusters (three) (Figure 35B), we observe that the most dissimilar localities are Moneva and Cerin. Only two genera co-occur in Moneva and Cerin (†*Phorcynis* and †*Spathobatis*), which are also widely distributed across the other studied localities, suggesting that the (Moneva+Cerin) cluster is the result of shared absences of taxa (i.e., these localities present a higher number of absent taxa and because of that are the most different of the rest). The well-documented diversity of bony fishes of Cerin [5,81] suggests that, in the case of this locality, the differences are not a product of external factors (e.g., biases in sampling methods) and that the apparent lack of chondrichthyans is likely to be characteristic of this locality, which represents a restricted lagoonal setting [5].

The (Nusplingen+Mahlstetten) cluster groups two very different palaeoenvironments, with one being a protective lagoon (Nusplingen) and the other (Mahlstetten) representing an open marine environment. These localities have six genera in common, which are also widely distributed across the studied localities, suggesting again that this grouping also relate to the shared absences in the composition of taxa. Whether this is characteristic of these localities or the result of some bias remains uncertain, as the taxonomic classification of some fossil remains under discussion, such is the case of a possible †*Protospinax* sp. Specimen, some material collected from Nusplingen deposits (GPIT 1210/13), and with material form Mahlstetten is currently under study.

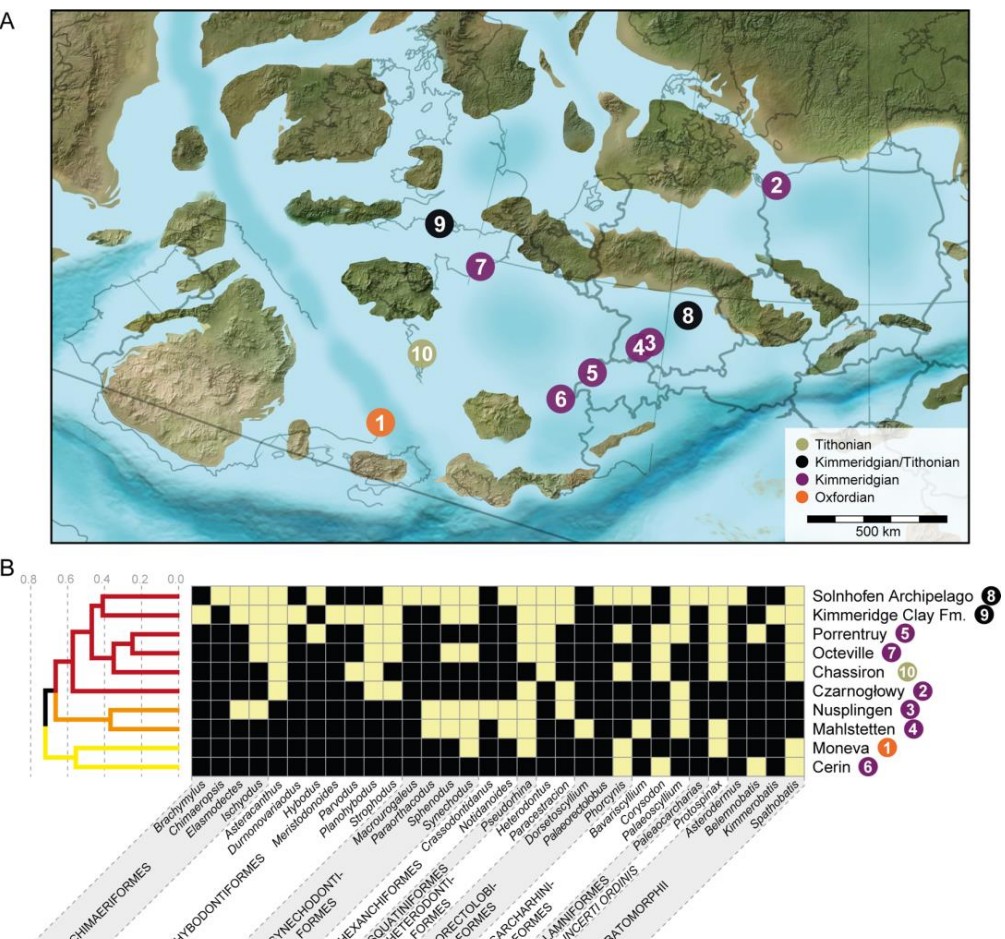

**Figure 35.** (**A**) Rough reconstruction of Europe during the Late Jurassic (150 Ma), depicting the position of the localities studied. Paleogeography of Europe Series © 2020 Colorado Plateau Geosystems Inc. Scottsdale Arizona. (**B**) Heat map and dendrogram showing the dissimilarity relations between geological areas based on the presence and absence of taxa. Colours in dendrogram indicate the assigned clusters in the analysis. Heat map colours: White (presence), black (absence). Numbers in the map refer to the localities in the heat map and dendrogram.

Finally, a large cluster comprising the localities of Czarnogłowy, Chassiron, Porrentruy, and Octeville, as well as the Kimmeridge Clay and Solnhofen Archipelago is recovered, with the Kimmeridge Clay and Solnhofen Archipelago subsenquently grouped within this cluster. The shared presence of the holocephalian (†*Ischyodus*), several hybodontiforms (†*Asteracanthus*, †*Planohybodus*, †*Strophodus*), and a batomorph (†*Spathobatis*) support this large aggregation. The similarity between these areas seems to be in part the result of the taxonomic level employed in the present study (genus). Whilst a more exclusive study unit could provide more refined results. Considering the heterogeneity of the fossil remains included in the present study (e.g., fin spines, teeth, fragmentary skeletal remains, and holomorphic specimens), and the taxonomic uncertainties surrounding them, especially the fragmentary remains, we consider that the genus was smallest study unit we could use without underestimating or overestimating the taxonomic diversity of these areas.

The Kimmeridge Clay and Solnhofen Archipelago concentrate most of the known Late Jurassic chondrichthyan diversity (Figure 35). However, this grouping does not imply their placement in a single biogeographic region, as both areas represent associations of composite faunas consisting of fossils derived from different localities, stratigraphic levels, and environments (e.g., [95,203,332]). Thus, the composite nature of these assemblages certainly affects the relative genus richness and diversity patterns, ultimately affecting the

similarity estimates between both areas. An example of this apparent similarity between these localities is the synechodontiform shark †*Sphenodus*. While this genus is relatively common in the Solnhofen Archipelago, only a single tooth of the species †*S. macer* has been reported from the Kimmeridge Clay up to now [203]. While the use of frequency similarity/dissimilarity indexes (e.g., Bray–Curtis dissimilarity) could provide additional information regarding this relation, a robust assessment of the frequency would be a limitation, considering the differences in sampling effort between these localities, along with the differences in the preservation probabilities between the areas or the approximation to the frequency (teeth belonging to the same individual) [327]. Another example of apparent similarity between these localities is that of the angel shark †*Pseudorhina* which is present in both Kimmeridge Clay and the Solnhofen Archipelago, but displays differences in the species composition, with †*Pseudorhina* frequens being yet only from the lower Kimmeridge Clay of the Weymouth area (known only by teeth) [203], while †*Pseudorhina acanthoderma* occurs in Southern Germany, this again suggest that our results are more laxus due to the taxonomic level of study. However, the uncertainties surrounding the taxonomic affinities of these taxa need to be considered (see [203,246,333]).

Among the groups included there are some that show interesting patterns like batomorphs, which seemingly show differences in their biogeographic distribution, with †*Asterodermus* (Solnhofen Archipelago), †*Kimmerobatis* (Kimmeridge Clay), and †*Belemnobatis* (Cerin and Porrentruy) being exclusive in their distribution, but with †*Spathobatis* being widely distributed across the European localities. However, the taxonomic affinities of these taxa remain understudied (Türtscher et al., in prep.).

The presence of numerous taxa with widespread distributions (present in at least three localities) (Figure 35) suggests that many components of the Late Jurassic chondrichthyan faunas had wider distributions, resulting in a homogenization of the chondrichthyan faunas. However, the absence of some of these widely distributed groups might suggest that some areas were strongly environmentally controlled. A similar pattern has been recovered by Kriwet and Klug [327] using a different approach, showing that a lack of ecological barriers might have caused a homogenization of Late Jurassic European chondrichthyan faunas.

## 5. Conclusions

Despite the steady study of the chondrichthyan faunas from the Plattenkalks of Southern Germany spawning from approx. 160 years, and the increase in the systematic studies of these biotas in the last 20 years, the taxonomy and systematics of many chondrichthyans groups from these time remains poorly understood. As the chondrichthyan diversity in the Solnhofen Archipelago continues to increase as new specimens and taxa are described, the need for comprehensive studies becomes more evident. Unfortunately, currently there is an increasing tendency towards macroevolutionary studies, which for poorly understood groups such as chondrichthyans is hampered by the uncertainties in the systematic interrelationships surrounding them, resulting in stagnation of such research endeavours. The presence of these groups (such as chondrichthyans) in localities with high preservation potential provides a unique opportunity for comprehensive studies and major breakthroughs in our understanding of the evolutionary history of chondrichthyan fishes in deep time and through time. After the specimen and bibliographic review, the present study found a total of 32 chondrichthyan genera in the Solnhofen Archipelago, comprising four holocephalians, eight hybodonts, and 20 neoselachians.

The included localities to which the Solnhofen Archipelago was compared show a relatively similar chondrichthyan fauna as suggested by the MDS analysis under the Sorensen index (under the R package 'vegan' summarises the dissimilarity between localities), which recovered most of the clusters within the 0.7–0.3 interval (0 = equal composition and 1 = completely different composition) (Figure 35), suggesting that several components of the Late Jurassic cartilaginous fish faunas, at least for the included areas, were widely distributed resulting a relatively homogenous in their composition, at least in Europe (also see [327]).

A major aspect that was not yet appreciated for Late Jurassic chondrichthyan assemblages in detail is the environmental setting of the various localities and correlated environmental adaptations of chondrichthyans (as has been carried out, e.g., for Middle Jurassic elasmobranchs from the UK by Underwood [334]), which also would lead to a better understanding of Late Jurassic chondrichthyan diversities. However, in order to do so more detailed stratigraphic information about the provenance of the Solnhofen Archipelago chondrichthyan association, is necessary, along with the inclusion of younger localities (Cretaceous), which go beyond the scope of the present work, but would be essential to better understand faunal compositions and dynamics on small temporal and spatial scales that would allow identifying possible diversity drivers and sustainers, such as migration patterns or cladogenetic events leading to speciation.

**Supplementary Materials:** The following supporting information can be downloaded at: https: //www.mdpi.com/article/10.3390/d15030386/s1, Table S1: R code for the faunal composition and table with taxa occurrences. Or upon request to the authors.

**Author Contributions:** E.V.-S.: conceptualization; investigation; data revision; analysis interpretation; writing; manuscript—editing; manuscript—review; figures. S.S.: conceptualization; investigation; data assemblage; analysis interpretation; writing; manuscript—editing; figures. J.T.: investigation; writing; manuscript—editing; figures. P.L.J.: investigation; writing; figures. A.B.: investigation; writing; data revision; analysis interpretation. F.A.L.-R.: investigation; methodology; formal analysis; analysis interpretation; figures. J.F.: conceptualization; writing; manuscript review; manuscript-revision; figures. J.K.: conceptualization; investigation; data assemblage; writing; figures; project administration; supervision; funding acquisition. All authors have read and agreed to the published version of the manuscript.

**Funding:** This research was funded in whole by the Austrian Science Fund (FWF) (P 35357, P 33820). For the purpose of open access, the author has applied a CC BY public copyright licence to any Author Accepted Manuscript version arising from this submission.

**Data Availability Statement:** All data used by the authors for the analysis are available in the supplementary materials, which can be downloaded at: https://www.mdpi.com/article/10.3390/d1 5030386/s1, or upon request to the authors.

**Acknowledgments:** Many people contributed to our studies in the last 25 years by providing access to collections or sharing information and their knowledge. It is impossible to name all and we apologize to everybody who is missing in the following list: U. Albert (Urwelt-Museum—Oberfränkisches Erdgeschichtliches Museum Bayreuth, Germany); G. Arratia and H.-P. Schultze (formerly Natural History Museum Berlin, Germany, now Biodiversität Institute & Natural History Museum, The University of Kansas); G. Bergér (Museum Bergér, Eichstätt, Germany); R. Böttcher (now retired) and E. Maxwell (Staatliches Museum für Naturkunde, Stuttgart, Germany); the late P. Forey, A. Longbottom (now retired), Z. Johanson, E. Bernard, and C. Duffin (all The Natural History Museum London, UK); H. Furrer (Paläontologisches Institut und Museum der Universität Zürich, Switzerland); M. Gerhäußer (Fossilien- und Steindruck Museum, Gunzenhausen, Germany); C. Ifrim (Jura-Museum Eichstätt, Germany); C. Kettler (Geologische Bundesanstalt, Vienna, Austria) for providing the photogrammetric models of SNSB-BSPG 1899 I 2 and SNSB-BSPG 2010 I 91 shown in Figures 10C and 12C,D, respectively; R. Kindlimann (Haimuseum und Sammlung, Aathal-Seegräben, Switzerland); S. Klug (Georg-August Universität Göttingen, Germany); M. Kölbl-Ebert (Munich, Germany); R. and B. Lauer (Lauer Foundation for Paleontology, Science and Education NFP, Wheaton, Illinois, USA) for generously providing a high-resolution photo of a holomorphic specimen of †*Chimaeropsis paradoxa* depicted in Figure 3B; the late M. Mäuser and O. Wings (Naturkunde Museum Bamberg, Germany); W. Munk (Staatliches Museum für Naturkunde Karlsruhe, Germany); S. Neiderhell (Urweltmuseum Neiderhell, Raubling, Germany); O. Rauhut, M. Kölbl, W. Werner (now retired), and M. Krings (all Bayerische Staatssammlung für Geologie und Paläontologie); U. Resch (Eichstätt, Germany); D. Thies (Leibniz Universität Hannover, Germany); R. Blakey (Colorado Plateau Geosystems Inc.) is thanked for permission to use his palaeogeographic map. Finally, we would like to thank the three reviewers of the present work for the comments and valuable insights. Open Access Funding by the Austrian Science Fund (FWF).

**Conflicts of Interest:** The authors declare no conflict of interest. The funders had no role in the design of the study; in the collection, analyses, or interpretation of data; in the writing of the manuscript, or in the decision to publish the results.

## Abbreviations

AMNH, American Museum of Natural History, New York, USA; BMMS, Bürgermeister Müller Museum Solnhofen, Germany; BT, Urwelt-Museum—Oberfränkisches Erdgeschichtliches Museum Bayreuth; GPIT, Geologisch-Paläontologisches Institut, Universität Tübingen, Germany; FSM, Fossilien- und Steindruck-Museum, Gunzenhausen, Germany; JME, Jura-Museum Eichstätt, Germany, (SOS indicates specimens from the Solnhofen area) (Scha indicates specimens from the Schamhaupten, Bavaria, Germany); LF, Lauer Foundation for Paleontology, Science, and Education, Illinois, USA.; MB, Museum Bergér, Eichstätt, Germany; MB.f., Museum für Naturkunde, Berlin, Germany; MCZ, The Harvard Museum for Comparative Zoology, Cambridge, Massachusetts, USA; MHNL, Musée d'Histoire Naturelle de Lyon, France; NMB, Naturkundemuseum Bamberg, Germany; NHMUK, The Natural History Museum, London, UK (P indicates specimens from the Pisces collection) (PV indicates specimens from the Palaeontological Vertebrate collection); PBP-SOL, Wyoming Dinosaur Center, Thermopolis, USA; SMF, Senckenberg Naturmuseum, Frankfurt, Germany; SMNK, Staatliches Museum für Naturkunde Karlsruhe, Germany; SMNS, Staatliches Museum für Naturkunde Stuttgart, Germany; SNSB-BSPG, Bayerische Staatssammlung für Paläontologie und Geologie, München, Germany.

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
