# Peer review of "A Synoptic Review of the Cartilaginous Fishes (Chondrichthyes: Holocephali, Elasmobranchii) from the Upper Jurassic Konservat-Lagerstätten of Southern Germany: Taxonomy, Diversity, and Faunal Relationships"

_diversity, doi:10.3390/d15030386_

Round 1

Reviewer 1 Report

This is a very well put together and well written paper summarising the knowledge, to date, of probably the most important Jurassic chondrichthyan fauna known. It is essentially ready to publish and there are not really any points that I think need including other then a few details:

There is no mention of holocephalans in the introduction where the clades present are mentioned.

It may be worth mentioning that the complete Chimaeropsis mentioned is currently under study (Chris Duffin)

The pre-Kimmeridgian species of Paracestracion seem to lack molariform teeth in adults too, so this may be a less derived state within the genus.

There are a number of small sharks with poorly preserved dentitions in collections- maybe mention that these suggest there may be more orectolobiformes and scyliorhinids in collections.

Not sure about putting Palaeocarcharias in the Lamniformes. You discuss its Orectolobus-like features and I would suggest it would better be a Geleomorph inc. sedis until more is known

amusing typo with "Angle sharks" rather than "Angel sharks"

I think that the discussion on the faunas could be enlarged a bit. Some of the species are represented largely by small juveniles- does that suggest early life stages of these were predominantly in environments nearby (ie shallow)?. 

The comparison to other sites is useful, but I assume that is total species count and does not take frequency into account. At least a discussion of that would be useful, as some species are common at some sites but rare elsewhere. 

You mention different environments and it is interesting (and something that could be discussed) that the Kimmeridge Clay is not dissimilar to Solnhofen despite being a cool/Boreal offshore mudstone with no reefs nearby. Even then, the faunas of the shallower water Kimmeridge Clay (Ringstead, N France) is very different to the laminated black mudstones of Kimmeridge (Etches Collection), with the latter having diverse hybodonts but no Strophodus.

Author Response

Reviewer.1 There is no mention of holocephalans in the introduction where the clades present are mentioned.

Response: It has been added (see, lines 50-55).

Reviewer.1 It may be worth mentioning that the complete Chimaeropsis mentioned is currently under study (Chris Duffin)

Response: This has been added (see, lines 299-300).

Reviewer.1 The pre-Kimmeridgian species of Paracestracion seem to lack molariform teeth in adults, too, so this may be a less derived state within the genus.

Response: This has been added (see, lines 981-982).

Reviewer.1 There are a number of small sharks with poorly preserved dentitions in collections- maybe mention that these suggest there may be more orectolobiformes and scyliorhinids in collections.

Response: A short paragraph making reference to these specimens has been added (see, lines 1069–1073 and 1158–1160).

Reviewer.1 Not sure about putting Palaeocarcharias in the Lamniformes. You discuss its Orectolobus-like features, and I would suggest it would better be a Geleomorph inc. sedis until more is known.

Response: We agree on the uncertainties surrounding Palaeocarcharias. However, the current knowledge suggests that a lamniform affiliation is adequate, pending a more detailed revision currently being carried out by the authors.

Reviewer.1 amusing typo with "Angle sharks" rather than "Angel sharks."

Response: We apologize for the mistake. It has been corrected through the text.

Reviewer.1 I think that the discussion on the faunas could be enlarged a bit. Some of the species are represented largely by small juveniles- does that suggest early life stages of these were predominantly in environments nearby (ie, shallow)? 

Response: We currently do not provide this as part of the discussion as currently, it's a work in process. 

Reviewer.1 The comparison to other sites is useful, but I assume that is the total species count and does not take frequency into account. At least a discussion of that would be useful, as some species are common at some sites but rare elsewhere.

Response: We included an example showing the possible frequency differences among the different sites used in the analysis. We also briefly discuss why the genus was used as a study unit and why frequency indexes were not included (see, lines 1828–1856).

Reviewer.1 You mention different environments, and it is interesting (and something that could be discussed) that the Kimmeridge Clay is not dissimilar to Solnhofen despite being a cool/Boreal offshore mudstone with no reefs nearby. Even then, the faunas of the shallower water Kimmeridge Clay (Ringstead, N France) are very different from the laminated black mudstones of Kimmeridge (Etches Collection), with the latter having diverse hybodonts but no Strophodus.

Response: Stumpf et al., 2022 (https://doi.org/10.3390/d14020085) suggest a wide distribution of Strophodus across Europe. This ubiquitous distribution indicates that there might not be a direct environmental factor limiting the distribution, but rather it seems to be responding to a lack of barriers.

Reviewer 2 Report

Dear editor and authors,

this manuscript is a very extensive and exhaustive synopsys on the Solhonfen Arcipelago chondrichthyan assemblage. The work is really huge and well-written. It is surely of international interesting. Methods, discussion and conclusion are solid.

There are only minor issue addressed in the annotated version of the manuscript attached. The more relavant one could be that in some part of the description of the taxa it is not clear if the authors are introducing novel considerations about the characters or are just summarising characters already considered in literature, so that I would recommend stressing out if these are new considerations/novelties introduced by their manuscript compared to the previous literature or adding the references if needed. 

I think that this work is a milestone in the literature of the Solnhofen chondrichthyan assemblage and must be published.

Best regards

Author Response

Typos were corrected through the text, references were added when needed, figure 34 no longer includes abbreviations. 

All corrections to the text are highlighted in yellow

Reviewer 3 Report

This paper gives a very thorough review on chondrichthyans from southern Germany, and provides many very delicate figures of fossils, some of which are even not described before. The very detailed review will be very useful for researchers who are interested in chondrichthyans of Germany. Since the author already have such comprehensive understanding of the chondrichthyans, it will make the paper much more informative if the author can extend the discussion part a bit. Comments and a few syntax errors are marked in the attached PDF.

Author Response

Corrections have been made across the text and are highlighted in yellow: 

R3: Does the author mean "extended distribution in Europe"? If the author meant the global distribution of chondrichthyans, the global data of the organism should be collected and analyzed.

Response: This has been modified (see, lines 32–33)

R4: This is a very interesting point, which is however not revealed in the discussion part. I suggest that the author can extend the discussion on this point.

Response: We removed this section from the present work, as we (the authors) would like to go into further detail on these point, in a future publication. Were this will be the main aspect of the parper. 

R3: "Considering that tropical reef environments and archipelagos accumulate enormous proportions of global biodiversity today (e.g. [318]) due to their dynamic as barriers of dispersal (e.g. [319]), it is possible that the Solnhofen Archipelago represents a significant proportion of the Late Jurassic marine global biodiversity...". This statement should be not able to be proven by the previous sentence. Although this area yields abundant well-preserved fossils, it still only represents a small part of the whole earth ecosystem.

Response: The statement has been modified (see, lines 1753–1756)

R3: Does "KITI" mean Kimmeridgian-Tithonian"? Those abbreviations should be explained in the text or caption. Also, the color of symbol of "Chassiron" is a bit too light.

Response: Figure 34 has been modified.

R.3: For the  "4.3" part, the author seems to want to compare the Solnhofen Archipelago fauna with other contemporaneous faunas from Europe. I suggest that the author can compare the similarity and difference among these faunas, and discuss the reason why some faunas are similar with each other (e.g., environmental factors, sampling bias, etc.).

Also, the situation of the 10 localities chosen in this studies might not be very known researchers from other areas. If they can be added in a paleogeographical map, it will be easier for reader to understand, and also we might be able to know why some faunas are clustered together.

Response: A map has been include and the discussion has been expanded, including some examples of the similarities and differences between the locations included (see, figure 35 and lines 1753-1856).

R3: It seems that all the 10 localities are from Europe, and faunas from these areas are divided into several clusters. Moreover, the similarity index value of fauna is lower than 0.5 for many clusters. Therefore, it is not very clear why the author concluded that composition of chondrichthyan fauna was quite uniform across Europe. 

Response: A clarification has been included regarding the scales and the measurements on the dendrogram (see 1873–1881)

R3: " Of the 32 genera, 21 (65.62%) are present in at least two localities and 18 are present in more than two localities" It seems a bit hard to understand these two sentences. Maybe the author can check these two sentences.

Response: This section was removed 

Round 2

Reviewer 3 Report

The paper has been revised accordingly and should be ready to be published after a minor revision of a few grammatical errors and misspellings. See line 1842, 1845, 1851, 1852, 1855, 1856, 1876, 1877.

Author Response

We thank the reviewer, for this second round of reviews. Typos were corrected